# TAME: A TASK-AGNOSTIC FRAMEWORK FOR ROBUST GRAPH NEURAL NETWORK EXPLANATIONS VIA STRUCTURAL MIXUP

## ABSTRACT

Graph Neural Networks (GNNs) have demonstrated remarkable performance across a range of applications involving graph-structured data, particularly in high-stakes domains. However, the opaque nature of their decision-making processes limits their trustworthiness and broader adoption. Existing post-hoc explanatory methods aim to improve explainability by identifying subgraphs that influence GNN predictions. Yet, these approaches are typically restricted to a specific type of task, such as classification with discrete decision boundaries or regression with continuous ones, which limits their general applicability. In this work, we propose TAME, a unified, task-agnostic framework for GNN explanation that addresses both the limitations of task-specific methods and the distribution shift caused by subgraph extraction. Our approach integrates contrastive learning into the Graph Information Bottleneck (GIB) framework, enabling consistent explanation across both classification and regression tasks. Furthermore, we introduce a novel mixup strategy built upon graph pooling, which generates in-distribution explanations through hard structural perturbations. Extensive experiments on diverse tasks demonstrate that TAME achieves state-of-the-art performance in generating robust and interpretable explanations across both synthetic and real-world datasets.

## 1 INTRODUCTION

Graph Neural Networks (GNNs) are well-suited for processing graph-structured data (Scarselli et al., 2008) and are widely applied to tasks such as community detection (Huang et al., 2018), traffic flow prediction (Lei et al., 2022), recommendation systems (Fan et al., 2020; Du et al., 2022), and molecular modeling (Gasteiger et al., 2021; Liu et al., 2023). Despite their success, the "black-box" nature of GNN decision-making hinders their adoption in high-stakes domains including healthcare (Choi et al., 2020), fraud detection (Dou et al., 2020), and drug discovery (Qu et al., 2025).

To address this challenge, recent research has focused on improving model explainability through model-level (Yuan et al., 2020) or instance-level (Ying et al., 2019; Luo et al., 2020) post-hoc explanatory methods, which aim to uncover the underlying decision logic of GNNs by identifying explanatory subgraphs. Based on the Information Bottleneck (IB) principle (Tishby et al., 2000; Tishby & Zaslavsky, 2015), a previous work (Wu et al., 2020) developed the Graph Information Bottleneck (GIB) framework, which formulates the explanation task as follows:

$$\arg \min_{G^*} I(G, G^*) - \alpha I(G^*, Y), \tag{1}$$

where the objective aims to maximize the mutual information $I(G^*, Y)$ between the explanation $G^*$ and the ground-truth label $Y$, while minimizing $I(G, G^*)$ to constrain the explanation size from the original graph $G$, with $\alpha$ balancing the two terms. Under the GIB framework, GFlowExplainer (Li et al., 2023) learns to select informative nodes that preserve predictive relevance while ensuring compactness. MATCHExplainer (Wu et al., 2023) adopts a non-parametric approach, matching shared subgraph patterns to identify concise explanatory subgraphs.

However, minimizing $I(G, G^*)$ encourages the extraction of $G^*$ by removing label-irrelevant information from $G$, which may result in a distribution shift and lead to the out-of-distribution (OOD) problem (Wang et al., 2021; Yuan et al., 2021). To alleviate the distribution shift,

MixupExplainer (Zhang et al., 2023) blends the explanatory subgraph with a label-irrelevant subgraph to generate augmented in-distribution graph instances. ProxyExplainer (Chen et al., 2024) leverages graph autoencoders to reconstruct the explanatory and label-irrelevant subgraphs, and fuses them to generate proxy graphs. Despite mitigating the OOD issue, existing methods still exhibit the following limitations, as illustrated in Figure 1. First, soft-mask-based explanatory methods struggle to effectively drive edge weights toward a binary form, resulting in mixup graph $G_{\text{soft}}^{(\text{mix})}$ that entangles explanatory and non-explanatory regions with comparable importance, rather than forming a clean composition of explanatory and label-irrelevant subgraphs. Second, existing mixup strategies introduce uninterpretable edges when connecting explanatory and label-irrelevant subgraphs. For example, MixupExplainer relies on random sampling to establish such connections, while ProxyExplainer generates them via learned graph decoders. These synthetic edges lead to distributional shifts in the mixup graphs and ultimately compromise the fidelity of the generated explanations.

In addition, estimating the mutual information $I(G^*, Y)$ between the explanatory subgraph $G^*$ and the label $Y$ remains a challenging prob-

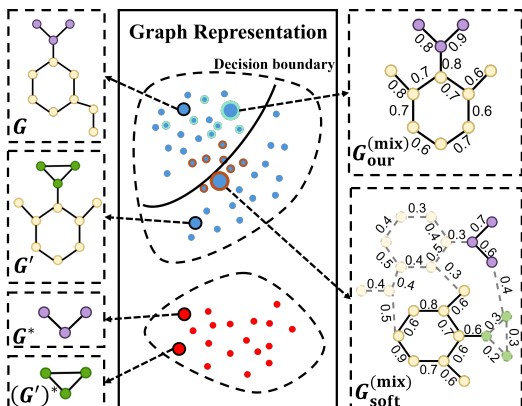

Figure 1: Intuitive illustration of the OOD problem and mixup strategies. The left part shows two original graphs $G$ and $G'$ with their corresponding explanatory subgraphs $G^*$ and $(G')^*$, while the right part presents mixup graphs generated by our structural mixup and the prior soft-mask-based mixup, denoted as $G_{\text{our}}^{(\text{mix})}$ and $G_{\text{soft}}^{(\text{mix})}$. The middle part illustrates the distribution in the latent space, where the blue and red regions represent the original graphs and explanatory subgraphs. $G^*$ and $(G')^*$ deviate from the original graph distribution, whereas both $G_{\text{our}}^{(\text{mix})}$ and $G_{\text{soft}}^{(\text{mix})}$ fall inside it, with the latter drifting away from the decision cluster of $G$ due to redundant information.

lem. Since the label $Y$ is discrete in classification tasks and continuous in regression tasks, existing methods approximate $I(G^*, Y)$ using task-dependent approaches. For classification tasks, cross-entropy loss is typically employed (Xin et al.), while for regression tasks, InfoNCE loss is used as an approximation (Zhang et al., 2024). These task-dependent approximation strategies hinder the integration of existing works within a unified GIB framework, resulting in a lack of generalization across diverse tasks.

To address these challenges, we propose **TAME**, a **T**ask-**A**gnostic structural **M**ixup **E**xplanation framework for GNN explainability. TAME introduces a theoretically grounded objective based on GIB, reformulated through contrastive learning (Oord et al., 2018; Zhang et al., 2024), which captures informative representations aligned with the graph structure rather than simply fitting label values, allowing for a unified estimation of $I(G^*, Y)$ that seamlessly supports both classification and regression tasks. To further address the distribution shift caused by optimizing $I(G, G^*)$, we design a novel structural mixup strategy built upon graph pooling, which fuses explanatory and non-explanatory subgraphs via structural replacement, generating in-distribution, structurally faithful mixup graphs that preserve natural connectivity and improve explanation fidelity. Our contributions are summarized as follows:

• We are the first to propose a contrastive learning-based, task-agnostic GIB objective for GNN explanation, enabling a unified and theoretically grounded approach that generalizes across both classification and regression.

• We design a novel mixup strategy that structurally replaces explanatory subgraphs rather than soft mixup. This addresses distribution shift and produces mixup explanations that are structurally faithful to the original graphs, as confirmed by qualitative visualizations.

• Comprehensive experiments on both synthetic and real-world benchmarks demonstrate that TAME consistently outperforms existing methods across diverse tasks, achieving up to a 30.1% improvement in AUC, while maintaining explanation consistency and generalization.

## 2 Preliminary

### 2.1 Notations and Problem Formulation

We give a graph as $G = (\mathcal{V}, \mathcal{E}, \boldsymbol{X}, \boldsymbol{A})^1$ from a graph dataset $\mathcal{G}$, where $\mathcal{V} = \{v_1, v_2, \ldots, v_n\}$ denotes the node set with $n$ denoting the number of nodes, and $\mathcal{E} \in \mathcal{V} \times \mathcal{V}$ denotes the edge set. The graph feature matrix is denoted by $\boldsymbol{X} \in \mathbb{R}^{n \times d}$, where $d$ is the feature dimension and the $i$-th row $x_i \in \mathbb{R}^{i \times d}$ corresponds to node $v_i$. The adjacency matrix $\boldsymbol{A} \in \{0, 1\}^{n \times n}$ of $G$ determines the edge set $\mathcal{E}$ such that $A_{ij} = 1$ if there is an edge$(i, j) \in \mathcal{E}$ and $A_{ij} = 0$ otherwise. In this paper, we focus on the graph-level classification and regression tasks, where node-level tasks can be converted to graph-level problems (Ying et al., 2019; Luo et al., 2020). Notably, each graph $G_i$ has a label $Y_i \in \mathcal{Y}$, where $i \in \{1, \ldots, \mathcal{N}\}$ and $\mathcal{N}$ is the number of graphs in the dataset, with a pretrained GNN model $f$ to make prediction. The label set $\mathcal{Y}$ can be continuous in regression task and discrete in classification task. We use the node embeddings matrix $\boldsymbol{H}$ as the input for our explainer network $f_\psi(\cdot)$, which is extracted prior to the readout operation of the GNN $f(\cdot)$. In addition, we define $f_{\text{enc}}(\boldsymbol{X}, \boldsymbol{A})$ as the component of $f$ that generates the graph representation $\boldsymbol{h}_G$.

**Problem 1** (Post-hoc Instance-level GNN Explanatory Method). Given a pretrained GNN model $f$ and an arbitrary graph $G \in \mathcal{G}$, the objective of *post-hoc instance-level GNN explanatory method* is to find a subgraph $G^* \in G$ that can explain the prediction of model $f$ on $G$.

## 3 Methods

In this section, we first introduce a task-agnostic GIB objective based on InfoNCE, which unifies graph classification and regression tasks within a single explanatory framework. Next, we present a hard-perturbation explanation pipeline that leverages graph pooling to extract explanatory subgraphs and employs a structural mixup strategy to generate in-distribution mixup graphs, effectively addressing the distribution shift induced by GIB optimization. To further enhance robustness, we incorporate a similarity-guided contrastive learning mechanism, built upon structural mixup and aligned with the proposed GIB objective, to facilitate self-supervised explanation learning. An overview of the proposed framework is illustrated in Figure 2.

### 3.1 Task-Agnostic Graph Information Bottleneck

As introduced in the Introduction, prior works typically adopt distinct strategies to approximate the mutual information term $I(G^*; Y)$ in Equation 1. Since directly computing $I(G^*; Y)$ is generally intractable, a common workaround is to approximate it using the prediction label of the explanatory subgraph, denoted as $Y^*$. In classification tasks, $I(G^*; Y)$ is commonly approximated using cross-entropy $\text{CE}(Y^*, Y)$ (Ying et al., 2019; Zhang et al., 2023), while in regression tasks, $\text{InfoNCE}(Y^*, Y)$ is adopted (Zhang et al., 2024). Since the label variable $Y$ exhibits discrete and continuous characteristics in classification and regression tasks, existing explanatory methods are inherently task-specific and lack a unified framework.

**Optimizing the Lower Bound of** $I(G^*; Y)$. To overcome the limitations of task-dependent estimation, we aim to directly approximate the mutual information $I(G^*; Y)$ through a generalized representation-based approach. To this end, we adopt a lower bound to approximate $I(G^*; Y)$, enabling tractable and unified estimation across different tasks.

Specifically, we apply the Donsker–Varadhan (DV) variational representation of mutual information (Poole et al., 2019), which introduces a critic function to assess the dependence between $G^*$ and $Y$. This yields a surrogate lower bound $\text{DV}(G^*; Y)$ that approximates $I(G^*; Y)$ in Equation 1, leading to the following reformulation of the GIB objective:

$$\arg\min_{G^*} I(G, G^*) - \alpha \, \text{DV}(G^*; Y). \tag{2}$$

The validity of this approximation is guaranteed by the following property of $\text{DV}(G^*; Y)$:

**Property 1.** $\text{DV}(G^*; Y)$ is a lower bound of $I(G^*; Y)$.

---

$^1$To simplify notation, we denote graph as $G = (\boldsymbol{X}, \boldsymbol{A})$ in the remainder of this paper.

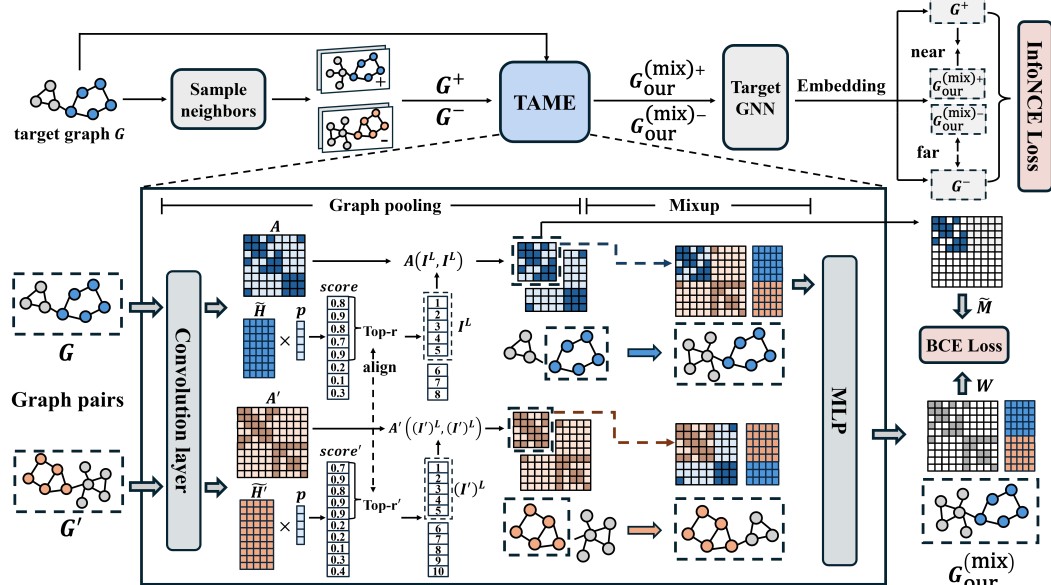

Figure 2: The overview of TAME framework. TAME takes a target graph $G$ and a randomly sampled neighbor $G'$ as inputs, where $G'$ is either positive $G^+$ or negative $G^-$. Graph pooling selects the top-$r$ nodes based on similarity scores to crop the adjacency matrix and obtain the explanatory subgraph. Structural mixup then exchanges explanatory structures between $G$ and $G'$ to generate in-distribution mixup graph $G_{\text{our}}^{(\text{mix})}$, followed by an MLP that produces the explanatory mask $\boldsymbol{W}$. The InfoNCE loss minimizes the representation distance between $G_{\text{our}}^{(\text{mix})+}$ and $G^+$ and maximizes the distance between $G_{\text{our}}^{(\text{mix})-}$ and $G^-$, whereas the BCE loss constrains $\boldsymbol{W}$ to remain faithful to the structure of the explanatory subgraph. The explainer is optimized jointly with these two objectives.

For any critic function $T : \mathcal{G} \times \mathcal{Y} \to \mathbb{R}$, the mutual information $I(G^*; Y)$ satisfies the following variational lower bound:

$$\text{DV}(G^*; Y) := \mathbb{E}_{p(G^*, Y)}\big[T(G^*, Y)\big] - \log \mathbb{E}_{p(G^*)p(Y)}\big[e^{T(G^*, Y)}\big]. \tag{3}$$

This variational form yields the lower bound $I(G^*; Y) \geq \text{DV}(G^*; Y)$ and enables learning-based estimation of $I(G^*; Y)$ by designing an appropriate critic function $T$. The detailed proof is provided in Appendix F.

**Estimating** $\text{DV}(G^*; Y)$ **with InfoNCE.** The challenge now becomes the definition of a generalizable critic function $T(G^*, Y)$ for estimating $\text{DV}(G^*; Y)$. Inspired by recent advances in contrastive representation learning (Oord et al., 2018; Chen et al., 2020a), we instantiate the critic function $T(G^*, Y)$ in a representation-based manner, enabling a tractable estimation of $\text{DV}(G^*; Y)$. Specifically, the critic function is defined by measuring the similarity between the embeddings of the explanatory subgraph $G^*$ and the graph $G_Y$ paired with the label $Y$:

$$T(G^*, Y) := \text{sim}(\boldsymbol{h}_{G^*}, \boldsymbol{h}_Y) + \log(N-1), \tag{4}$$

where $\boldsymbol{h}_{G^*} = f_{\text{enc}}(G^*)$ and $\boldsymbol{h}_Y = f_{\text{enc}}(G_Y)$ are graph representations obtained from a pretrained encoder $f_{\text{enc}}(\cdot)$, $\text{sim}(\cdot, \cdot)$ denotes the dot-product similarity between $\ell_2$-normalized vectors, and $N$ is the number of sampled graph labels, including one positive and $N-1$ negatives. Under this instantiation, the DV bound in Equation 3 admits the following lower bound, which we derive formally in Appendix G.

**Property 2.** InfoNCE is a lower bound of $\text{DV}(G^*; Y)$.

$$\text{DV}(G^*; Y) \geq \log(N-1) + \mathbb{E}_{\mathcal{P}}[\ell(G^*, Y)], \tag{5}$$

where $\ell(G^*, Y)$ is defined as the InfoNCE:

$$\ell(G^*, Y) := \log \frac{\exp\big(\text{sim}(\boldsymbol{h}_{g^*}, \boldsymbol{h}_{y^+})\big)}{\sum_{j=0}^{N-1} \exp\big(\text{sim}(\boldsymbol{h}_{g^*}, \boldsymbol{h}_{y_j})\big)}, \tag{6}$$

and $\mathcal{P} := p(g^*, y^+) \prod_{j=1}^{N-1} p(y_j^-)$ denotes the joint sampling distribution comprising a sampled explanatory graph $g^*$, its corresponding positive label sample $y^+$, and $N-1$ independent negative label samples $\{y_j^-\}_{j=1}^{N-1}$ drawn from the marginal label distribution. This property enables contrastive optimization of the GIB objective in a task-agnostic manner. Substituting $\text{DV}(G^*; Y)$ in Equation 2, we obtain the final training objective:

$$\arg\min_{G^*} I(G, G^*) - \alpha \, \mathbb{E}_{\mathcal{P}}[\ell(G^*, Y)]. \tag{7}$$

## 3.2 STRUCTURAL MIXUP WITH GRAPH POOLING

In the previous section, we employ InfoNCE to approximate the second term $I(G^*, Y)$ in GIB across different tasks. Building upon this, a structural mixup strategy grounded in graph pooling is proposed to address distribution shift and eliminate the introduction of redundant structural information. Specifically, the method first employs structural pooling to apply hard perturbations to the original graph, then mixes them with sampled neighbors and finally generates a soft mask for explanation. Each stage is detailed as follows.

**Extracting the Explanatory Subgraph $G^*$.** The graph pooling process can be implemented as multiple pooling and Graph Convolution layers (Kipf & Welling, 2016), as shown below:

$$\tilde{\boldsymbol{H}}^l = \text{Conv}^l(\boldsymbol{H}^{l-1}, \boldsymbol{A}^{l-1}), \quad \boldsymbol{A}^l, \boldsymbol{H}^l = \text{pooling}(\tilde{\boldsymbol{H}}^l, \boldsymbol{A}^{l-1}, r_l), \tag{8}$$

where $l$ denotes the $l$-th layer, $\text{Conv}^l$ is a single Graph Convolution layer that updates node embeddings, the pooling layer preserves the Top-$r_l$ most essential nodes with $r_l$ denoting the ratio of nodes preserved, and $\boldsymbol{H}^l$ and $\boldsymbol{A}^l$ represent the node embeddings and adjacency matrix. More specifically, in the $l$-th pooling layer, $\text{score} = \tilde{\boldsymbol{H}}^l \cdot \boldsymbol{p}_l / \|\boldsymbol{p}_l\|$ measures the directional similarity between node embeddings $\tilde{\boldsymbol{H}}^l$ and a learnable projection vector $p_l$. The $\lfloor r_l \cdot n_{l-1} \rfloor$ nodes with the highest scores are preserved, where $n_{l-1}$ denotes the number of nodes in the subgraph at the $(l-1)$-th layer. The preserved node indices are denoted as $I^l$. After $L$ pooling layers, the graph pooling process progressively crops the original graph and yields the final preserved node indices $I^L = \{i_1, ..., i_m\} \subseteq \{1, ..., n\}$, with $m$ denoting the number of preserved node indices. Ultimately, the explanatory subgraph is represented as $G^* = (\boldsymbol{H}^*, \boldsymbol{A}^*)$, where $\boldsymbol{H}^* = \boldsymbol{H}(I^L, :)$ and $\boldsymbol{A}^* = \boldsymbol{A}(I^L, I^L)$.

**Sampling Neighbors.** For an input $G$ to be explained, we randomly sample $N$ graphs and identify its positive neighbor $G^+$ and negative neighbors $\{G_i^-\}_{i=1}^{N-1}$ based on representational similarity. Specifically, $\boldsymbol{h}_{G^+}$ and $\boldsymbol{h}_{G_i^-}$ denote the representations of $G^+$ and $G_i^-$ for each $i \in \{1, \ldots, N-1\}$. The similarity between $\boldsymbol{h}_{G^+}$ and $\boldsymbol{h}_G$ is higher than that between $\boldsymbol{h}_{G_i^-}$ and $\boldsymbol{h}_G$ for all $i$, i.e., $\text{sim}(\boldsymbol{h}_G, \boldsymbol{h}_{G^+}) > \text{sim}(\boldsymbol{h}_G, \boldsymbol{h}_{G_i^-})$. Then, we mix the input graph $G$ with sampled neighbors $G^+$ and $\{G_i^-\}_{i=1}^{N-1}$ to construct the mixup graphs.

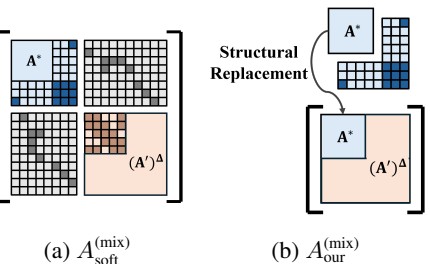

(a) $A_{\text{soft}}^{(\text{mix})}$      (b) $A_{\text{our}}^{(\text{mix})}$

Figure 3: Comparison of adjacency matrices from (a) the soft-mask-based strategy and (b) our structural mixup strategy. The solid-colored regions represent the structural information that should ultimately be preserved and $(\boldsymbol{A}')^\Delta$ denotes the label-irrelevant regions. $\boldsymbol{A}_{\text{soft}}^{(\text{mix})}$ retains redundant information, whereas our method generates a clean composition $\boldsymbol{A}_{\text{our}}^{(\text{mix})}$ by replacing $(\boldsymbol{A}')^*$ with $\boldsymbol{A}^*$ at the same node positions.

**Structural Mixup for Constructing $G_{\text{our}}^{(\text{mix})}$.** To generate mixup graphs that remain close to the original graph distribution, we replace the explanatory subgraph from a sampled neighbor $G' \in \{G^+, G^-\}$ with that from the input graph $G$. Let $\boldsymbol{A}'$ and $\boldsymbol{H}'$ denote the adjacency and feature matrices of $G'$, and let $(\boldsymbol{A}')^* \in \{0,1\}^{m \times m}$ and $\boldsymbol{A}^* \in \{0,1\}^{m \times m}$ denote the adjacency matrices of the explanatory subgraphs extracted from $G'$ and $G$, respectively, where $m$ is the number of preserved nodes. The $\boldsymbol{S}' \in \{0,1\}^{n' \times m}$ is a binary selection matrix that maps the preserved node indices $(I')^L$ of $G'$ to the full node space, constructed as $\boldsymbol{S}' = \boldsymbol{I}_{n'}[:, (I')^L]$. We define the mixup graph $G_{\text{our}}^{(\text{mix})} = (\boldsymbol{H}_{\text{our}}^{(\text{mix})}, \boldsymbol{A}_{\text{our}}^{(\text{mix})})$ as:

$$\boldsymbol{A}_{\text{our}}^{(\text{mix})} = \boldsymbol{A}' - \boldsymbol{S}'(\boldsymbol{A}')^* \boldsymbol{S}'^\top + \boldsymbol{S}' \boldsymbol{A}^* \boldsymbol{S}'^\top, \quad \boldsymbol{H}_{\text{our}}^{(\text{mix})} = \boldsymbol{H}' - \boldsymbol{S}'(\boldsymbol{H}')^* + \boldsymbol{S}' \boldsymbol{H}^*. \tag{9}$$

Intuitively, we replace the explanatory subgraph of $G'$ with that of $G$ at the same node positions, enabling structure-level replacement while preserving semantic locality. The number of preserved node indices $m$ is aligned across graphs to ensure compatibility of subgraph replacement. When the sampled neighbors are $G^+$ and $G^-$, these mixup graphs $G_{\text{our}}^{(\text{mix})+}$ and $G_{\text{our}}^{(\text{mix})-}$ are used in Equation 6 to replace $G^*$ in contrastive learning, thereby addressing the distribution shift introduced during explanatory subgraph extraction. As shown in Figure 3, the soft-mask-based mixup strategy fails to disentangle explanatory and label-irrelevant subgraphs and introduces synthetic edges, whereas our method generates a clean mixup adjacency matrix $\boldsymbol{A}_{\text{our}}^{(\text{mix})}$.

**Generating Soft Mask.** In practice, to improve the representational capacity of the explainer, an MLP layer is employed to generate edge weights for the mixup graph $G_{\text{our}}^{(\text{mix})}$. Specifically, the MLP takes the concatenated node embeddings $[\boldsymbol{h}_i; \boldsymbol{h}_j]$ from $\boldsymbol{H}_{\text{our}}^{(\text{mix})}$ and outputs $w_{ij} \in [0, 1]$ for each edge$(i, j)$ in $G_{\text{our}}^{(\text{mix})}$. The final mixup graph is $G_{\text{our}}^{(\text{mix})} = (\boldsymbol{H}_{\text{our}}^{(\text{mix})}, \boldsymbol{M}_{\text{our}}^{(\text{mix})} \odot \boldsymbol{A}_{\text{our}}^{(\text{mix})})$, where $\boldsymbol{M}_{\text{our}}^{(\text{mix})}$ can be expressed as $\boldsymbol{M}_{\text{our}}^{(\text{mix})} = (1 - \boldsymbol{W}) \odot (\boldsymbol{A}' - \boldsymbol{S}'(\boldsymbol{A}')^*\boldsymbol{S}'^{\top}) + \boldsymbol{W} \odot (\boldsymbol{S}'\boldsymbol{A}^*\boldsymbol{S}'^{\top})$, and $\boldsymbol{W}$ is the matrix of edge weights, whose elements $w_{ij}$ are generated by the MLP layer.

### 3.3 OVERALL LOSS FUNCTION

**InfoNCE Loss.** We implement the InfoNCE loss to train our explainer. $G^+$ and $\{G_i^-\}_{i=1}^{N-1}$ are positive and negative neighbors randomly sampled based on graph representational similarity. Via structural mixup, the graphs $G_{\text{our}}^{(\text{mix})+}$ and $\{G_{\text{our},i}^{(\text{mix})-}\}_{i=1}^{N-1}$ are constructed by combining the explanatory subgraph $G^*$ with the label-irrelevant parts of $G^+$ and $\{G_i^-\}_{i=1}^{N-1}$, respectively. The above Equation 6 is defined as follows:

$$\mathcal{L}_{\text{InfoNCE}} = -\log \frac{\exp(\text{sim}(\boldsymbol{h}_{G_{\text{our}}^{(\text{mix})+}}, \boldsymbol{h}_{G^+}))}{\exp(\text{sim}(\boldsymbol{h}_{G_{\text{our}}^{(\text{mix})+}}, \boldsymbol{h}_{G^+})) + \sum_{i=1}^{N-1} \exp(\text{sim}(\boldsymbol{h}_{G_{\text{our},i}^{(\text{mix})-}}, \boldsymbol{h}_{G_i^-}))}, \quad (10)$$

where $\text{sim}(\cdot, \cdot)$ denotes the cosine similarity between two representations.

**Binary Cross-Entropy (BCE) Loss.** To ensure that the generated edge weights are faithful to the explanatory and label-irrelevant subgraphs obtained from the graph pooling process, the BCE loss is defined as follows:

$$\mathcal{L}_{\text{BCE}} = -\sum_{(i,j) \in \mathcal{E}^{(\text{mix})}} \left[ \tilde{M}_{ij} \log w_{ij} + (1 - \tilde{M}_{ij}) \log(1 - w_{ij}) \right], \quad (11)$$

where $\tilde{\boldsymbol{M}} = \boldsymbol{S}'\boldsymbol{A}^*\boldsymbol{S}'^{\top}$ is a binary mask reflecting the structure of the explanatory subgraph and $w_{ij}$ are the elements of the edge weights matrix $\boldsymbol{W}$.

**Overall Loss Function.** The final overall loss is formulated as follows:

$$\mathcal{L} = \mathcal{L}_{\text{InfoNCE}}(G, G^+, \{G_i^-\}_{i=1}^{N-1}) + \beta \mathcal{L}_{\text{BCE}}(\tilde{\boldsymbol{M}}, \boldsymbol{W}), \quad (12)$$

where $\beta$ is a hyperparameter balancing the contribution of two parts, $\tilde{\boldsymbol{M}}$ is a binary mask derived from the adjacency matrix of the explanatory subgraph, and $\boldsymbol{W}$ is the edge weights matrix. $G^+$ and $\{G_i^-\}_{i=1}^{N-1}$ denote the positive and negative sampled neighbors, respectively. The training algorithm and computational complexity analysis are provided in Appendix D.

## 4 EXPERIMENTS

To comprehensively evaluate the effectiveness of TAME, we conduct experiments on synthetic and real-world datasets covering both classification and regression tasks. First, the basic experimental setup is presented in Section 4.1. Then, we evaluate the performance of TAME against baselines in identifying explanatory subgraphs in Section 4.2. Furthermore, the effectiveness of TAME in addressing the distribution shift problem is demonstrated in Section 4.3. Lastly, we present case studies in Section 4.5 to examine whether TAME accurately extracts explanatory subgraphs and generates natural in-distribution mixup graphs compared with baseline methods. Additional experiments, including hyperparameter sensitivity analysis and ablation studies, can be found in Appendix H.

Table 1: AUC-ROC edge-level explanation accuracy on seven graph classification datasets. Higher scores indicate better performance. SingleMotif datasets contain a single explanatory structure, whereas MultipleMotif datasets include multiple structures.

| Method | SingleMotif | | | MultipleMotif | | | |
| --- | --- | --- | --- | --- | --- | --- | --- |
| | BA-2motifs | BA-HouseGrid | SPMotif | BA-HouseAndGrid | Alkane-Carbonyl | Fluorid-Carbonyl | Benzene |
| Grad | $0.752 \pm 0.009$ | $0.609 \pm 0.005$ | $0.644 \pm 0.007$ | $0.419 \pm 0.010$ | $0.526 \pm 0.009$ | $0.687 \pm 0.006$ | $0.506 \pm 0.005$ |
| MetaGNN | $0.665 \pm 0.197$ | $0.840 \pm 0.096$ | $0.631 \pm 0.102$ | $0.806 \pm 0.069$ | $0.762 \pm 0.060$ | $0.667 \pm 0.041$ | $0.658 \pm 0.175$ |
| GNNExplainer | $0.512 \pm 0.004$ | $0.503 \pm 0.005$ | $0.510 \pm 0.003$ | $0.507 \pm 0.005$ | $0.512 \pm 0.012$ | $0.520 \pm 0.006$ | $0.497 \pm 0.004$ |
| PGExplainer | $0.677 \pm 0.069$ | $0.611 \pm 0.221$ | $0.607 \pm 0.023$ | $0.731 \pm 0.199$ | $0.619 \pm 0.373$ | $0.635 \pm 0.079$ | $0.750 \pm 0.182$ |
| TAGExplainer | $0.676 \pm 0.148$ | $0.848 \pm 0.039$ | $0.531 \pm 0.026$ | $0.647 \pm 0.072$ | $0.842 \pm 0.059$ | $0.752 \pm 0.046$ | $0.760 \pm 0.067$ |
| MatchExplainer | $0.802 \pm 0.001$ | $0.757 \pm 0.001$ | $0.499 \pm 0.000$ | $0.773 \pm 0.001$ | $0.603 \pm 0.002$ | $0.779 \pm 0.000$ | $0.512 \pm 0.001$ |
| MixupExplainer | $0.878 \pm 0.107$ | $0.811 \pm 0.078$ | $0.631 \pm 0.082$ | $0.804 \pm 0.190$ | $0.791 \pm 0.148$ | $0.686 \pm 0.049$ | $0.796 \pm 0.148$ |
| ProxyExplainer | $0.896 \pm 0.029$ | $0.745 \pm 0.327$ | $0.607 \pm 0.079$ | $0.704 \pm 0.191$ | $0.859 \pm 0.097$ | $0.729 \pm 0.129$ | $0.809 \pm 0.129$ |
| Our | $\mathbf{0.971 \pm 0.021}$ | $\mathbf{0.965 \pm 0.026}$ | $\mathbf{0.748 \pm 0.123}$ | $\mathbf{0.979 \pm 0.004}$ | $\mathbf{0.944 \pm 0.020}$ | $\mathbf{0.807 \pm 0.011}$ | $\mathbf{0.861 \pm 0.004}$ |

Table 2: Explanation accuracy measured by edge-level AUC-ROC on six graph regression datasets.

| Method | SingleMotif | | MultipleMotif | | | |
| --- | --- | --- | --- | --- | --- | --- |
| | BA-Motif-Volume | House-Grid-Volume | BA-Motif-Counting | Triangles | Crippen | House-OrGrid-Volume |
| Grad | $0.448 \pm 0.000$ | $0.544 \pm 0.000$ | $0.498 \pm 0.000$ | $0.587 \pm 0.000$ | $0.540 \pm 0.000$ | $0.477 \pm 0.000$ |
| ATT | $0.512 \pm 0.002$ | $0.499 \pm 0.002$ | $0.512 \pm 0.003$ | $0.512 \pm 0.003$ | $0.501 \pm 0.003$ | $0.521 \pm 0.003$ |
| TAGExplainer | $0.548 \pm 0.262$ | $0.856 \pm 0.105$ | $0.763 \pm 0.354$ | $0.610 \pm 0.163$ | $0.486 \pm 0.004$ | $0.698 \pm 0.167$ |
| MixupExplainer | $0.741 \pm 0.205$ | $0.854 \pm 0.082$ | $0.613 \pm 0.370$ | $0.561 \pm 0.139$ | $0.530 \pm 0.016$ | $0.721 \pm 0.128$ |
| RegExplainer | $0.766 \pm 0.139$ | $0.858 \pm 0.062$ | $\mathbf{0.946 \pm 0.095}$ | $0.560 \pm 0.132$ | $0.497 \pm 0.004$ | $0.741 \pm 0.107$ |
| Our | $\mathbf{0.997 \pm 0.001}$ | $\mathbf{0.966 \pm 0.003}$ | $\mathbf{0.946 \pm 0.004}$ | $\mathbf{0.648 \pm 0.020}$ | $\mathbf{0.565 \pm 0.038}$ | $\mathbf{0.941 \pm 0.012}$ |

## 4.1 EXPERIMENT SETTINGS

We evaluate TAME on thirteen datasets with ground-truth explanations, spanning graph classification and regression tasks across both synthetic and real-world molecular data. The synthetic datasets include BA-2Motifs (Luo et al., 2020), BA-HouseGrid (Amara et al., 2023), BA-HouseAndGrid (Bui et al., 2024), SPMotif (Wu et al., 2022), BA-Motif-Volume, BA-Motif-Counting (Zhang et al., 2024), House-Grid-Volume, House-OrGrid-Volume, and Triangles (Chen et al., 2020b), while the real-world molecular datasets include Alkane-Carbonyl, Fluoride-Carbonyl, Benzene (Sanchez-Lengeling et al., 2020), and Crippen (Delaney, 2004). To assess effectiveness, we benchmark TAME against seven representative post-hoc methods, including GNNExplainer (Ying et al., 2019), PGExplainer (Luo et al., 2020), TAGExplainer (Xie et al., 2022), MetaGNN (Spinelli et al., 2022), MatchExplainer (Wu et al., 2023), MixupExplainer (Zhang et al., 2023), ProxyExplainer (Chen et al., 2024), and RegExplainer (Zhang et al., 2024), as well as the gradient-based GRAD (Ying et al., 2019) and the attention-based ATT (Veličković et al., 2017).

Following prior works (Ying et al., 2019; Zhang et al., 2023), each dataset is divided into training, validation, and test sets in a ratio of $0.8$, $0.1$, and $0.1$. A three-layer GCN is adopted as the backbone model, with an additional linear layer for regression tasks. Analyses with other backbone models are reported in Appendix H.4. Regarding hyperparameters, we apply grid search to determine the loss weight $\beta$ and set the top-$r$ ratios according to the ground-truth explanations. Evaluation is conducted using the AUC-ROC score for ground-truth subgraph identification, while distributional shifts are assessed by comparing graph representations using cosine similarity and Euclidean distance. Further experimental details are provided in Appendix E.

## 4.2 QUANTITATIVE EVALUATION

In this section, we compare TAME with baselines using the AUC-ROC score. For MixupExplainer, which was originally designed for graph classification tasks, we replace the Cross-Entropy loss with MSE loss when applying it to regression tasks. All experiments are repeated 10 times with different random seeds. Tables 1 and 2 report the average scores and standard deviations for classification and regression tasks, respectively. Overall, our method consistently outperforms all baselines, providing the most accurate explanations across datasets. On graph classification datasets, TAME achieves an average improvement of 0.10 (11.39%) over the best baseline, while on regression datasets it

Table 3: Cosine similarity and Euclidean distance are computed between the graph representations of the original graph and those of the ground-truth explanatory subgraph (GT), as well as the mixup graphs from MixupExplainer (MixupE), ProxyExplainer (ProxyE), RegExplainer (RegE), and our method (TAME). Higher values for cosine similarity and lower values for Euclidean distance denote greater distributional similarity. Bold values indicate better performance.

| | | Classification | | | | | Regression | | |
|---|---|---|---|---|---|---|---|---|---|
| Dataset | GT | MixupE | ProxyE | TAME | Dataset | GT | RegE | TAME |
| **Cosine Similarity ↑** | | | | | **Cosine Similarity ↑** | | | |
| Ba-HouseGrid | 0.688 | 0.781 | 0.802 | **0.868** | BA-Motif-Volume | 0.696 | 0.876 | **0.953** |
| Ba-HouseAndGrid | 0.613 | 0.966 | 0.970 | **0.971** | BA-Motif-Counting | 0.663 | 0.950 | **0.969** |
| Benzene | 0.821 | 0.907 | 0.565 | **0.941** | Crippen | 0.869 | 0.847 | **0.936** |
| Fluorid-Carbonyl | 0.918 | 0.895 | 0.723 | **0.937** | Triangles | 0.897 | 0.953 | **0.985** |
| **Euclidean Distance ↓** | | | | | **Euclidean Distance ↓** | | | |
| Ba-HouseGrid | 1.044 | 0.973 | 0.706 | **0.661** | BA-Motif-Volume | 1.138 | 0.775 | **0.458** |
| Ba-HouseAndGrid | 1.436 | 0.477 | 0.442 | **0.425** | BA-Motif-Counting | 1.196 | 0.485 | **0.400** |
| Benzene | 0.953 | 0.771 | 1.471 | **0.578** | Crippen | 0.653 | 0.794 | **0.431** |
| Fluorid-Carbonyl | 0.684 | 0.782 | 1.272 | **0.604** | Triangles | 0.389 | 0.247 | **0.143** |

improves by 0.10 (13.43%) on average, with the highest gain reaching 0.231 (30.16%). These results highlight the consistent effectiveness of TAME across different task settings.

Compared with representative methods, TAME consistently captures the invariant explanatory factors. For instance, MixupExplainer performs well on classification datasets but underperforms on regression datasets, as it lacks a general contrastive mechanism and relies solely on MSE loss, which is insufficient for extracting explanatory subgraphs in regression tasks with continuous decision boundaries. Although RegExplainer achieves strong performance on BA-Motif-Counting, it employs a soft mixup strategy based on matrix concatenation, which introduces redundant edges and fails to maintain stable performance across all datasets. ProxyExplainer, which adopts a generative approach, often introduces irrelevant edges on complex datasets, resulting in poor performance on MultipleMotif benchmarks such as BA-HouseAndGrid, as further confirmed by the visualizations in Appendix H.5.1. Similarly, while TAGExplainer aims for task-agnostic explanation, its reliance on gradient-based techniques for the downstream explainer, coupled with OOD issues in its self-supervised embedding learning, compromises the quality of the extracted explanatory subgraphs on all datasets. In contrast, TAME achieves stable and superior performance across all benchmarks, demonstrating robustness and adaptability. Appendix H.1 provides additional validation on node classification and link prediction tasks.

### 4.3 ALLEVIATING DISTRIBUTION SHIFTS

In this section, we quantitatively analyze the distance between the original graph $G$ and the ground truth explanation subgraph $G^*$ to demonstrate the presence of distribution shift in both classification and regression tasks, and to show that our proposed method effectively mitigates this issue.

Specifically, we use the output of the penultimate layer of the target model as the graph representation vector and distribution shift is evaluated by computing the average cosine similarity and Euclidean distance between original graph representations and their ground-truth or mixup counterparts. As shown in Table 3, "GT" denotes the average cosine similarity and Euclidean distance between the original graph and the ground-truth explanatory subgraph. Meanwhile, "MixupE", "ProxyE", and "TAME" denote the same metrics between the original graph and the mixup graphs generated by MixupExplainer, ProxyExplainer, and our method, respectively. The results reveal lower similarity and higher Euclidean distance between the representations of original graph and ground-truth explanatory subgraph, demonstrating the existence of distributional shift in both classification and regression tasks. Moreover, since existing soft-mask-based methods inherently introduce redundant and irrelevant edges, they fail to adequately approximate the original graph distribution in more complex real-world datasets such as Fluorid-Carbonyl and Crippen. In contrast, the mixup graphs generated by our method exhibit higher cosine similarity and lower Euclidean distance to the original graph representations, indicating that the proposed mixup strategy effec-

tively mitigates the distributional shift problem and enables the explainer to more precisely identify explanatory subgraphs, thereby enhancing overall explanation performance.

## 4.4 ABLATION STUDY

In this section, we conduct ablation studies to evaluate the contribution of each component in the TAME framework. Specifically, we design the following variants: 1) w/o Mix: Removes the structral mixup step after extracting the explanation, directly feeding the explanation subgraph into the objective function. 2) w/o CTL: Retains structral mixup strategy and BCE loss, but replaces the contrastive learning term with cross-entropy loss for classification and mean squared error (MSE) loss for regression. 3) w/o BCE: Removes only the binary mask behavior regularization term from the training loss. All variants are configured with the same hyperparameters as the original TAME, including the learning rate, number of training epochs, and loss weight $\beta$.

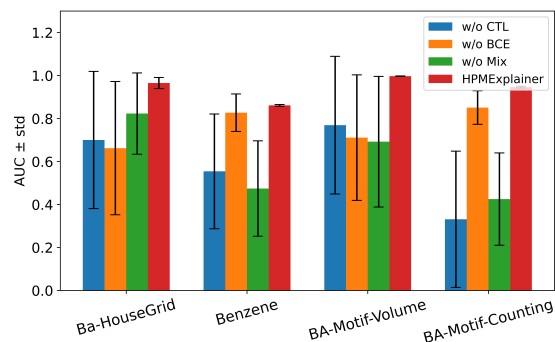

Figure 4: Ablation study of TAME across classification and regression tasks. We evaluate the AUC-ROC performance of the original TAME and its variants. Standard deviations are indicated by black error bars.

We conduct experiments on representative datasets covering both classification and regression tasks, with BA-HouseGrid and Benzene used for classification, and BA-motif-volume and BA-Motif-Counting used for regression. The results are presented in Figure 4. Experimental results show that TAME consistently outperforms all its variants across all datasets, indicating that each component contributes positively to the overall performance. Notably, the performance of the *w/o CTL* variant drops significantly on both classification and regression datasets, indicating that our proposed contrastive learning mechanism plays a critical role in both tasks.

## 4.5 CASE STUDIES

To verify whether our method can effectively capture explanatory subgraphs, we conduct a case study on the real-world classification dataset Benzene and the synthetic regression dataset BA-Motif-Volume. The visualization results are presented in Figure 6. Explanatory subgraphs are highlighted with bold red edges for Benzene and bold black for BA-Motif-Volume. For TAME, the nodes identified through graph pooling are additionally marked with orange overlays in Benzene and red overlays in BA-Motif-Volume. The extracted explanation subgraph is obtained by ranking all edges according to their weight scores and retaining the top-$K$ edges, where $K$ is set to the number of edges in the ground-truth explanation. The visualization results demonstrate that TAME consistently extracts the most ac-

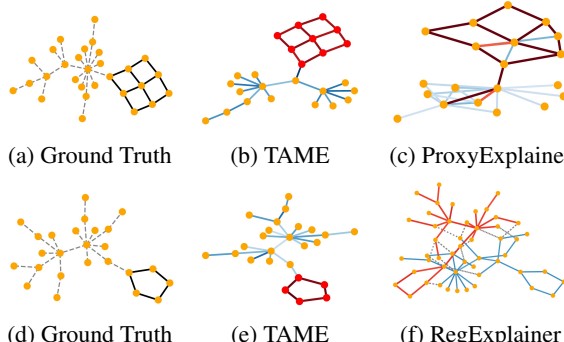

Figure 5: Visualization of the ground truth and mixup graphs generated by different methods. Subfigures (a)–(c) show results on the BA-HouseGrid classification dataset, while Subfigures (d)–(f) correspond to the BA-Motif-Volume regression dataset. Edge weights are indicated by color intensity.

curate explanations across both classification and regression tasks. Moreover, the nodes identified through graph pooling perfectly align with the ground-truth nodes, demonstrating that the pooling process is effectively optimized. Additional results can be found in Appendix H.5.1.

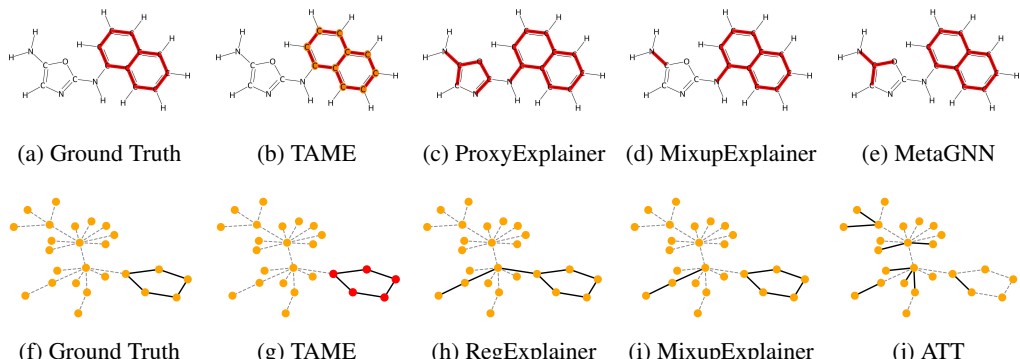

Figure 6: Visualization of explanation results obtained by different methods. Subfigures (a)–(e) show results on the Benzene classification dataset, while Subfigures (f)–(j) correspond to the BA-Motif-Volume regression dataset.

To further validate the effectiveness of the proposed structural mixup strategy, we conduct another case study comparing the quality of the mixup graphs generated by TAME and baseline methods, as shown in Figure 5. Specifically, ProxyExplainer is evaluated on BA-HouseGrid for the classification task, and RegExplainer on BA-Motif-Volume for the regression task, with additional examples provided in Appendix H.5.2. The visualizations are obtained from intermediate results of the final training epoch under the best-performing hyperparameters.

In the mixup graphs, edges in the explanatory subgraph are highlighted in red and those in the label-irrelevant subgraph in blue, with color intensity indicating weight values in $[0, 1]$, while random connections generated by RegExplainer are shown as gray dashed lines. Our structural mixup strategy naturally integrates explanatory and label-irrelevant subgraphs, producing concise mixup graphs that preserve the original structural distribution. In contrast, ProxyExplainer employs a generative approach that may introduce redundant edges and distort the explanatory structure, while RegExplainer performs a soft mixup via matrix concatenation, resulting in randomly sampled uninterpretable edges and generating nearly identical weights for both explanatory and label-irrelevant parts, thereby deviating from the original structure. Overall, the visualization results across both classification and regression tasks confirm the superiority of our strategy in constructing in-distribution mixup graphs, enabling TAME to generate more robust and precise explanations.

## 5 CONCLUSION

In this work, we systematically investigate the OOD problem in post-hoc instance-level GNN explanation frameworks grounded in the GIB principle, and uncover the task dependency inherent in the estimation of the GIB objective, which has been largely overlooked in prior studies. To address these limitations, we propose TAME, which leverages InfoNCE to reformulate a task-agnostic GIB objective that consistently supports both classification and regression tasks, establishing a theoretical foundation for future research on GNN explainability. Moreover, we introduce a novel structural mixup strategy built upon graph pooling that generates naturally connected in-distribution mixup graphs, innovatively addressing the OOD problem. Extensive experiments on both synthetic and real-world benchmarks demonstrate that TAME consistently outperforms existing baselines, effectively extracting faithful explanations across diverse tasks. Note that TAME is primarily designed for graph- and node-level explanations with well-defined subgraph dependencies, whereas edge-level tasks may require further adaptation, which we leave for future exploration.

## 6 ETHICS STATEMENT

This work focuses on improving the explainability of GNNs through a unified task-agnostic framework, with the primary goal of contributing to the academic community by enhancing GNN explanations. There are many potential societal consequences of our work, we believe that our work raises no direct or immediate ethical concerns that require particular emphasis in this section, and may ultimately facilitate safer and more transparent deployment of GNNs in practice.

# 7 REPRODUCIBILITY STATEMENT

Our work is fully reproducible. Specifically, the detailed proofs of the properties in the main paper are provided in Appendix F and Appendix G. The algorithm process and computational complexity analysis are presented in Appendix D. Appendix E contains comprehensive experimental details, including the experimental environment, implementation details, dataset description, and a description of the baseline methods for comparison. Additionally, extensive experiments are provided in Appendix H, to enable a thorough understanding of our approach. Finally, to ensure reproducibility, our implementation can be accessed through the following anonymous repository: https://anonymous.4open.science/r/TAME-main-2FB7. To reproduce the AUC-ROC performance experiments, one can directly run experiment_replication.py in the main directory. The full implementation of our explainer is available in ExplanationEvaluation/explainers/TAME.py.

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

## A  THE USE OF LARGE LANGUAGE MODELS (LLMS)

In preparing this paper, Large Language Models (LLMs) were used solely as general-purpose writing assistants. Specifically, we employed LLMs to polish the text by carefully correcting grammar, spelling, and occasionally translating our original sentences into fluent academic English when necessary. Importantly, the LLMs were not used to generate new research ideas, formulate novel claims, or alter the meaning of our original sentences. All outputs generated by LLMs were carefully reviewed and verified by the authors for consistency with our intended meaning and to avoid any issues of plagiarism or scientific misconduct.

## B  RELATED WORK

**GNN Explainability and Out-Of-Distribution.** Current GNN explanatory methods (Shin et al., 2024; Wang & Shen, 2022; Yu & Gao, 2024; Zhang et al., 2022; Wu et al., 2023; Li et al., 2023) aim to enhance model decision transparency through model-level or instance-level methods (Yuan et al., 2022). Model-level methods (Yuan et al., 2020) enhance GNN explainability by generating input-independent explanations, while instance-level methods (Yuan et al., 2021) identify input features most relevant to the model prediction. In this work, we focus on post-hoc instance-level methods. However, such methods face growing OOD challenges (Zheng et al., 2023; Fang et al., 2023). While some works (Fang et al., 2024; Faber et al., 2020) have already made attempts, but they require that the model be retrained. Separately, traditional mixup-based data augmentation methods (Zhang et al., 2023; Chen et al., 2024; Zhang et al., 2024) also alleviate the OOD problem, while still struggle to ensure the quality of mixup graphs for approximating the original distribution.

**Graph Contrastive Learning.** Graph Contrastive Learning (GCL) has become a dominant paradigm for graph self-supervised learning, aiming to learn discriminative label-independent representations by maximizing mutual information across different views. For example, GraphCL (You et al., 2020) leverages various data augmentation strategies to enhance the robustness and transferability of unsupervised representations. AutoGCL (Yin et al., 2022) employs learnable view generators, guided by auto-augmentation, to produce semantically consistent yet diverse graph views. SimGRACE (Xia et al., 2022) generates contrastive views via perturbed encoders for representation alignment. Among current post-hoc explanatory methods, RegExplainer (Zhang et al., 2024) is a pioneering work that introduces contrastive learning in the graph regression task. However, a unified task-agnostic explanation framework remains unexplored.

**Graph Pooling.** Graph pooling methods typically capture global information through hierarchical compression to obtain generalizable and expressive representations. Some methods learn a score function from node features and select the top $r$ nodes to form a new subgraph (Gao & Ji, 2019). For example, SAGPooling (Lee et al., 2019) refines node scoring by integrating structural context via GNNs. GSAPool (Zhang et al., 2021) enhances pooling process by dynamically fusing node features and topology information. DMIPool (Zhao et al., 2023) further captures multi-level dependencies from dual-view representations to improve robustness. MLAP (Itoh et al., 2022) preserves cross-layer structural information through layer-wise pooling, enhancing node distinguishability. Our method introduces pooling strategies to apply hard perturbations to graph structure, extracting more distinguishable explanatory and label-irrelevant subgraphs, ultimately enabling the mixup graphs to more faithfully approximate the original distribution.

## C  NOTATIONS

In Table 4, we summarized the important notations we used and their descriptions in this paper.

Table 4: Important symbols and notations.

| Symbols | Descriptions |
|---|---|
| $G$ | Graph instance |
| $\boldsymbol{X}$ | Node feature matrix |
| $\boldsymbol{A}$ | Adjacency matrix |
| $Y$ | Label for $G$ |
| $\mathcal{G}$ | Graph dataset |
| $\mathcal{V}, \mathcal{E}$ | Node set and edge set |
| $\mathcal{Y}$ | Label set |
| $v_i$ | The $i$-th node |
| $x_i$ | The feature of node $v_i$ |
| $n$ | Number of nodes in $G$ |
| $\mathrm{edge}(i, j)$ | Edge from node $i$ to node $j$ |
| $i$ | The $i$-th node index |
| $j$ | The $j$-th node index |
| $d$ | Feature dimension |
| $G^*$ | Optimal explanatory subgraph |
| $Y^*$ | Label for $G^*$ |
| $\boldsymbol{H}^*, \boldsymbol{A}^*$ | Node embeddings and adjacency matrix of $G^*$ |
| $I(\cdot)$ | Mutual information |
| $\alpha$ | Hyper parameter for mutual information term in GIB |
| $\mathcal{N}$ | Number of graphs in $\mathcal{G}$ |
| $f(\cdot)$ | To-be-explained GNN model |
| $\boldsymbol{H}$ | Node embeddings matrix |
| $f_\psi(\cdot)$ | Explainer network |
| $f_{\mathrm{enc}}(\cdot)$ | Encoder component of $f$ that generates graph representations |
| $\boldsymbol{h}_G$ | Graph representation of $G$ |
| $\mathrm{DV}(\cdot)$ | Donsker–Varadhan variational representation |
| $T(\cdot)$ | Critic function |
| $G_Y$ | The graph paired with label $Y$ |
| $\mathrm{sim}(\cdot, \cdot)$ | Dot-product similarity between $\ell_2$-normalized vectors |
| $r_l$ | Node preservation ratio at layer $l$ |
| $I^L$ | Final preserved node indices after $L$ pooling layers |
| score | Node scores |
| $p_l$ | Learnable projection vector at layer $l$ |
| $N$ | Number of sampled neighbors |
| $G'$ | Sampled neighbor |
| $\boldsymbol{A}_{\mathrm{our}}^{(\mathrm{mix})}$ | Adjacency matrix obtained via structural mixup strategy |
| $\boldsymbol{H}_{\mathrm{our}}^{(\mathrm{mix})}$ | Node embeddings matrix obtained via structural mixup strategy |
| $G_{\mathrm{our}}^{(\mathrm{mix})}$ | Graph obtained via structural mixup strategy |
| $S'$ | Binary node preserved matrix from $G'$ |
| $\boldsymbol{W}$ | Edge weights matrix |
| $\boldsymbol{M}_{\mathrm{our}}^{(\mathrm{mix})}$ | Edge weights mask for mixup graph |
| $\mathcal{L}_{\mathrm{InfoNCE}}$ | InfoNCE loss |
| $\mathcal{L}_{\mathrm{BCE}}$ | Binary cross-entropy loss |
| $\beta$ | Hyper parameter balancing $\mathcal{L}_{\mathrm{InfoNCE}}$ and $\mathcal{L}_{\mathrm{BCE}}$ |

# D  ALGORITHMS

---

**Algorithm 1** Training Explainer

---

**Input:** A graph dataset $\mathcal{G}$, pretrained GNN model $f$, explainer network $f_\psi(\cdot)$.
**Output:** Trained explainer network $f_\psi(\cdot)$.
1: Initialize explainer network $f_\psi(\cdot)$.
2: **for** $e \in$ epochs **do**
3:      **for** $G \in \mathcal{G}$ **do**
4:          Randomly sample $N$ graphs $\{G_i\}_{i=0}^{N-1}$ from graph dataset $\mathcal{G}$
5:          $G^+, \{G_i^-\}_{i=1}^{N-1} \leftarrow$ Similarity-based Sampling$(G, \{G_i\}_{i=0}^{N-1})$
6:          $G_{\text{our}}^{(\text{mix})+} \leftarrow$ Structural Mixup$(G, G^+)$
7:          $\{G_{\text{our},i}^{(\text{mix})-}\}_{i=1}^{N-1} \leftarrow$ Structural Mixup$(G, \{G_i^-\}_{i=1}^{N-1})$
8:          Generate $\boldsymbol{M}_{(\text{our})}^{(\text{mix})+}$ and $\{\boldsymbol{M}_{\text{our},i}^{(\text{mix})-}\}_{i=1}^{N-1}$ with $G_{\text{our}}^{(\text{mix})+}$ and $\{G_{\text{our},i}^{(\text{mix})-}\}_{i=1}^{N-1}$
9:          Compute $\mathcal{L}_{\text{InfoNCE}}(G, G^+, \{G_i^-\}_{i=1}^{N-1})$ with Equation 10
10:        Compute $\mathcal{L}_{\text{BCE}}$ with Equation 11
11:        Compute overall loss $\mathcal{L}$ with Equation 12
12:      **end for**
13:      Update $f_\psi(\cdot)$ with back propagation
14: **end for**
15: **Return** Trained explainer network $f_\psi(\cdot)$

---

**Algorithm 2** Similarity-based Sampling

---

**Input:** Target graph $G$, pre-sampled graphs $\{G_i\}_{i=0}^{N-1}$.
**Output:** Positive graph $G^+$, negative graphs $\{G_i^-\}_{i=1}^{N-1}$.
1: $\mathcal{S} \leftarrow [\ ]$
2: **for** $i \leftarrow 0$ to $N-1$ **do**
3:      $s_i \leftarrow$ CosineSimilarity$(f_{\text{enc}}(G), f_{\text{enc}}(G_i))$
4:      Append $s_i$ to $\mathcal{S}$
5: **end for**
6: Sort $\{G_i\}_{i=0}^{N-1}$ by $\mathcal{S}$ descendingly
7: $G^+ \leftarrow G_0$
8: $\{G_i^-\}_{i=1}^{N-1} \leftarrow \{G_1, \ldots, G_{N-1}\}$
9: **Return** $G^+, \{G_i^-\}_{i=1}^{N-1}$

---

**Algorithm 3** Structural Mixup

---

**Input:** To-be-explained graph $G$, $G'$ sampled from graph dataset $\mathcal{G}$.
**Output:** Graph $G_{\text{our}}^{(\text{mix})}$.
1: Obtain $I^L$ via $L$-layer graph pooling following Equation 8
2: Obtain $(I')^L$ via $L$-layer graph pooling following Equation 8
3: Generate explanatory subgraphs $G^*$ and $(G')^*$ based on $I^L$ and $(I')^L$
4: Structural Mixup adjacency $\boldsymbol{A}_{\text{our}}^{(\text{mix})}$ and node embeddings $\boldsymbol{H}_{\text{our}}^{(\text{mix})}$ matrix with Equation 9
5: **Return** $G_{\text{our}}^{(\text{mix})} = (\boldsymbol{H}_{\text{our}}^{(\text{mix})}, \boldsymbol{A}_{\text{our}}^{(\text{mix})})$

---

To address the distribution shift issue in the GIB objective, a graph-pooling-based structural mixup strategy is employed to generate in-distribution, naturally connected mixup graphs $G_{\text{our}}^{(\text{mix})}$ that preserve the information of explanatory subgraphs $G^*$. Built upon this, contrastive learning is introduced as an optimization objective within GIB to capture underlying representations, thereby approximating the mutual information between $G^*$ and the label $Y$ without relying on task-specific supervision. Together, task-agnostic structural mixup strategy and GIB objective form a unified framework for explainability. For completeness, we present the pseudocode of our method below, which encompasses both explainer training and the structural mixup strategy.

In Algorithm 1, at each epoch, we first randomly sample $N$ neighbors $\{G_i\}_{i=0}^{N-1}$ from the graph dataset $\mathcal{G}$ for each graph $G$. Based on the similarity between their representations and that of $G$,

we identify the positive neighbor $G^+$ and the negative neighbors $\{G_i^-\}_{i=1}^{N-1}$. Then, the structural mixup strategy combines the explanatory subgraph $G^*$ with the label-irrelevant parts of $G^+$ and $\{G_i^-\}_{i=1}^{N-1}$, producing the positive and negative mixup graphs $G_{\text{our}}^{(\text{mix})+}$ and $\{G_{\text{our},i}^{(\text{mix})-}\}_{i=1}^{N-1}$. For each mixup graph, an MLP is employed to predict the edge weight of each edge, and the structural information obtained from graph pooling process is incorporated to generate the final masks $\boldsymbol{M}_{\text{our}}^{(\text{mix})+}$ and $\{\boldsymbol{M}_{\text{our},i}^{(\text{mix})-}\}_{i=1}^{N-1}$. The explainer is optimized by minimizing the InfoNCE loss Equation 10 and the BCE loss Equation 11, while the overall loss function Equation 12 sums the two objectives with a trade-off parameter. Finally, the explainer parameters $f_\psi(\cdot)$ are updated through backpropagation with the overall loss.

Specifically, Algorithm 2 details the similarity-based sampling process. For each sampled graph $G_i$, we compute the cosine similarity between the graph representations of $G$ and $G_i$ using the encoder $f_{\text{enc}}$. All sampled graphs are then sorted by similarity scores in descending order, with the most similar graph as $G^+$ and the remaining graphs as $\{G_i^-\}_{i=1}^{N-1}$. Algorithm 3 shows the structural mixup process. Given a to-be-explained graph $G$ and a sampled neighbor graph $G'$, where $G'$ can be instantiated as the positive neighbor $G^+$ or one of the negative neighbors $\{G_i^-\}_{i=1}^{N-1}$, we perform $L$-layer graph pooling to obtain the preserved node indices $I^L$ and $(I')^L$. Then, the explanatory subgraph $G^*$ is represented as $G^* = (\boldsymbol{H}^*, \boldsymbol{A}^*)$, where $\boldsymbol{H}^* = \boldsymbol{H}(I^L, :)$ and $\boldsymbol{A}^* = \boldsymbol{A}(I^L, I^L)$. Similarly, the explanatory subgraph of $G'$ is obtained in the same way based on $(I')^L$. The adjacency matrix $\boldsymbol{A}_{\text{our}}^{(\text{mix})}$ and node embeddings $\boldsymbol{H}_{\text{our}}^{(\text{mix})}$ of the mixup graph are obtained by replacing $(\boldsymbol{A}')^*$ with $\boldsymbol{A}^*$, as defined in Equation 9. Finally, the mixup graph $G_{\text{our}}^{(\text{mix})} = (\boldsymbol{H}_{\text{our}}^{(\text{mix})}, \boldsymbol{A}_{\text{our}}^{(\text{mix})})$ is returned.

**Computational Complexity Analysis.** Given graph $G$, the time complexity of $L$-layer graph pooling is $O(L \cdot (|\mathcal{E}|d + nd))$, where $d$ is the feature dimension, $\mathcal{E}$ denotes the edge set, and $n$ is the number of nodes in $G$. Then, structural replacement replaces the explanatory subgraph via $m$ node indices, with a complexity of $O(m^2)$, where $m$ is the number of final preserved nodes. Edge weights for the mixup graph are generated by an MLP and the time complexity is $O(|\mathcal{E}^{(\text{mix})}| \cdot d)$, where $\mathcal{E}^{(\text{mix})}$ denotes the edge set of the mixup graph. Since $m \ll n$, $|\mathcal{E}| \ll n^2$, and the sizes of $|\mathcal{E}|$ and $|\mathcal{E}^{(\text{mix})}|$ are comparable, the overall time complexity of our method is dominated by $O(L \cdot |\mathcal{E}|d)$.

# E FULL EXPERIMENTAL SETUP

Detailed experimental settings are provided in this section, including implementation details, datasets, and baseline methods. All experiments are conducted on a Linux workstation running Ubuntu 22.10 (kernel 5.19.0-46-generic) equipped with 8 NVIDIA GeForce RTX 4090 GPUs (24 GB each). The system uses NVIDIA driver version 535.86.05 and CUDA 12.2. All code is implemented in Python 3.9.21, using PyTorch 2.0.1+cu118, PyTorch Geometric 2.6.1, torch-scatter 2.1.2+pt20cu118, and torch-sparse 0.6.18+pt20cu118. The complete experimental details can be found in our anonymous repository: https://anonymous.4open.science/r/TAME-main-2FB7.

## E.1 IMPLEMENTATION DETAILS

Following the configuration in previous work (Ying et al., 2019; Zhang et al., 2023), we divide each dataset into training, validation, and test sets with a ratio of 0.8, 0.1, and 0.1, respectively. We choose GCN as the backbone model for graph-level and node-level tasks, given its stable performance across diverse datasets, with SEAL Zhang & Chen (2018) adopted for link prediction task following established practices. The detailed comparison of the prediction performance of GCN and other GNN architectures are provided in Appendix H.4. A three-layer GCN is used as the target model, and for graph regression tasks, an additional linear layer is appended. Additionally, a two-layer MLP serves as the explanation network. For the compared baselines, we strictly adhered to the original implementations without any architectural modifications. For link prediction, we use SEAL Zhang & Chen (2018) as the base model to be explained. We train the GCN model to a reasonable performance, as shown in the GCN column of Tables 8 and 9. For explanation, we follow the sample selection strategy used in GNNExplainer (Ying et al., 2019) and randomly select 200 graph instances per dataset to reduce computational cost. All explanation methods are optimized using the Adam optimizer with a weight decay of $5 \times 10^{-4}$ (Kingma, 2014). Regarding hyperparameters, we apply grid search to determine the loss weights $\beta$, and set the top-$r$ ratios according to the ground truth explanations. For baseline methods with overlapping hyperparameters, we adopt a

unified setting; otherwise, we retain their default configurations. We ensure consistent learning rates and training epochs across all explanation methods. We evaluate the performance of our proposed method against baseline approaches on ground-truth explanation tasks using the AUC-ROC score and robust fidelity. To quantitatively assess the effectiveness of our method in addressing distributional shifts, we measure the distances between graph representations using cosine similarity and Euclidean distance.

## E.2 DATASETS

We evaluate our method on a range of graph datasets, including both synthetic and real-world benchmarks.

**BA-Shapes (Ying et al., 2019).** This dataset is based on a BA graph with 80 "house" motifs, where node labels define a 4-class classification task and motif edges provide explanation ground truth.

**BA-Community (Ying et al., 2019).** This is an extension of BA-Shapes where two different motifs are attached to the base graph, and nodes across two motifs are assigned different labels, resulting in an 8-class classification task.

**Tree-Grid (Ying et al., 2019).** This node classification dataset is based on a single 8-layer balanced binary tree with 80 grid motifs. The task is to distinguish motif nodes from tree nodes, with the motif edges serving as the ground-truth explanations.

**BA-2Motifs (Luo et al., 2020).** The BA-2Motifs dataset comprises 1,000 synthetic graphs, each generated from a Barabási–Albert graph and extended with either a house or a five-cycle motif. The label of each graph is determined by its attached motif, forming a binary classification task in which the motif serves as the ground-truth explanation.

**BA-HouseGrid (Bui et al., 2024).** The BA-HouseGrid dataset employs the house motif and the $3 \times 3$ grid motif. These motifs are chosen to minimize structural overlap and ensure that models learn the full motif structure instead of relying on local substructures for prediction.

**SPMotif (Wu et al., 2022).** In the SPMotif dataset, each graph consists of a base structure (Tree, Ladder, or Wheel) and a motif (Cycle, House, or Crane). A parameter $b$ controls the degree of the spurious correlation between the base structure and the motif, with $b = \frac{1}{3}$ indicating no spurious correlation. In our experiments, $b = 0.7$. The label and ground-truth explanation for each graph are determined solely by the motif it contains.

**BA-HouseAndGrid (Bui et al., 2024).** The BA-HouseAndGrid dataset contains Barabási–Albert graph graphs extended with the house motif, the grid motif, or both. Graphs are labeled 1 if they contain both motifs, otherwise, they are labeled 0.

**Alkane-Carbonyl (Sanchez-Lengeling et al., 2020).** The Alkane-Carbonyl dataset comprises 4,326 molecular graphs partitioned into two classes based on the functional groups. A molecule is labeled positive when it contains both alkane and carbonyl groups, which also serve as the ground-truth explanation.

**Fluorid-Carbonyl (Sanchez-Lengeling et al., 2020).** The Fluorid-Carbonyl dataset comprises 8,671 molecular graphs partitioned into two classes based on functional groups. A molecule is labeled positive if it contains both fluoride atoms and a carbonyl group, which also serve as the ground-truth explanation.

**Benzene (Sanchez-Lengeling et al., 2020).** The Benzene dataset comprises 12,000 molecular graphs from the ZINC15 (Sterling & Irwin, 2015) database, partitioned into two classes based on the presence of benzene rings. A molecule is labeled positive if it contains at least one benzene ring. Each benzene ring in the molecule serves as a distinct ground-truth explanation.

**BA-Motif-Volume (Zhang et al., 2024).** In BA-Motif-Volume dataset, each graph is constructed from a Barabási–Albert graph with an attached five-cycle motif. Node features are assigned random

float values in the range [0.00, 100.00], and the regression label for each graph is defined as the sum of node feature values over the motif.

**House-Grid-Volume.** The House-Grid-Volume dataset is derived from BA-HouseGrid (Bui et al., 2024) by replacing all node features with random floats sampled from [0.00, 100.00]. The regression label for each graph is defined as the sum of node features within its motif.

**BA-Motif-Counting (Zhang et al., 2024).** The BA-Motif-Counting dataset consists of graphs created by attaching a randomly sampled number of five-cycle motifs (with the number varying from $\{0, \ldots, 10\}$ in our experiment) to a Barabási–Albert random graph. The number of motifs in each graph serving as its regression label.

**Triangles (Chen et al., 2020b).** The Triangles dataset is constructed following the prior work (Chen et al., 2020b), consisting of 5,000 Erdős–Rényi random graphs denoted as $ER(m, p)$, where $m = 30$ is the number of nodes in each graph and $p = 0.1$ is the edge existence probability. The regression label for each graph is the number of triangles it contains. In our experiments, we use a subset of 1,000 graphs randomly sampled from this dataset.

**Crippen (Delaney, 2004).** The Crippen dataset contains 1,127 molecules with corresponding aqueous solubility measurements from the Delaney solubility (Delaney, 2004) dataset and assigns node weights using the Crippen model (Wildman & Crippen, 1999). Following prior work (Sanchez-Lengeling et al., 2020), we adopt this dataset and generate edge weights as the average of incident node weights.

**House-OrGrid-Volume.** The House-OrGrid-Volume dataset is derived from BA-HouseOrGrid (Bui et al., 2024) by replacing all node features with random floats sampled from [0.00, 100.00]. The regression label for each graph is defined as the sum of node features within its motif.

### E.3 BASELINES

To assess effectiveness, we incorporate various post-hoc methods, including GNNExplainer, PGExplainer, MetaGNN, MatchExplainer, MixupExplainer, ProxyExplainer, and RegExplainer, as well as the gradient-based GRAD and the attention-based ATT.

**GRAD (Ying et al., 2019).** GRAD is a gradient-based method that generates weights to edges and nodes by computing the gradients of the loss function of GNN with respect to the adjacency matrix and node features.

**ATT (Veličković et al., 2017).** ATT is a graph attention network that learns attention weights for edges in the input graph, with these weights are adopted as a proxy measure of edge importance.

**GNNExplainer (Ying et al., 2019).** GNNExplainer learns soft masks over edges and node features for each instance by minimizing the mutual information between the original graph and the prediction results. The explanatory subgraphs are obtained via element-wise multiplication of the learned soft masks with the original graph.

**PGExplainer (Luo et al., 2020).** PGExplainer extends GNNExplainer by parameterizing the explanation generation process with a trainable explainer, and generates a soft mask to produce the explanatory subgraph.

**MetaGNN (Spinelli et al., 2022).** MetaGNN trains GNNs to be inherently interpretable by incorporating a meta-explainer, which generates post-hoc explanations during training.

**TAGExplainer (Xie et al., 2022).** TAGExplainer is a task-agnostic explainer trained in a self-supervised manner. It explains GNN predictions by separating the explainer into embedding and downstream components, enabling the explanation of GNN embedding models with unseen downstream tasks and allowing efficient explanation of multitask models.

**MatchExplainer (Wu et al., 2023).** MatchExplainer explains GNN predictions by matching shared subgraph patterns using graph edit distance (GED) as the similarity metric, and this non-parametric subgraph matching approach inherently avoids optimization bias.

**MixupExplainer (Zhang et al., 2023).** MixupExplainer mitigates the distribution shift present in previous methods by mixing explanatory subgraphs with the label-irrelevant parts of other randomly sampled graphs in a non-parametric manner.

**ProxyExplainer (Chen et al., 2024).** ProxyExplainer performs autoencoders to reconstruct the explanatory and label-irrelevant parts, then combines them to generate proxy graphs in a parametric manner, approximating the original distribution to mitigate the OOD issue.

**RegExplainer (Zhang et al., 2024).** RegExplainer addresses explainability in graph regression tasks by combining mixup with contrastive learning to mitigate distribution shift and tackle the challenge of continuously ordered labels.

## F   PROOF OF PROPERTY 1

*Proof.* The mutual information between the explanatory subgraph $G^*$ and the label $Y$ can be formulated as the Kullback–Leibler (KL) divergence between the joint distribution $p(G^*, Y)$ and the product of its marginals:

$$I(G^*; Y) = \mathrm{KL}(p(G^*, Y) \,\|\, p(G^*)p(Y)).$$

The Donsker–Varadhan (DV) variational representation of KL divergence Poole et al. (2019) provides the following inequality:

$$\mathrm{KL}(p \,\|\, q) = \sup_T \left\{ \mathbb{E}_p[T] - \log \mathbb{E}_q[e^T] \right\},$$

where the supremum is taken over all measurable functions $T : \mathcal{G} \times \mathcal{Y} \to \mathbb{R}$ such that the expectations exist. The bound is tight when $T(x) = \log \frac{p(x)}{q(x)}$.

Applying this representation to the mutual information yields:

$$I(G^*; Y) \geq \mathbb{E}_{p(G^*, Y)}[T(G^*, Y)] - \log \mathbb{E}_{p(G^*)p(Y)} \left[ e^{T(G^*, Y)} \right].$$

This variational characterization establishes a theoretical basis for estimating mutual information by choosing tractable parameterizations of the critic function $T$.

$\square$

## G   PROOF OF PROPERTY 2

*Proof.* To instantiate the critic function $T(G^*, Y)$, we leverage the fact that each label $Y$ in the dataset is paired with a real input graph $G_Y$, i.e., $(G_Y, Y) \sim p(G, Y)$. Instead of encoding the label $Y$ directly, we obtain its representation via the associated graph $G_Y$, which reflects the semantics of $Y$ in the same space as $G^*$. This design ensures that both $G^*$ and $Y$ are represented in the same embedding space, while preserving the theoretical foundation of estimating $I(G^*; Y)$. Let $f(\cdot)$ denote an encoder that maps a graph to its representation, and define $\boldsymbol{h}_{G^*} = f(G^*)$ and $\boldsymbol{h}_Y = f(G_Y)$. The similarity between two representations is defined as $\mathrm{sim}(u, v) := u^\top v$, where $u$ and $v$ are $\ell_2$-normalized vectors.

We instantiate the Donsker–Varadhan critic function $T : \mathcal{G} \times \mathcal{Y} \to \mathbb{R}$ as:

$$T(G^*, Y) := \mathrm{sim}(\boldsymbol{h}_{G^*}, \boldsymbol{h}_Y) + \log(N-1),$$

where $N$ is the total number of samples (including one positive and $N-1$ negatives) in each estimation batch. We first evaluate the expectation under the joint distribution $p(G^*, Y)$ (positive pairs):

$$\mathbb{E}_{p(G^*, Y)}[T(G^*, Y)] = \mathbb{E}_{p(G^*, Y)} \left[ \mathrm{sim}^+ + \log(N-1) \right],$$

where $\text{sim}^+ := \text{sim}(\boldsymbol{h}_{g^*}, \boldsymbol{h}_{y^+})$ denotes the similarity score of the positive pair.

To evaluate the negative term, we apply Jensen's inequality to the expectation under the product of marginals:

$$- \log \mathbb{E}_{p(G^*)p(Y)} \left[ e^{T(G^*, Y)} \right] \geq -\mathbb{E}_{G^* \sim p(G^*)} \left[ \log \mathbb{E}_{Y \sim p(Y)} \left[ e^{T(G^*, Y)} \right] \right].$$

We approximate the inner expectation using $N-1$ i.i.d. samples $\{y_j^-\}_{j=1}^{N-1}$ drawn from $p(Y)$:

$$\mathbb{E}_Y[e^{T(G^*, Y)}] \approx \frac{1}{N-1} \sum_{j=1}^{N-1} \exp\left(\text{sim}_j^- + \log(N-1)\right) = (N-1) \cdot \frac{1}{N-1} \sum_{j=1}^{N-1} \exp\left(\text{sim}_j^-\right),$$

where $\text{sim}_j^- := \text{sim}(\boldsymbol{h}_{g^*}, \boldsymbol{h}_{y_j^-})$ denotes the similarity between $G^*$ and each negative sample $y_j^-$.

Taking the logarithm, we obtain:

$$\log \mathbb{E}_Y[e^{T(G^*, Y)}] \approx \log(N-1) + \log\left(\frac{1}{N-1} \sum_{j=1}^{N-1} \exp\left(\text{sim}_j^-\right)\right),$$

and thus, the negative term becomes:

$$-\log \mathbb{E}_{p(G^*)p(Y)}[e^{T(G^*, Y)}] \geq -\log(N-1) - \mathbb{E}_{\mathcal{P}}\left[\log\left(\frac{1}{N-1} \sum_{j=1}^{N-1} \exp\left(\text{sim}_j^-\right)\right)\right],$$

where $\mathcal{P} := p(G^*, Y_0) \prod_{j=1}^{N-1} p(Y_j^-)$ denotes the joint sampling distribution comprising one positive pair $(G^*, Y_0)$ and $N-1$ independent negative samples $\{Y_j^-\}$.

Combining the positive and negative components of the DV bound:

$$I(G^*; Y) \geq \mathbb{E}_{\mathcal{P}}\left[\text{sim}^+ + \log(N-1)\right] - \log(N-1) - \mathbb{E}_{\mathcal{P}}\left[\log\left(\frac{1}{N-1} \sum_{j=1}^{N-1} \exp\left(\text{sim}_j^-\right)\right)\right]$$

$$= \mathbb{E}_{\mathcal{P}}\left[\text{sim}^+\right] - \mathbb{E}_{\mathcal{P}}\left[\log\left(\frac{1}{N-1} \sum_{j=1}^{N-1} \exp\left(\text{sim}_j^-\right)\right)\right]$$

$$= \mathbb{E}_{\mathcal{P}}\left[\log\left(\exp\left(\text{sim}^+\right)\right) - \log\left(\frac{1}{N-1} \sum_{j=1}^{N-1} \exp\left(\text{sim}_j^-\right)\right)\right]$$

$$= \mathbb{E}_{\mathcal{P}}\left[\log\left(\frac{\exp\left(\text{sim}^+\right)}{\frac{1}{N-1} \sum_{j=1}^{N-1} \exp\left(\text{sim}_j^-\right)}\right)\right]$$

$$= \log(N-1) + \mathbb{E}_{\mathcal{P}}\left[\log\left(\frac{\exp\left(\text{sim}^+\right)}{\sum_{j=1}^{N-1} \exp\left(\text{sim}_j^-\right)}\right)\right].$$

Since the logarithm is monotonically increasing, augmenting the denominator with the positive term yields a smaller ratio and hence a looser (but valid) bound:

$$I(G^*; Y) \geq \log(N-1) + \mathbb{E}_{\mathcal{P}}\left[\log\left(\frac{\exp\left(\text{sim}^+\right)}{\exp\left(\text{sim}^+\right) + \sum_{j=1}^{N-1} \exp\left(\text{sim}_j^-\right)}\right)\right].$$

We define the InfoNCE estimator as:

$$\ell(G^*, Y) := \log \frac{\exp\left(\text{sim}(\boldsymbol{h}_{g^*}, \boldsymbol{h}_{y^+})\right)}{\sum_{j=0}^{N-1} \exp\left(\text{sim}(\boldsymbol{h}_{g^*}, \boldsymbol{h}_{y_j})\right)},$$

where $\{y_j\}_{j=0}^{N-1}$ includes one positive label $y_0 = y^+$ and $N-1$ negative samples $y_j^-$.

Thus, we arrive at the final variational lower bound:

$$\boxed{I(G^*; Y) \geq \log(N-1) + \mathbb{E}_{\mathcal{P}}\left[\ell(G^*, Y)\right]}.$$

$\square$

# H EXTENSIVE EXPERIMENTS

## H.1 MULTI-TASK EXPLANATION QUALITY EVALUATION

Table 5: AUC-ROC edge-level explanation accuracy on three node classification datasets. Higher scores indicate better performance.

| Method | BA-Shapes | BA-Community | Tree-Grid |
|---|---|---|---|
| PGExplainer | $0.902 \pm 0.027$ | $0.653 \pm 0.111$ | $0.205 \pm 0.113$ |
| MixupExplainer | $0.801 \pm 0.145$ | $0.654 \pm 0.113$ | $0.179 \pm 0.054$ |
| TAME | $\mathbf{0.991 \pm 0.001}$ | $\mathbf{0.906 \pm 0.154}$ | $\mathbf{0.678 \pm 0.039}$ |

Table 6: Robust Fidelity scores for explanations on three link prediction tasks.

| Method | BA-Shapes | | BA-Community | | Tree-Grid | |
|---|---|---|---|---|---|---|
| | $Fid_{\alpha_1,+} \uparrow$ | $Fid_{\alpha_2,-} \downarrow$ | $Fid_{\alpha_1,+} \uparrow$ | $Fid_{\alpha_2,-} \downarrow$ | $Fid_{\alpha_1,+} \uparrow$ | $Fid_{\alpha_2,-} \downarrow$ |
| PGExplainer$_\text{link}$ | $0.201 \pm 0.197$ | $0.285 \pm 0.177$ | $0.132 \pm 0.135$ | $0.229 \pm 0.101$ | $0.112 \pm 0.048$ | $0.178 \pm 0.053$ |
| TAME$_\text{link}$ | $\mathbf{0.225 \pm 0.139}$ | $\mathbf{0.044 \pm 0.077}$ | $\mathbf{0.221 \pm 0.101}$ | $\mathbf{0.016 \pm 0.038}$ | $\mathbf{0.166 \pm 0.057}$ | $\mathbf{0.021 \pm 0.028}$ |

TAME introduces a task-agnostic GIB objective that supports explanation learning through representation-level contrastive training, making it applicable to a broad range of tasks beyond graph-level explanations, including node classification and link prediction. For node classification task, we extract a 3-hop ego subgraph centered on the target node as the target graph to be explained. For link prediction task, we follow SEAL Zhang & Chen (2018) to construct a local subgraph around the target edge. TAME then applies the same perturbation-based procedure as in graph-level tasks to identify critical subgraphs within these target graphs as explanations. Further experimental details for both tasks are provided in Appendix E.

For the node classification task, we conduct experiments on the BA-Shapes, BA-Community, and Tree-Grid datasets, comparing our method TAME against representative baselines, including PG-Explainer Luo et al. (2020) and MixupExplainer Zhang et al. (2023). As shown in Table 5, TAME consistently outperforms these baselines across all datasets. The most significant improvement is observed on the Tree-Grid dataset, where TAME achieves a performance gain of $0.473$. As ground-truth explanations are unavailable for these datasets in link prediction task, we adopt the robust fidelity metrics $Fid_{\alpha_1,+}$ and $Fid_{\alpha_2,-}$ Zheng et al. (2023) for evaluation. The results in Table 6 show that TAME$_\text{link}$ consistently outperforms PGExplainer$_\text{link}$ across all datasets under both metrics. Overall, the experimental results across node classification, link prediction, graph classification, and graph regression demonstrate that, compared to baseline methods, TAME consistently generates in-distribution mixup graphs and produces faithful explanations.

## H.2 EXPLANATION QUALITY EVALUATION VIA ROBUST FIDELITY METRICS

Additionally, we follow existing works Yuan et al. (2021); Bui et al. (2024) to evaluate the quality of the identified explanations via fidelity-based metrics. Owing to the reliability issue caused by the OOD problem in standard fidelity metrics Amara et al. (2023), we utilize the robust fidelity measures SimOAR Fang et al. (2023) with default perturbation ratio $0.1$ and $(Fid_{\alpha_1,+}, Fid_{\alpha_2,-})$ Zheng et al. (2023)with default parameters $\alpha_1 = 0.1$, $\alpha_2 = 0.9$. Table 7 demonstrates that the quality of the explanatory subgraphs obtained by TAME outperform all baselines under both metrics.

## H.3 HYPER-PARAMETER SENSITIVITY STUDY

We further investigate the sensitivity of our proposed method to the hyperparameter $\beta$, which controls the weight of the BCE loss. Specifically, we conduct experiments on four representative datasets, including the Ba-HouseAndGrid and Benzene classification datasets, as well as the BA-Motif-Volume and House-Grid-Volume regression datasets. We vary $\beta$ from 0.1 to 1.0 in increments of 0.1 while keeping the contrastive learning weight fixed at 1.0. The experiments are conducted under ten random seeds, where the solid lines represent the average AUC of our method and the dashed

Table 7: Explanation Quality on Classification (Alkane-Carbonyl, BA-HouseGrid) and Regression (House-OrGrid-Volume, BA-Motif-Volume) datasets via SimOAR and Robust Fidelity.

| Method | Alkane-Carbonyl | BA-HouseGrid | BA-HouseGrid | |
|---|---|---|---|---|
| | SimOAR ↑ | SimOAR ↑ | $Fid_{\alpha_1,+}$ ↑ | $Fid_{\alpha_2,-}$ ↓ |
| GNNExplainer | $0.780 \pm 0.006$ | $0.649 \pm 0.006$ | $0.021 \pm 0.049$ | $0.241 \pm 0.370$ |
| PGExplainer | $0.802 \pm 0.050$ | $0.667 \pm 0.080$ | $0.012 \pm 0.036$ | $0.319 \pm 0.433$ |
| MixupExplainer | $0.839 \pm 0.035$ | $0.767 \pm 0.092$ | $0.026 \pm 0.045$ | $0.281 \pm 0.408$ |
| ProxyExplainer | $0.829 \pm 0.011$ | $0.720 \pm 0.082$ | $0.016 \pm 0.041$ | $0.245 \pm 0.392$ |
| Ours | $\mathbf{0.857 \pm 0.050}$ | $\mathbf{0.834 \pm 0.067}$ | $\mathbf{0.038 \pm 0.076}$ | $\mathbf{0.072 \pm 0.244}$ |

| Method | House-OrGrid-Volume | BA-Motif-Volume | BA-Motif-Volume | |
|---|---|---|---|---|
| | SimOAR ↑ | SimOAR ↑ | $Fid_{\alpha_1,+}$ ↑ | $Fid_{\alpha_2,-}$ ↓ |
| MixupExplainer | $0.810 \pm 0.075$ | $0.215 \pm 0.058$ | $0.158 \pm 0.315$ | $1.298 \pm 1.442$ |
| RegExplainer | $0.752 \pm 0.086$ | $0.204 \pm 0.057$ | $0.184 \pm 0.332$ | $1.162 \pm 1.439$ |
| Ours | $\mathbf{0.840 \pm 0.094}$ | $\mathbf{0.247 \pm 0.056}$ | $\mathbf{0.413 \pm 0.396}$ | $\mathbf{0.006 \pm 0.117}$ |

lines correspond to the second-best performing method. The experimental results are presented in Figure 7. The results indicate that TAME maintains stable performance across both classification and regression tasks. For some datasets, the AUC slightly decreases and exhibits larger variance when the weight of the BCE loss is small. This phenomenon can be attributed to the role of the BCE loss in constraining the edge weights to remain faithful to the explanation structure extracted by graph pooling, while simultaneously reducing the weights of non-explanatory structures. Without this constraint, the edge weights may not be effectively optimized, leading to a decline in performance.

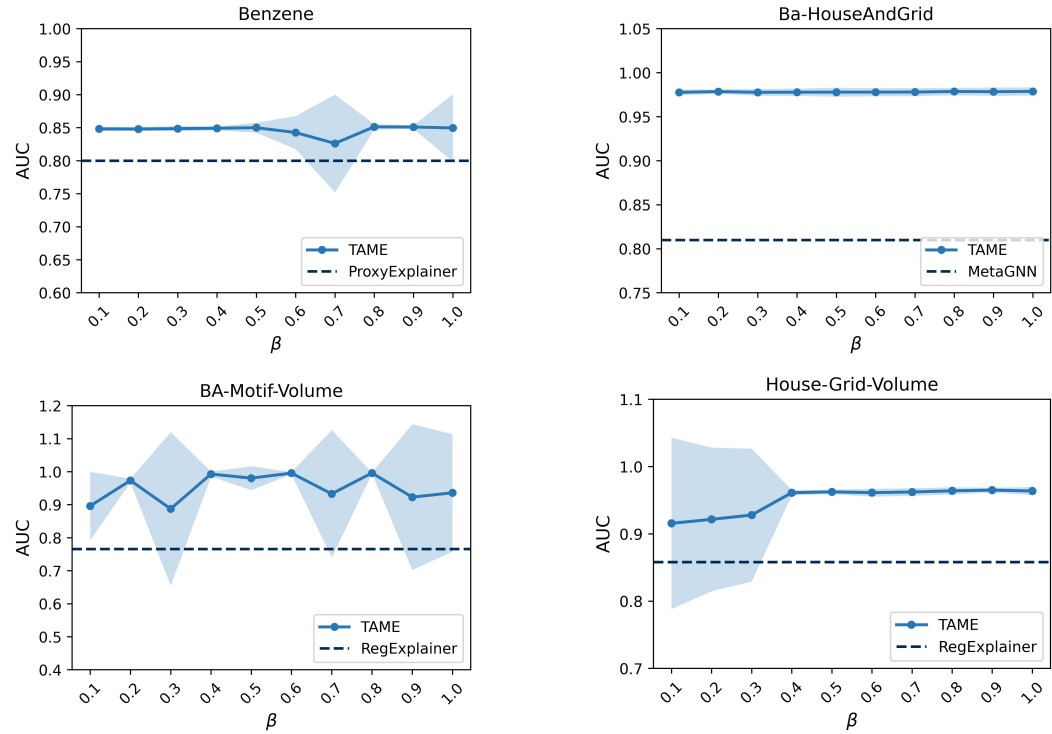

Figure 7: Sensitivity analysis of hyperparameter $\beta$ for BCE loss.

## H.4 Backbone Models Analysis

We conduct a performance analysis of common backbone models employed by explainability methods on both classification and regression tasks. For the classification task, model performance is evaluated using ACC and Macro_F1 metrics on four synthetic datasets, including BA-2Motifs, BA-HouseGrid, BA-HouseAndGrid, and SPMotif, as well as three real-world datasets including Alkane-Carbonyl, Fluoride-Carbonyl, and Benzene. For the regression task, RMSE and MAPE are adopted as evaluation metrics on five synthetic datasets including BA-Motif-Volume, BA-Motif-Counting, Triangles, House-Grid-Volume, and House-OrGrid-Volume, along with one real-world dataset, Crippen.

Experimental results demonstrate that GCN and GIN achieve strong performance in classification tasks, while GCN consistently attains the best performance across regression benchmarks. Therefore, to ensure that the explanation quality is not confounded by the performance of the backbone model, we choose GCN as the backbone model for explainer evaluation.

| Dataset | GCN | | GIN | | GAT | |
|---|---|---|---|---|---|---|
| | ACC | Macro_F1 | ACC | Macro_F1 | ACC | Macro_F1 |
| BA-2motifs | 99.00 | 98.98 | **100.00** | **100.00** | 44.00 | 30.55 |
| BA-HouseGrid | 99.90 | 99.89 | **100.00** | **100.00** | 49.10 | 32.93 |
| BA-HouseAndGrid | 98.40 | 98.39 | **99.90** | **99.89** | 48.90 | 32.84 |
| SPMotif | **96.16** | **96.15** | 95.88 | 95.89 | 34.05 | 18.98 |
| Alkane-Carbonyl | **95.57** | **95.13** | 92.03 | 91.24 | 94.69 | 94.31 |
| Fluoride-Carbonyl | 94.35 | 89.99 | **97.23** | **95.03** | 83.29 | 45.44 |
| Benzene | 90.58 | 90.53 | **92.66** | **92.65** | 79.08 | 79.06 |

Table 8: Comparison of GCN, GIN, and GAT on classification datasets using ACC and Macro-F1 metrics.

| Dataset | GCN | | GIN | | GAT | |
|---|---|---|---|---|---|---|
| | RMSE | MAPE | RMSE | MAPE | RMSE | MAPE |
| BA-Motif-Volume | **234.20** | **7.47** | 398.96 | 14.94 | 485.31 | 17.45 |
| BA-Motif-Counting | **0.39** | — | 1.94 | — | 2.84 | — |
| Triangles | **2.00** | — | 2.46 | — | 2.55 | — |
| House-Grid-Volume | **11.32** | **2.45** | 94.54 | 29.56 | 49.85 | 13.88 |
| House-OrGrid-Volume | **34.15** | **6.69** | 152.14 | 28.17 | 53.64 | 10.27 |
| Crippen | **0.92** | **30.71** | 1.91 | 123.17 | 1.16 | 74.68 |

Table 9: Comparison of GCN, GIN, and GAT on regression datasets using RMSE and MAPE metrics. MAPE values are marked with dashes for BA-Motif-Counting and Triangles due to the presence of label-zero samples that make MAPE undefined.

### H.5 EXTENSIVE CASE STUDY

#### H.5.1 VISUALIZATION OF ADDITIONAL EXPLANATION RESULTS

We further present extensive visualization experiments on both classification and regression datasets. The classification datasets include BA-2Motifs, BA-HouseAndGrid, and Benzene, as shown in Figure 8, Figure 9, and Figure 10, respectively. The regression datasets include BA-Motif-Volume, and House-OrGrid-Volume, as presented in Figure 11, and Figure 12, respectively. For each dataset, we randomly select three target graphs to evaluate different explainers.

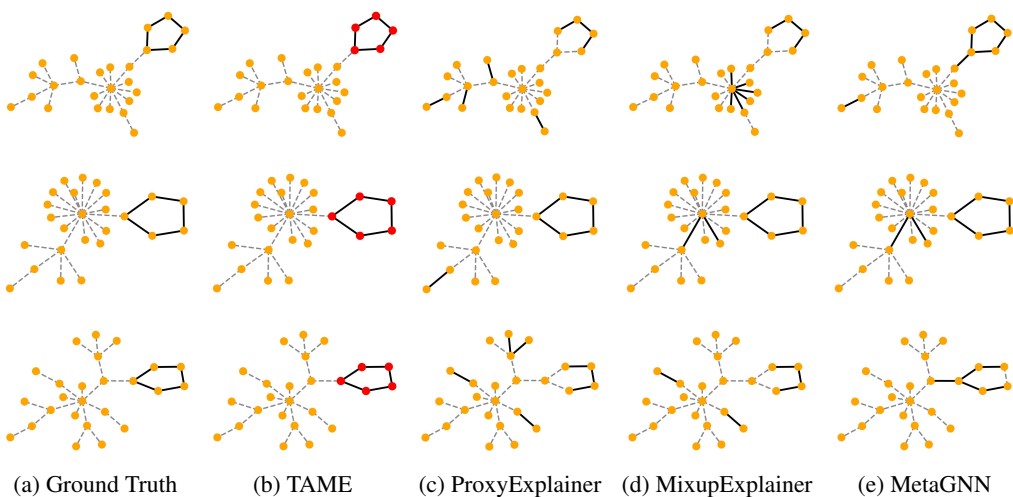

(a) Ground Truth      (b) TAME      (c) ProxyExplainer    (d) MixupExplainer    (e) MetaGNN

Figure 8: Visualization of explanation on BA-2motifs.

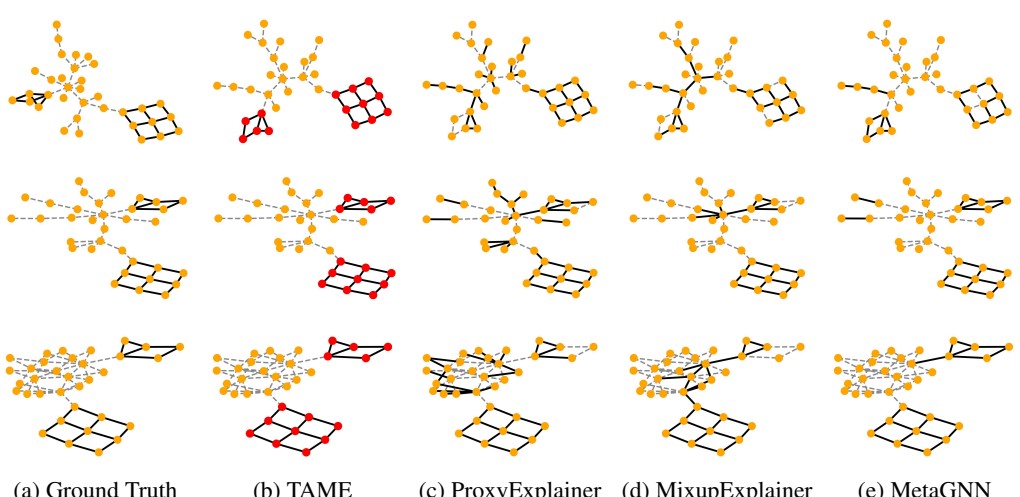

(a) Ground Truth      (b) TAME      (c) ProxyExplainer    (d) MixupExplainer    (e) MetaGNN

Figure 9: Visualization of explanation on BA-HouseAndGrid.

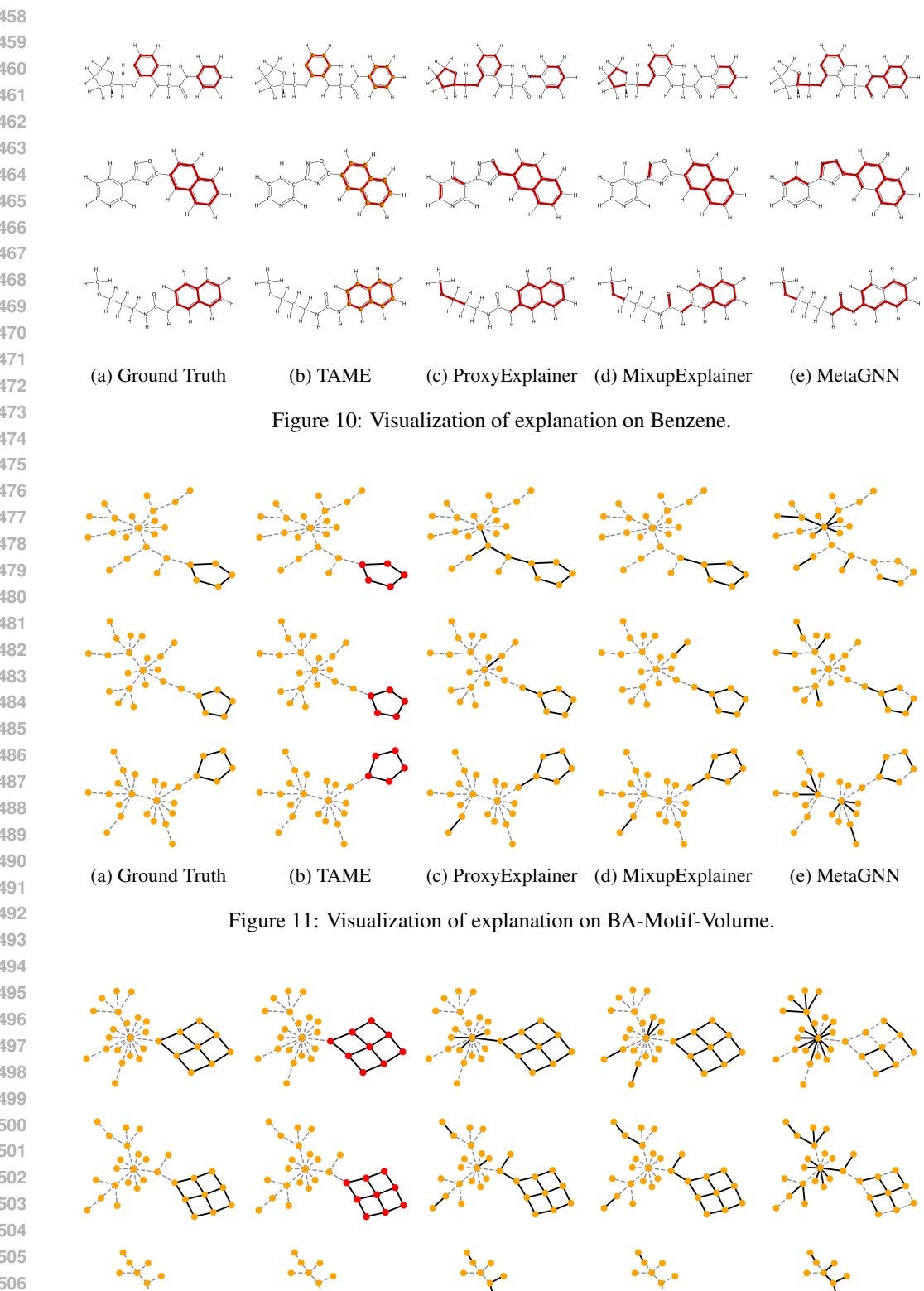

(a) Ground Truth    (b) TAME    (c) ProxyExplainer    (d) MixupExplainer    (e) MetaGNN

Figure 10: Visualization of explanation on Benzene.

(a) Ground Truth    (b) TAME    (c) ProxyExplainer    (d) MixupExplainer    (e) MetaGNN

Figure 11: Visualization of explanation on BA-Motif-Volume.

(a) Ground Truth    (b) TAME    (c) ProxyExplainer    (d) MixupExplainer    (e) MetaGNN

Figure 12: Visualization of explanation on House-Grid-Volume.

### H.5.2 VISUALIZATION OF ADDITIONAL MIXUP RESULTS

We present the mixup results on the BA-HouseGrid classification dataset and the BA-Motif-Volume regression dataset in Figure 13 and Figure 14, respectively. For each dataset, three graph samples are randomly selected to generate the mixup results, where the edge color intensity indicates the corresponding weight magnitude. It should be noted that ProxyExplainer employs a graph generation strategy to produce mixup results, which leads to fully connected edges. To facilitate visualization, we display only the top 64 edges with the highest weights, corresponding to the maximum number of edges considered in this experiment.

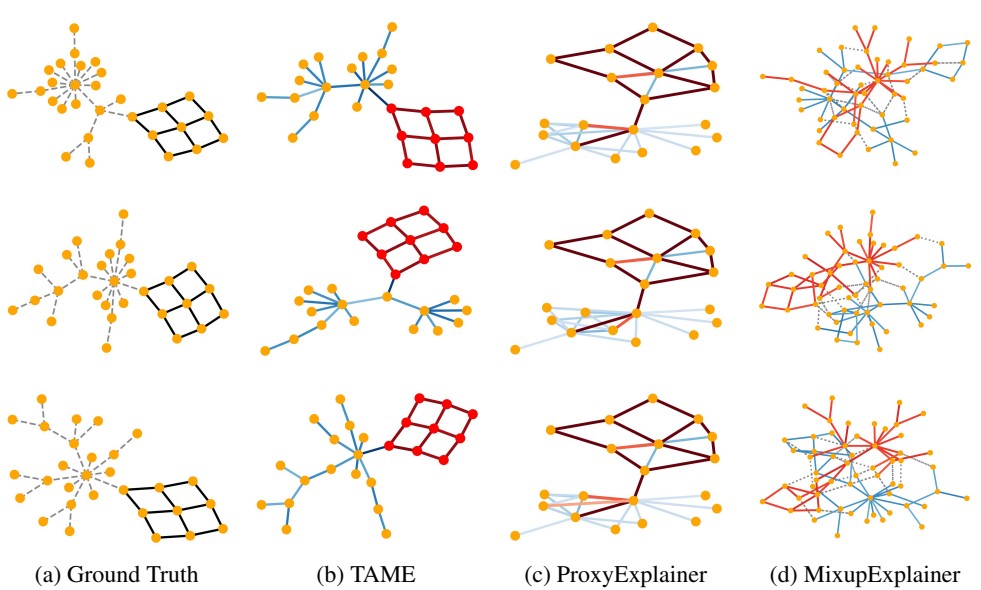

| (a) Ground Truth | (b) TAME | (c) ProxyExplainer | (d) MixupExplainer |

Figure 13: Visualization of mixup graphs on BA-HouseGrid.

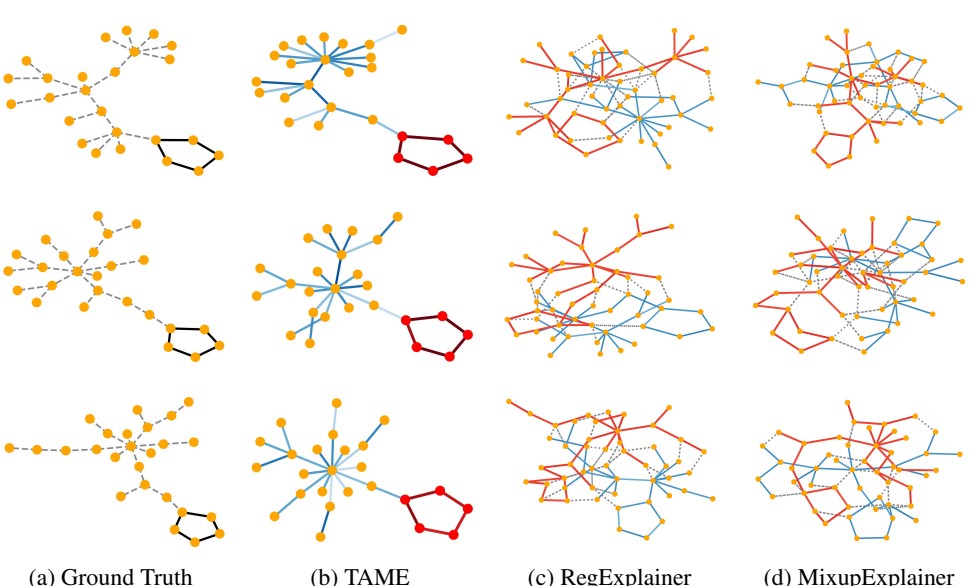

| (a) Ground Truth | (b) TAME | (c) RegExplainer | (d) MixupExplainer |

Figure 14: Visualization of mixup graphs on BA-Motif-Volume.

### H.5.3 GRAPH INVARIANCE ANALYSIS

To account for the rotation and translation invariance of graphs, we additionally shuffle the node indices of the extracted explanatory subgraph before replacing it into the neighbor graph during mixup. We conduct experiments on four classification datasets and four regression datasets, reporting the AUC results for both the original mixup process and the shuffled-index variant. The results in Table 10 and Table 11 show negligible differences between the two settings, indicating that TAME does not rely on node-order–specific artifacts or learn spurious features tied to fixed node indices. This behavior is likely due to the fact that node indices are implicitly determined by learnable value-based ordering during training, preventing the model from associating fixed index positions with spurious patterns.

Table 10: Graph-classification AUC comparison between original TAME and shuffled TAME.

| Method | Ba-HouseGrid | BaHouse-AndGrid | Benzene | Fluorid-Carbony |
|---|---|---|---|---|
| shuffle | $0.966 \pm 0.025$ | $0.979 \pm 0.003$ | $0.861 \pm 0.001$ | $0.795 \pm 0.027$ |
| ori | $0.965 \pm 0.026$ | $0.979 \pm 0.004$ | $0.861 \pm 0.004$ | $0.807 \pm 0.011$ |

Table 11: Graph-regression AUC comparison between original TAME and shuffled TAME.

| Method | BA-Motif-Volume | House-Grid-Volume | BA-Motif-Counting | Crippen |
|---|---|---|---|---|
| shuffle | $0.995 \pm 0.002$ | $0.958 \pm 0.006$ | $0.945 \pm 0.003$ | $0.554 \pm 0.028$ |
| ori | $0.997 \pm 0.001$ | $0.966 \pm 0.003$ | $0.946 \pm 0.004$ | $0.565 \pm 0.038$ |

To further examine this phenomenon, we perform a case study, as shown in Figure 15 and Figure 16. On BA-HouseGrid and BA-Motif-Volume, we randomly select a target graph and visualize the target graph, the sampled neighbor graph, the generated mixup graph, and the extracted explanatory subgraph. We compare the visualizations before and after shuffling node indices. The results show that although the internal ordering of nodes within the explanatory subgraph changes after shuffling, TAME consistently produces high-quality in-distribution mixup graphs and extracts accurate explanations.

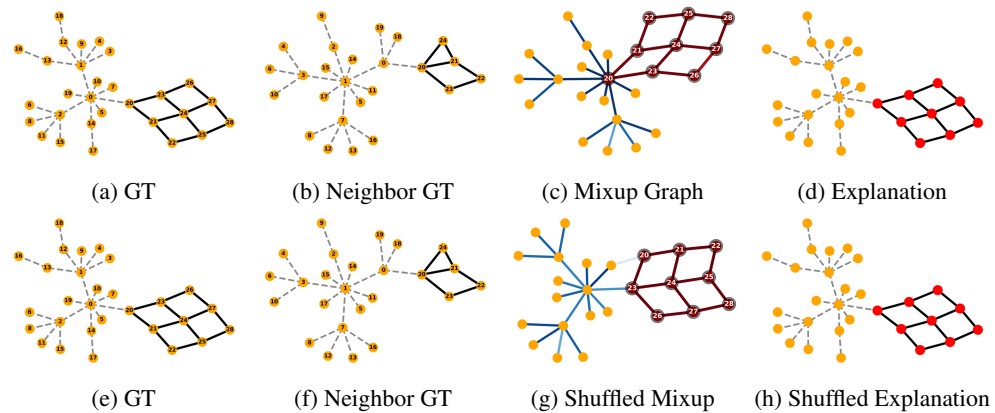

(a) GT    (b) Neighbor GT    (c) Mixup Graph    (d) Explanation

(e) GT    (f) Neighbor GT    (g) Shuffled Mixup    (h) Shuffled Explanation

Figure 15: Visualization on BA-HouseGrid.

### H.5.4 ANALYSIS OF POOLING SIZE IN STRUCTURAL MIXUP

TAME's structural mixup first applies top-k graph pooling to extract explanatory subgraphs from both the target graph and its sampled neighbor graph. These extracted subgraphs are then exchanged, which requires them to have matching sizes. In this section, we further examine this size constraint.

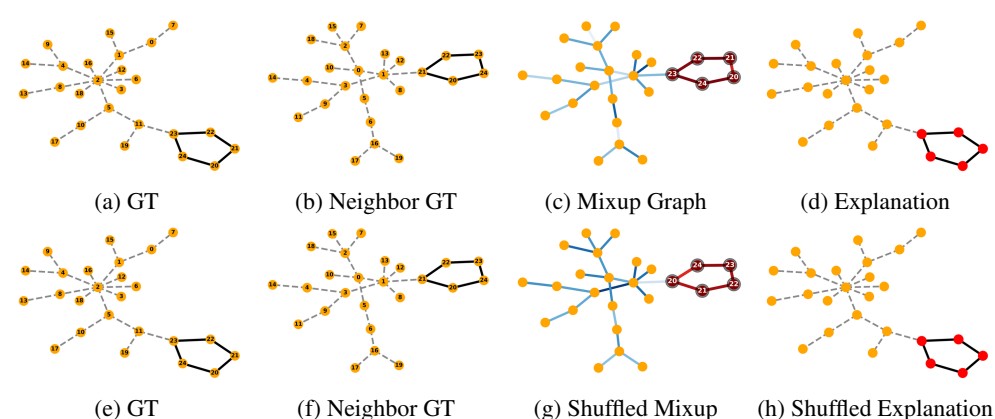

Figure 16: Visualization on BA-Motif-Volume.

When mixing with the near graph, the use of embedding-similarity sampling typically selects neighbors with similar labels—often sharing the same functional motifs or structural patterns. As a result, the target graph and its near graph generally produce explanatory subgraphs of similar size, allowing structural mixup to operate reliably.

When mixing with the far graph, the far graph usually differs in label and motif type, leading to potential mismatch in the intrinsic size of the explanatory subgraphs. In such cases, we rely on the assumption that explanatory subgraphs occupy only a small fraction of the full graph. By using a slightly larger top-k ratio, we implicitly pool a superset that includes the far graph's explanatory region, while any redundant nodes are later refined through the MLP-generated mask. As shown in Tables 1 and 2, TAME maintains strong performance on datasets where explanation sizes vary considerably (e.g., BA-HouseGrid, BAHouse-AndGrid, Fluorid-Carbony, BA-Motif-Counting), demonstrating good robustness and generalization ability.

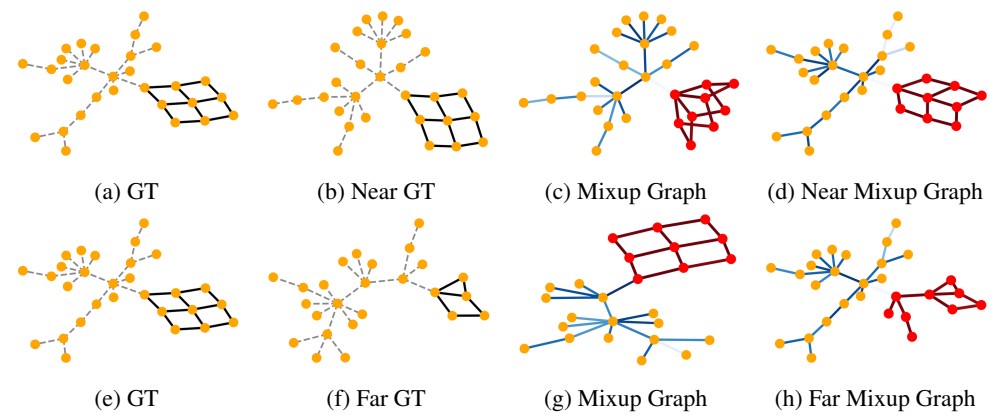

Figure 17: Visualization on BA-HouseGrid.

We further conduct a case study on BA-HouseGrid. For a randomly selected target graph, we visualize the structural mixup process with both the near graph and the far graph. The results shown in Figure 17 demonstrate that TAME consistently identifies and exchanges the critical motif structures in both cases, enabling TAME to extract accurate and faithful explanations.

