# OpenReview forum: "TAME: A Task-Agnostic Framework for Robust Graph Neural Network Explanations via Structural Mixup"
_ICLR.cc/2026/Conference — Submitted to ICLR 2026_

### Official Review · Reviewer_N8oL · 2025-10-17

**Soundness:** 3
**Presentation:** 4
**Contribution:** 2
**Rating:** 4
**Confidence:** 3

**Summary:**

This paper presents TAME, a unified, task-agnostic framework for GNN explanation that addresses both the limitations of task-specific methods and the distribution shift caused by subgraph extraction. The proposed approach integrates existing contrastive learning-based GIB objective for GNN explanation, enabling consistent explanation across both classification and regression tasks. To better generate in-distribution explanations, a novel mixup strategy built upon graph pooling is introduced.
Experiments show TAME achieves state-of-the-art performance in generating robust and interpretable explanations across both synthetic and real-world datasets.

**Strengths:**

- This paper extends the experimental boundaries of the GIB objective to classification tasks.

- This paper introduces a novel mixup strategy built upon graph pooling, which generates in-distribution explanations through hard structural perturbations.

**Weaknesses:**

1. The contrastive learning-based GIB objective for GNN explanation was proposed by RegExplainer [1], and its motivation is to address the issue that estimating I(G∗; Y) in regression tasks is not as straightforward as in classification tasks. Applying RegExplainer objective to classification tasks is also straightforward, and TAME merely further validates this objective in classification tasks—its contribution in this regard is somewhat limited.

2. The idea of Mix-up is also very similar to that of RegExplainer [1]. The innovation of TAME lies in replacing the soft-mask-based mixup strategy with a structural mixup strategy, which explicitly disentangles explanatory subgraphs from label-irrelevant ones. However, the structural mixup strategy involves the matching problem of two graphs and has many limitations:
   - Structural Mixup requires (A')* and A* to have the same size, but in practice, the sizes of effective explanatory subgraphs vary across different graphs.
   - A* replaces (A')* according to fixed-order node indices, which may introduce spurious features. Considering the rotation and translation invariance of graphs, it would be better to shuffle the node indices of A* first before replacing (A')*.

3. There is a lack of verification on whether the structural mixup strategy is more effective than other strategies (e.g., the soft-mask-based mixup strategy) in addressing the out-of-distribution (OOD) problem. The evaluation method for GNN explanation in benchmark [2] takes OOD into account, and the authors are advised to refer to [2] for further analysis.

4. Can TAME train a unified state-of-the-art explainer with shared weights for all tasks? If not, the term "TASK-AGNOSTIC FRAMEWORK" is somewhat confusing to me, and it is recommended to use "TASK-AGNOSTIC Objective" instead.

In summary, TAME's claim of being the first to propose a contrastive learning-based, task-agnostic GIB objective for GNN explanation is somewhat overstatement for me. The main contributions of TAME are twofold: first, it extends the experimental boundaries of the GIB objective (rather than proposing or optimizing the GIB objective itself); second, it replaces the soft-mask-based mixup strategy with a structural mixup strategy based on graph pooling, though the verification of this design is insufficient.

[1] RegExplainer: Generating Explanations for Graph Neural Networks in Regression Tasks
[2] Evaluating Post-hoc Explanations for Graph Neural Networks via Robustness Analysis

**Questions:**

Please refer to the above weakness section for suggestions and questions.

---

> ### Author Response · Authors · 2025-11-23
> **Response to W2**
>
> **W2:The idea of Mix-up is also very similar to that of RegExplainer[1]. The innovation of TAME lies in replacing the soft-mask-based mixup strategy with a structural mixup strategy, which explicitly disentangles explanatory subgraphs from label-irrelevant ones. However, the structural mixup strategy involves the matching problem of two graphs and has many limitations:**
> 1. Structural Mixup requires (A')* and A* to have the same size, but in practice, the sizes of effective explanatory subgraphs vary across different graphs.
> 2. A* replaces (A')* according to fixed-order node indices, which may introduce spurious features. Considering the rotation and translation invariance of graphs, it would be better to shuffle the node indices of A* first before replacing (A')*.
>
> **Response to W2.1:**
>
> We thank the reviewer for this valuable point. We acknowledge that TAME swaps explanation subgraphs of the same size during mixup, which may impose certain limitations. However, when sampling near graphs, the embedding-similarity–based strategy ensures that their explanation subgraphs have similar sizes. Thus, despite this constraint, the method is able to maintain reliable explanation performance.
>
> Importantly, our work overcomes the shortcomings of prior mixup-based methods that rely on soft masks, which tend to amplify distribution shift. As shown in the table, TAME produces mixup graphs that remain most faithful to the original graph distribution, providing a more reliable foundation for optimizing the GIB objective.
>
> We agree that developing a structural mixup strategy with dynamically sized explanations is of significant interest, and we plan to explore this direction in future work.
>
> | Method                                               | Ba-HouseGrid | Ba-HouseAndGrid | Benzene | Fluorid-Carbonyl |
> | -------------- | ------------ | --------------- | ------- | ---------------- |
> | $\text{Cos}(h_G, h_{G^*}) \uparrow$           | 0.688        | 0.613           | 0.821   | 0.918            |
> | $\text{Cos}(h_G, h_{G_{\text{mixup}}})$ | 0.781        | 0.966           | 0.907   | 0.895            |
> | $\text{Cos}(h_G, h_{G_{\text{proxy}}})$ | 0.802        | 0.970           | 0.565  | 0.723           |
> | $\text{Cos}(h_G, h_{G_{\text{our}}^{\text{(mix)}}})$         | **0.868**        | **0.971**           | **0.941**   | **0.937**            |
> | $\text{Euc}(h_G, h_{G^*}) \uparrow$             | 1.044        | 1.436           | 0.953   | 0.684            |
> | $\text{Euc}(h_G, h_{G_{\text{mixup}}})$ | 0.973       | 0.477          | 0.771  | 0.782           |
> | $\text{Euc}(h_G, h_{G_{\text{proxy}}})$ | 0.706       | 0.4424          | 1.471  | 1.272           |
> | $\text{Euc}(h_G, h_{G_{\text{our}}^{(\text{mix})}})$           | **0.661**        | **0.425**           | **0.578**   | **0.604**            |
>
> | Method                                               | BA-Motif-Volume | BA-Motif-Counting | Crippen | Triangles |
> | ---------------------------------------------------- | --------------- | ----------------- | ------- | --------- |
> | $\text{Cos}(h_G, h_{G^*}) \uparrow$           | 0.696           | 0.663             | 0.869   | 0.897     |
> | $\text{Cos}(h_G, h_{G_{\text{reg}}})$ | 0.876           | 0.950             | 0.847   | 0.953     |
> | $\text{Cos}(h_G, h_{G_{\text{our}}^{\text{(mix)}}})$         | **0.953**           | **0.969**             | **0.936**   | **0.985**     |
> |$\text{Euc}(h_G, h_{G^*}) \uparrow$           | 1.138           | 1.196             | 0.653   | 0.389     |
> | $\text{Euc}(h_G, h_{G_{\text{reg}}})$ | 0.775           | 0.485             | 0.794   | 0.247     |
> | $\text{Euc}(h_G, h_{G_{\text{our}}^{(\text{mix})}})$         | **0.458**           | **0.400**             | **0.431**   | **0.143**     |
>
>
> **Response to W2.2:**
>
> We thank the reviewer for raising the concern regarding node ordering and graph invariance. As noted, strictly replacing substructures according to fixed node indices may theoretically introduce spurious features related to node order, and we agree that this concern is reasonable. In TAME's structural replacement, reordering nodes does not affect the exchanged substructure itself, but it may alter the connections between the substructure and its surrounding environment, causing a minor impact on the generated mixup graphs.
>
> We consider shuffling node indices a promising improvement and are currently conducting experiments to evaluate its effect. We will update the manuscript with the results in the coming days.

---

> ### Author Response · Authors · 2025-11-23
> **Response to W3**
>
> **W3:There is a lack of verification on whether the structural mixup strategy is more effective than other strategies (e.g., the soft-mask-based mixup strategy) in addressing the out-of-distribution (OOD) problem. The evaluation method for GNN explanation in benchmark [2] takes OOD into account, and the authors are advised to refer to [2] for further analysis.**
>
> Response to W3:
>
> We thank the reviewer for pointing out the limitations in the OOD experiments and for suggesting new evaluation metrics.
>
> First, we further compared TAME with ProxyExplainer, MixupExplainer, and RegExplainer on four classification and four regression datasets, measuring the similarity and distance between the generated mixup graphs and the original graphs. The tabulated results provided in our response to W2 show that TAME achieves the best performance in both regression and classification tasks, indicating that the structural mixup strategy, via graph-pooling-based hard perturbations, more effectively mitigates distribution shift. Visualization in Appendix H.4.2 further demonstrates that the mixup graphs generated by TAME more closely align with the original graph distributions.
>
> And then, the benchmark [2] discusses the out-of-distribution (OOD) issues present in existing metrics for evaluating explanatory subgraphs (e.g., fidelity, DSE [1]). To address this, [2] proposes two more robust evaluation metrics, OAR and SimOAR, which assess explanation quality by measuring robustness against multiple perturbations applied to the non-explanatory parts. Following the suggestion in [2], we adopt SimOAR to evaluate our method, where a higher value indicates a more accurate explanation. As shown in the table below, our method achieves strong performance on both classification and regression tasks.
>
> In addition, we also considered more robust evaluation metrics (robust Fidelity[3]) in our experiments. Following the common experimental setup of $\alpha_1=0.1$ and $\alpha_2=0.9$, our method achieved superior results across all tested datasets, as shown in the following table for BA-HouseGrid and BA-MotifVolume.
>
> | Method | Alkane-Carbonyl ($\text{SimOAR} \uparrow$) | BA-HouseGrid ($\text{SimOAR} \uparrow$) | BA-HouseGrid ($Fid_{\alpha_1,+} \uparrow$) | BA-HouseGrid ($Fid_{\alpha_2,-} \downarrow$) |
> |--------|--------------------------------------------|------------------------------------------|---------------------------------------------|-----------------------------------------------|
> | GNNExplainer    | 0.780 ± 0.006 | 0.649 ± 0.006 | 0.021 ± 0.049 | 0.241 ± 0.370 |
> | PGExplainer     | 0.802 ± 0.050 | 0.667 ± 0.080 | 0.012 ± 0.036 | 0.319 ± 0.433 |
> | MixupExplainer  | 0.839 ± 0.035 | 0.767 ± 0.092 | 0.026 ± 0.045 | 0.281 ± 0.408 |
> | ProxyExplainer  | 0.829 ± 0.011 | 0.720 ± 0.082 | 0.016 ± 0.041 | 0.245 ± 0.392 |
> | Ours            | **0.857 ± 0.050** | **0.834 ± 0.067** | **0.038 ± 0.076** | **0.072 ± 0.244** |
>
> | Method          | BA-MotifVolume ($\text{SimOAR} \uparrow$) | House-OrGridVolume ($\text{SimOAR} \uparrow$) | BA-MotifVolume ($Fid_{\alpha_1,+} \uparrow$) | BA-MotifVolume ($Fid_{\alpha_2,-} \downarrow$) |
> |-----------------|---------------------------------------------|------------------------------------------------|-----------------------------------------------|------------------------------------------------|
> | MixupExplainer  | 0.215 ± 0.058                               | 0.810 ± 0.075                                  | 0.158 ± 0.315                                 | 1.298 ± 1.442                                  |
> | RegExplainer  | 0.204 ± 0.057                               | 0.752 ± 0.086                                  | 0.184 ± 0.332                                 | 1.162 ± 1.439                                  |
> | Ours            | **0.247 ± 0.056**                           | **0.840 ± 0.094**                              | **0.413 ± 0.396**                             | **0.006 ± 0.117**                              |
>
> [1]Wu, Ying-Xin, et al. "Deconfounding to explanation evaluation in graph neural networks." arXiv preprint arXiv:2201.08802 (2022).
>
> [2]Fang, Junfeng, et al. "Evaluating post-hoc explanations for graph neural networks via robustness analysis." Advances in neural information processing systems 36 (2023): 72446-72463.
>
> [3]Zheng, Xu, et al. "Towards robust fidelity for evaluating explainability of graph neural networks." arXiv preprint arXiv:2310.01820 (2023).

---

> ### Author Response · Authors · 2025-11-23
> **Response to W1,W4**
>
> **W1:The contrastive learning-based GIB objective for GNN explanation was proposed by RegExplainer [1], and its motivation is to address the issue that estimating I(G∗; Y) in regression tasks is not as straightforward as in classification tasks. Applying RegExplainer objective to classification tasks is also straightforward, and TAME merely further validates this objective in classification tasks—its contribution in this regard is somewhat limited.**
>
> Response to W1:
>
> We thank the reviewer for raising this point. Although both RegExplainer and TAME employ a contrastive MI-based objectives, the two methods rely on different derivation routes. RegExplainer first reformulates $I(G^\*;Y)$ through an intermediate variable $Y^\*$ and then derives an InfoNCE-style loss under assumptions tailored to continuous regression outputs. In contrast, TAME directly derives a Donsker–Varadhan(DV) based variational lower bound for $I(G^\*;Y)$ without introducing task-specific intermediate variables, forming a task-agnostic GIB objective.
>
> RegExplainer's theoretical justification does not seamlessly extend to classification for two reasons.
>
> (1) Its InfoNCE objective is derived under the assumption that the similarity satisfies $\mathrm{sim}(Y^\*,Y)\propto p(Y\mid Y^\*)/p(Y)$, which is reasonable for continuous regression targets but is not parameterized or justified for discrete labels. As a result, the InfoNCE loss used by RegExplainer is not guaranteed to remain a valid lower bound on $I(Y^\*;Y)$ in classification tasks. Moreover, when $Y$ is categorical, neither $p(Y\mid Y^\*)$ nor the ratio $p(Y\mid Y^\*)/p(Y)$ admits a continuous or smoothly parameterizable form compatible with the similarity function, making the density-ratio interpretation fundamentally incompatible with classification.
>
> (2) The proof further relies on a deterministic many-to-one mapping $Y^\*=h(G^\*)$ and on grouping explanation instances via $G(y^\*)$, assumptions that are explicitly introduced for continuous regression outputs and are not established for discrete class labels or softmax predictions. In classification, $Y^\*$ is categorical or probabilistic rather than continuous, so it cannot induce neighborhoods or explanation sets that satisfy the structural requirements of the original proof, and the mapping $h(G^\*)$ is no longer well-defined in the sense required by the argument.
>
> In contrast, TAME adopts a task-agnostic derivation pathway grounded directly in the Donsker–Varadhan representation of mutual information:
> $$
> I(G^\*;Y)\ge \mathbb{E}\_{p(G^\*,Y)}[T(G^\*,Y)] - \log \mathbb{E}\_{p(G^\*)p(Y)}[e^{T(G^\*,Y)}],
> $$
> and instantiates the critic at the representation level:
> $$
> T(G^\*,Y)=\mathrm{sim}(\mathbf{h}_{G^\*},\mathbf{h}_Y)+\log(N-1),
> $$
> from which the InfoNCE form follows via a standard DV relaxation. This derivation requires only that positive and negative samples follow the joint distribution $p(G^\*,Y)$ and the product of marginals $p(G^\*)p(Y)$, respectively. These conditions hold uniformly for node/graph classification, graph regression, and edge-level prediction. Thus, unlike RegExplainer, TAME provides a theoretically grounded and task-agnostic GIB objective that is valid for both discrete and continuous prediction tasks.
>
> **W4:Can TAME train a unified state-of-the-art explainer with shared weights for all tasks? If not, the term "TASK-AGNOSTIC FRAMEWORK" is somewhat confusing to me, and it is recommended to use "TASK-AGNOSTIC Objective" instead.**
>
> Response to W4:
>
> We thank the reviewer for this thoughtful comment. We agree that TAME cannot serve as a unified explainer with shared weights across all tasks.
>
> TAME provides two main contributions: a task-agnostic GIB objective and a task-agnostic structural mixup strategy. Together, these components form a unified explanation framework that mitigates distribution shift and enables precise explanations across different tasks. For this reason, we note that the term “task-agnostic objective” does not fully reflect the scope of TAME’s contributions, and we therefore adopt the term “task-agnostic framework” in the manuscript. We will further clarify this point in the revised version.
>
> We also acknowledge that training a fully unified explainer with shared parameters for all tasks is a challenging open problem, closely connected to ongoing efforts in graph foundation/world models. We view this as a promising direction and hope to explore all-in-one explanation frameworks in future work.

---

> ### Author Response · Authors · 2025-11-23
> **Response to Weakness Summary**
>
> **Weakness Summary: In summary, TAME's claim of being the first to propose a contrastive learning-based, task-agnostic GIB objective for GNN explanation is somewhat overstatement for me. The main contributions of TAME are twofold: first, it extends the experimental boundaries of the GIB objective (rather than proposing or optimizing the GIB objective itself); second, it replaces the soft-mask-based mixup strategy with a structural mixup strategy based on graph pooling, though the verification of this design is insufficient.**
>
> Response to Weakness Summary:
> We thank the reviewer for the thoughtful summary and would like to clarify the conceptual and technical contributions of TAME.
>
> **1\) We would like to clarify that TAME does not merely extend the experimental scope of GIB, but introduces a task-agnostic formulation of the GIB objective along with a principled derivation pathway.**
>
> First, TAME provides a unified, DV-based derivation of the mutual-information objective for GNN explanation. Unlike prior work, which formulates the objective through task-specific intermediate variables, TAME directly applies the Donsker–Varadhan representation to $I(G^*;Y)$ and instantiates the critic at the representation level, leading to an InfoNCE-based lower bound that is valid for both discrete and continuous prediction targets. This yields a single, task-agnostic GIB objective that can be optimized uniformly across diverse prediction settings.
>
> Second, we empirically validate this unified objective across four tasks. The results show that TAME consistently achieves state-of-the-art results on node-classification explanations (Table 1) and link-prediction explanations (Table 2), in addition to graph-level classification and regression tasks presented in the response to W3.
>
> Thus, while TAME indeed broadens the experimental evaluation of GIB, its main contribution lies in establishing a theoretically grounded and practically effective task-agnostic GIB objective, rather than merely extending prior objectives to new settings.
>
>
>
> **Table 1. Node-classification explanation results.**
> | Method          | BA-Shapes        | BA-Community     | Tree-Grid       |
> |-----------------|----------------|----------------|----------------|
> | PGExplainer     | 0.902 ± 0.027  | 0.653 ± 0.111  | 0.205 ± 0.113  |
> | MixupExplainer  | 0.801 ± 0.145  | 0.654 ± 0.113  | 0.179 ± 0.054  |
> | TAME            | **0.991 ± 0.001**  | **0.906 ± 0.154**  | **0.678 ± 0.039**  |
>
> **Table 2. Link-prediction explanation results.**
> | Method             | BA-Shapes ($Fid_{\alpha_1,+} \uparrow$)     | BA-Shapes ($Fid_{\alpha_2,-} \downarrow$)     | BA-Community ($Fid_{\alpha_1,+} \uparrow$) | BA-Community ($Fid_{\alpha_2,-} \downarrow$) | Tree-Grid ($Fid_{\alpha_1,+} \uparrow$)    | Tree-Grid ($Fid_{\alpha_2,-} \downarrow$)    |
> |--------------------|-----------------------|------------------------|----------------------|-----------------------|----------------------|-----------------------|
> | PGExplainer_link   | 0.201 ± 0.197         | 0.285 ± 0.177          | 0.132 ± 0.135        | 0.229 ± 0.101         | 0.112 ± 0.048        | 0.178 ± 0.053         |
> | TAME_link      | **0.225 ± 0.139**     | **0.044 ± 0.077**      | **0.221 ± 0.101**    | **0.016 ± 0.038**     | **0.166 ± 0.057**    | **0.021 ± 0.028**     |
>
>
>
> **2) Regarding the reviewer's concern on the structural mixup strategy, we clarify that TAME's structural mixup mechanism effectively generates non-redundant, in-distribution mixup graphs, and we supplement this with additional comparative experiments to demonstrate its advantages.**
>
> TAME performs structural mixup through graph pooling, which explicitly identifies and extracts the explanation and environment subgraphs before mixing. These subgraphs are then fused at the structural level, producing clean, non-redundant mixup graphs without relying on soft masks. Then a MLP is subsequently used to ensure consistency with prior explanations. This design produces mixup graphs that better preserve the topology of the original graph distribution, leading to more reliable optimization of the GIB objective.
>
> We provide empirical evaluations on both regression and classification tasks (in the tables included in our response to W2). The results show that, compared with soft-mask–based mixup methods, the mixup graphs generated by TAME better align with the original graph distribution and mitigate distribution shift more effectively.
>
>
> **In summary, TAME makes two complementary contributions: (i) a theoretically grounded, task-agnostic GIB objective derived from the DV representation and applicable to both discrete and continuous prediction tasks; and (ii) a structural mixup mechanism that produces in-distribution, non-redundant mixup graphs and significantly improves the optimization of the GIB objective. These components jointly distinguish TAME from prior GIB-based explainers and enable its strong performance across diverse explanation tasks.**

---

> > ### Comment · Reviewer_N8oL · 2025-11-24
> >
> > I appreciate the authors' response, which effectively addresses my concerns regarding W3 (OOD) and W4. That said, I still have unresolved concerns about W1 and W2, and I echo the points raised by Reviewer 1m9m.
> >
> > - W1: The theoretical contribution related to GIB remains limited. While TAME offers an alternative reasoning pathway to explain RegExplainer’s objective function, the final objective function retains an identical form. As a result, much of the claimed novelty surrounding GIB comes across as incremental rather than transformative.
> >
> > - W2: Structural mixup is a core component introduced by TAME, yet the accompanying analysis is inadequate. Furthermore, I previously highlighted several obvious structural flaws in structural mixup within my initial W2 feedback. Even if these flaws cannot be fully resolved, the paper ought to discuss them and provide thorough analysis. This would enable subsequent researchers to fully grasp the nuances of the work. Regrettably, the original manuscript does not address any of these issues. For GNNs, properties like rotation invariance, translation invariance, and robust handling of variable graph sizes are foundational. When manipulating graph structures, neglecting these properties can readily introduce shortcut features, which in turn lead to artificially inflated performance metrics. Given this, addressing these considerations in the paper is critical.
> >
> >
> > One additional suggestion: The authors should clearly delineate their own contributions from those of prior work, rather than conflating them. Failure to do so will hinder the understanding of readers who are already familiar with the relevant literature in this field.​

---

> > > ### Author Response · Authors · 2025-11-28
> > > **Response to Latest Comment W1**
> > >
> > > **I appreciate the authors' response, which effectively addresses my concerns regarding W3 (OOD) and W4. That said, I still have unresolved concerns about W1 and W2, and I echo the points raised by Reviewer 1m9m.**
> > >
> > > We sincerely thank the reviewer for the follow-up response and for acknowledging our efforts in addressing the concerns raised in W3 and W4. We appreciate the reviewer’s careful reading of our rebuttal and the continued engagement with the paper.
> > >
> > > **W1.The theoretical contribution related to GIB remains limited. While TAME offers an alternative reasoning pathway to explain RegExplainer’s objective function, the final objective function retains an identical form. As a result, much of the claimed novelty surrounding GIB comes across as incremental rather than transformative.**
> > >
> > > Response to Latest Comment W1：
> > >
> > > We thank the reviewer for their continued engagement. We are pleased to have the opportunity to further clarify the distinctions between the GIB formulation proposed in TAME and prior GIB-based methods, and to emphasize that our contributions encompass both a **task-agnostic GIB objective and task-agnostic structural mixup method**, forming a unified and highly generalizable explainer adaptable to diverse tasks.
> > >
> > > Regarding OOD issue in the GIB objective $\arg\min_{G^\*} I(G, G^\*) - \alpha I(Y, G^\*)$, current soft-mask-based methods such as MixupExplainer and ProxyExplainer still produce mixup graphs that suffer from distributional shift problems, as explained in the introduction and illustrated in Fig1. Therefore, our proposed structural mixup framework, which employs hard perturbations, generates mixup graphs that are closer to the original graph distribution compared to soft-mask-based methods. This results in more effective mitigation of distribution shift and consistently enhances the quality of the identified explanatory subgraphs. These advantages are substantiated by our OOD experiments presented in response to W2.
> > >
> > > Additionally, while existing methods adhere to using the traditional cross-entropy loss $\text{CE}(Y, f(G^{(mix)}))$ to approximate $I(Y, G^\*)$ for an optimizable objective. RegExplainer, based on the continuous label assumption, proves that an InfoNCE-style loss is a lower bound for the $I(Y^\*, Y)$. Thus, existing methods' estimation of $I(Y, G^\*)$ **remains task-specific**. In contrast, TAME directly instantiates the Donsker-Varadhan representation with a critic function at the representation level to estimate the $I(Y, G^\*)$ in a **task-agnostic manner**, thus resulting in a unified GIB objective across diverse tasks. This proposed **task-agnostic** GIB objective, combined with our structural mixup framework, forms a **task-agnostic** and highly generalizable explainer adaptable to diverse tasks.

---

> > > ### Author Response · Authors · 2025-11-28
> > > **Response to Latest Comment W2**
> > >
> > > **W2.Structural mixup is a core component introduced by TAME, yet the accompanying analysis is inadequate. Furthermore, I previously highlighted several obvious structural flaws in structural mixup within my initial W2 feedback. Even if these flaws cannot be fully resolved, the paper ought to discuss them and provide thorough analysis. This would enable subsequent researchers to fully grasp the nuances of the work. Regrettably, the original manuscript does not address any of these issues. For GNNs, properties like rotation invariance, translation invariance, and robust handling of variable graph sizes are foundational. When manipulating graph structures, neglecting these properties can readily introduce shortcut features, which in turn lead to artificially inflated performance metrics. Given this, addressing these considerations in the paper is critical.
> > > One additional suggestion: The authors should clearly delineate their own contributions from those of prior work, rather than conflating them. Failure to do so will hinder the understanding of readers who are already familiar with the relevant literature in this field.**
> > >
> > > Response to Latest Comment W2:
> > >
> > > We thank the reviewer for highlighting the limitations of structural mixup. In the revised manuscript, we have added detailed discussions and new experiments in Appendices H.5.3 and H.5.4 regarding graph invariance and pooling-size considerations. We sincerely appreciate the reviewer’s suggestions, which helped us identify missing analyses and further clarify the behavior of structural mixup.
> > >
> > > **1) Graph invariance**
> > >
> > > We thank the reviewer again for pointing out the importance of permutation invariance. We acknowledge that the original submission lacked discussion on this aspect. Following the reviewer’s W2 suggestion, we now include quantitative AUC comparisons and visual analyses before and after shuffling node indices in Appendix H.5.3.
> > >
> > > Concretely, prior to performing structural mixup, we randomly shuffle the node indices of the pooled explanatory subgraph to avoid unintended reliance on index ordering. As shown in Tables 1 and 2, across four graph-classification and four graph-regression datasets, the AUC results of the shuffled and original versions are nearly identical. This indicates that TAME does not exploit spurious shortcut features tied to node index order. A possible reason is that node indices are implicitly determined by ordering learnable node values, which change throughout training and therefore prevent the model from associating fixed index positions with shortcut patterns.
> > >
> > > **Table 1. Graph-classification AUC results.**
> > > | Method  | Ba-HouseGrid        | BaHouse-AndGrid     | Benzene           | Fluorid-Carbony     |
> > > |---------|----------------------|----------------------|--------------------|----------------------|
> > > | shuffle | 0.966 ± 0.025        | 0.979 ± 0.003        | 0.861 ± 0.001      | 0.795 ± 0.027        |
> > > | ori     | 0.965 ± 0.026        | 0.979 ± 0.004        | 0.861 ± 0.004      | 0.807 ± 0.011        |
> > >
> > >
> > > **Table 2. Graph-regression AUC results.**
> > > | Method  | BA-Motif-Volume      | House-Grid-Volume   | BA-Motif-Counting  | Crippen             |
> > > |---------|------------------------|----------------------|----------------------|----------------------|
> > > | shuffle | 0.995 ± 0.002          | 0.958 ± 0.006        | 0.945 ± 0.003        | 0.554 ± 0.028        |
> > > | ori     | 0.997 ± 0.001          | 0.966 ± 0.003        | 0.946 ± 0.004        | 0.565 ± 0.038        |
> > >
> > > In addition, Appendix H.5.3 provides a case study comparing TAME before and after node-index shuffling. Using randomly selected examples from the BA-HouseGrid (classification) and BA-Motif-Volume (regression) datasets, we visualize the target graph, the mixup process, and the resulting explanations under both settings. The visualizations show that TAME consistently identifies and exchanges the correct motif structures, and produces accurate explanations regardless of whether node indices are shuffled, further confirming that the explainer does not rely on index-specific artifacts.

---

> > > ### Author Response · Authors · 2025-11-28
> > > **Additional Response to Latest Comment W2**
> > >
> > > **W2.Structural mixup is a core component introduced by TAME, yet the accompanying analysis is inadequate. Furthermore, I previously highlighted several obvious structural flaws in structural mixup within my initial W2 feedback. Even if these flaws cannot be fully resolved, the paper ought to discuss them and provide thorough analysis. This would enable subsequent researchers to fully grasp the nuances of the work. Regrettably, the original manuscript does not address any of these issues. For GNNs, properties like rotation invariance, translation invariance, and robust handling of variable graph sizes are foundational. When manipulating graph structures, neglecting these properties can readily introduce shortcut features, which in turn lead to artificially inflated performance metrics. Given this, addressing these considerations in the paper is critical.
> > > One additional suggestion: The authors should clearly delineate their own contributions from those of prior work, rather than conflating them. Failure to do so will hinder the understanding of readers who are already familiar with the relevant literature in this field.**
> > >
> > > Additional Response to Latest Comment W2:
> > >
> > > **2) Pooling Size in Structural Mixup**
> > >
> > > We thank the reviewer for pointing out the limitation that structural mixup requires explanatory subgraphs of matching size. We provide further clarification regarding the feasibility of this design. In TAME, the target graph is mixed with both a near graph and a far graph sampled via embedding-similarity–based retrieval. Near graphs typically share similar labels and motif types with the target graph, and therefore tend to produce explanatory subgraphs of comparable size, allowing structural mixup to operate reliably.
> > >
> > > For far graphs, the explanatory subgraph sizes may differ. In practice, we apply a slightly larger top-k ratio during pooling so that the extracted subgraph forms a superset that reliably includes the relevant explanatory structure. Any redundant components are further refined by the MLP-generated mask. Table 3 reports the AUC results on datasets with varying motif sizes, showing that TAME consistently achieves strong performance on both classification and regression benchmarks.
> > >
> > > **Table 3. AUC results on datasets with varying motif sizes.**
> > >
> > > | Method     | Ba-HouseGrid      | BaHouse-AndGrid    | BA-Motif-Counting | House-OrGrid-Volume |
> > > |------------|--------------------|---------------------|--------------------|----------------------|
> > > | **TAME**   | 0.965 ± 0.026      | 0.979 ± 0.004       | 0.946 ± 0.004      | 0.941 ± 0.012        |
> > > | Runner-up  | 0.848 ± 0.039      | 0.806 ± 0.069       | 0.946 ± 0.095      | 0.741 ± 0.107        |
> > >
> > >
> > > We further conduct a case study in Appendix H.5.4 on the Ba-HouseGrid dataset, visualizing the structural mixup process when the target and neighbor graphs have differing explanation sizes. The visualizations demonstrate that TAME accurately identifies and exchanges the key motif regions for both near and far graphs, ultimately yielding faithful explanations.
> > >
> > > These analyses summarize our extended discussion on graph invariance and pooling-size considerations in structural mixup. We sincerely thank the reviewer for highlighting these missing discussions of structural mixup, which helped us improve the completeness of the manuscript.
> > >
> > > We warmly welcome any additional feedback or comments from the reviewer.

---

### Official Review · Reviewer_zcgv · 2025-10-29

**Soundness:** 2
**Presentation:** 3
**Contribution:** 3
**Rating:** 6
**Confidence:** 3

**Summary:**

This paper proposes a Task-Agnostic structural Mixup Explanation (TAME) framework for GNN explanation, which addresses both the limitations of task-specific methods and the distribution shift caused by subgraph extraction.

**Strengths:**

1. This is the first work that integrates contrastive learning into the Graph Information Bottleneck (GIB) framework.

2. This paper designs a novel mixup strategy that structurally replaces explanatory subgraphs, which generates in-distribution explanations
through hard structural perturbations.

3. Experiments on both synthetic and real-world benchmarks demonstrate that TAME outperforms existing methods across diverse tasks, achieving up to a 30.1% improvement in AUC.

**Weaknesses:**

1. This paper focuses on post-hoc explanations, but ignores the scalability of dynamic graphs or large-scale GNNs, and does not mention the computational overhead.

2. The definition of Problem 1 (Post-hoc Instance-level GNN Explanatory Method) lacks a reference; is it widely acknowledged by other researchers, or is there any new novelty in this definition?

3. TAME is primarily designed for graph- and node-level explanations with well-defined subgraph dependencies, but not for edge-level tasks.

4. "Regarding hyperparameters, we apply grid search to determine the loss weight βand set the top-r ratios according to the ground-truth explanations." However, the detailed processes are not provided.  Are there any theoretical promises for selecting such hyperparameters?

5. "TAME introduces a theoretically grounded objective based on GIB." However, the theoretical contributions are weak, it lacks proof (such as the derivation of the lower bound of mutual information).

**Questions:**

1. The definition of Problem 1 (Post-hoc Instance-level GNN Explanatory Method) lacks a reference; is it widely acknowledged by other researchers, or is there any new novelty in this definition?

2. How to optimally choose hyperparameters is not well illustrated. Are there any theoretical promises for selecting such hyperparameters?

---

> ### Author Response · Authors · 2025-11-23
> **Response to W1,W2,Q1**
>
> **W1:This paper focuses on post-hoc explanations, but ignores the scalability of dynamic graphs or large-scale GNNs, and does not mention the computational overhead.**
>
> Response to W1:
>
> We thank the reviewer for raising this concern and are pleased  to make clearly clarify and explain. Our method is a foundational framework and effectively addresses the out-of-distribution (OOD) problem in the post-hoc instance-level explanation methods on static graphs.
>
> For dynamic graph, due to the temporal characteristic, its explanatory subgraph may need to satisfy additional temporal constraints (e.g., continuity), and our framework can be effectively extended to support dynamic graph tasks with appropriate constraints. Regarding large-scale graphs, our method obtains subgraphs and removes redundant structures through the graph pooling process. In contrast, existing soft mask methods (e.g., GNNExplainer, PGExplainer), as they need to output edge weights over the entire graph to identify explanatory subgraphs and the included irrelevant subgraphs would affect the generated explanation quality. Large-scale graphs contain more irrelevant substructures, and our method is naturally more effective in such scenarios. Overall, dynamic and large-scale graph explanation tasks are not covered in our current discussion, but we are interested in pursuing these in future work.
>
> Besides, we provide a detailed Computational Complexity Analysis in Appendix D (Algorithms) part. Specifically, our method involves an L-layer graph pooling process and the time complexity is $O\left(L \times (|\mathcal{E}|d + nd)\right)$, where d is the node feature dimension, $\mathcal{E}$ denotes the edge set, and n is the number of nodes. Structural replacement and generating edge weights for the mixup graph via an MLP are also two parts of our method. Since the sizes of the replaced subgraphs and the edges of the mixup graph are comparable to those of the original graph, the term $O\left(L \times (|\mathcal{E}|d)\right)$ dominates the overall complexity of our method. This complexity is linear in the number of edges, which is acceptable for common real-world sparse graphs.
>
> **W2:The definition of Problem 1 (Post-hoc Instance-level GNN Explanatory Method) lacks a reference; is it widely acknowledged by other researchers, or is there any new novelty in this definition? & Q1:The definition of Problem 1 (Post-hoc Instance-level GNN Explanatory Method) lacks a reference; is it widely acknowledged by other researchers, or is there any new novelty in this definition?**
>
> Response to W2&Q1:
>
> We sincerely thank the reviewer for the question raised regarding the definition of Problem 1 and provide the following detailed clarification. Indeed, the task of post-hoc instance-level explanation for GNNs has been widely studied, with some existing methods including GNNExplainer[1], PGExplainer[2] and more recent methods such as GraphMask[3], MixupExplainer[4] and GIBE[5].
>
> However, some existing methods focus on a single task and suffer from out-of-distribution (OOD) problem when identifying explanatory subgraphs, which can lead to unreliable explanations. Our work explicitly addresses these challenges to achieve more robust and faithful subgraph identification.
>
> TAME introduces a novel structural mixup strategy based on graph pooling to generate mixup graphs that remain aligned with the original graph distribution, effectively mitigating OOD problems. In addition, TAME introduces the first theoretically grounded task-agnostic GIB objective which, together with the in-distribution mixup graphs generated by the structural mixup strategy, can be optimized without distribution bias, enabling more accurate and faithful explanations.
>
> **References**
>
> [1]Ying, Zhitao, et al. "Gnnexplainer: Generating explanations for graph neural networks." Advances in neural information processing systems 32 (2019).
>
> [2]Luo, Dongsheng, et al. "Parameterized explainer for graph neural network." Advances in neural information processing systems 33 (2020): 19620-19631.
>
> [3]Schlichtkrull, Michael Sejr, Nicola De Cao, and Ivan Titov. "Interpreting Graph Neural Networks for NLP With Differentiable Edge Masking." International Conference on Learning Representations.
>
> [4]Zhang, Jiaxing, Dongsheng Luo, and Hua Wei. "Mixupexplainer: Generalizing explanations for graph neural networks with data augmentation." Proceedings of the 29th ACM SIGKDD conference on knowledge discovery and data mining. 2023.
>
> [5]Fang, Junfeng, et al. "On regularization for explaining graph neural networks: An information theory perspective." IEEE Transactions on Knowledge and Data Engineering (2024).

---

> ### Author Response · Authors · 2025-11-23
> **Response to W3,W4,Q2**
>
> **W3:TAME is primarily designed for graph- and node-level explanations with well-defined subgraph dependencies, but not for edge-level tasks.**
>
> Response to W3:
>
> We thank the reviewer for the valuable point. TAME introduces a task-agnostic GIB objective that enables explanation learning through representation-level contrastive learning, and is therefore applicable to a broad range of tasks, including edge-level tasks. Specifically, for edge-level tasks, we follow SEAL [1] to construct local subgraphs around each target edge. TAME can then apply the same perturbation-based procedure as in node-level tasks, extracting critical substructures from the local subgraphs as edge-level explanations.
>
> We use SEAL [1] as the base model to be explained and implement edge-level versions of both TAME and PGExplainer for link prediction. Experiments are conducted on BA-Shapes, BA-Community, and Tree-Grids. Since ground-truth explanations are unavailable in these datasets, we adopt FID+ and FID– [2] as evaluation metrics. The results show that TAME\_link consistently outperforms PGExplainer\_link, indicating that the structural mixup mechanism in TAME effectively mitigates distribution shift and produces more precise explanations.
>
>
> | Method             | BA-Shapes ($Fid_{\alpha_1,+} \uparrow$)     | BA-Shapes ($Fid_{\alpha_2,-} \downarrow$)     | BA-Community ($Fid_{\alpha_1,+} \uparrow$) | BA-Community ($Fid_{\alpha_2,-} \downarrow$) | Tree-Grid ($Fid_{\alpha_1,+} \uparrow$)    | Tree-Grid ($Fid_{\alpha_2,-} \downarrow$)    |
> |--------------------|-----------------------|------------------------|----------------------|-----------------------|----------------------|-----------------------|
> | PGExplainer_link   | 0.201 ± 0.197         | 0.285 ± 0.177          | 0.132 ± 0.135        | 0.229 ± 0.101         | 0.112 ± 0.048        | 0.178 ± 0.053         |
> | TAME_link      | **0.225 ± 0.139**     | **0.044 ± 0.077**      | **0.221 ± 0.101**    | **0.016 ± 0.038**     | **0.166 ± 0.057**    | **0.021 ± 0.028**     |
>
> [1]Zhang, Muhan, and Yixin Chen. "Link prediction based on graph neural networks." Advances in neural information processing systems 31 (2018).
>
> [2]Zheng, Xu, et al. "Towards robust fidelity for evaluating explainability of graph neural networks." arXiv preprint arXiv:2310.01820 (2023).
>
> **W4:"Regarding hyperparameters, we apply grid search to determine the loss weight βand set the top-r ratios according to the ground-truth explanations." However, the detailed processes are not provided. Are there any theoretical promises for selecting such hyperparameters? & Q2:How to optimally choose hyperparameters is not well illustrated. Are there any theoretical promises for selecting such hyperparameters?**
>
> Response to W4&Q2:
>
> We appreciate the reviewer’s careful attention to the hyperparameter configuration in our method. In the experiments, we follow prior related work[1] and employ a grid search to identify a suitable $\beta$ for BCE loss, use an empirical estimate of the explanatory subgraph size to set the parameter r.
>
> Moreover, the hyperparameter sensitivity study in Appendix H.2 shows that our method is robust to $\beta$ on both graph classification and regression datasets, consistently outperforming the best baseline method across a wide range of values. Therefore, we select a relatively performing parameter configuration from this range for our reported results.
>
> [1]Zhang, Jiaxing, Dongsheng Luo, and Hua Wei. "Mixupexplainer: Generalizing explanations for graph neural networks with data augmentation." Proceedings of the 29th ACM SIGKDD conference on knowledge discovery and data mining. 2023.

---

> ### Author Response · Authors · 2025-11-23
> **Response to W5**
>
> **W5:"TAME introduces a theoretically grounded objective based on GIB." However, the theoretical contributions are weak, it lacks proof (such as the derivation of the lower bound of mutual information).**
>
> Response to W5:
>
> We greatly appreciate the reviewer's interest in and concerns regarding the theoretically grounded of our method based on GIB.
> In the Section 3.1 in our Method, we provide theoretical guarantees demonstrating how InfoNCE can serve as a lower bound for mutual information.
>
> First, we apply the Donsker-Varadhan (DV) variational representation[1] to approximate the GIB objective I(G∗;Y). In Property 1 and Appendix F, we provide a rigorous justification and detailed derivation showing that DV(G∗;Y) is a valid lower bound of I(G∗;Y). Furthermore, to make this bound tractable and trainable, we instantiate the critic function T in Equation (3) in a representation-based manner. Under this instantiation, we introduce that the InfoNCE serves as a further lower bound of DV(G∗;Y) in Property 2 and Appendix G. Ultimately, we can optimize the GIB objective in a task-agnostic manner as presented in Equation (7).
>
> For clarity, we provide a simplified version of the derivation here, and refer the reader to Appendices F and G for the complete details.
> We begin with the mutual information between the explanatory subgraph $G^*$ and the label $Y$:
>
> $$
> I(G^*;Y)= \text{KL}(p(G^\*,Y) | p(G^\*)p(Y)).
> $$
>
> Applying the Donsker--Varadhan variational representation of KL divergence:
>
> $$I(G^*;Y)\\ge\\mathbb{E}_{p(G^\*,Y)}[T(G^\*,Y)]{-}\\log\\mathbb{E}\_{p(G^\*)p(Y)}[\\exp(T(G^\*,Y))].$$
>
> We instantiate the critic as the similarity between graph embeddings:
> $$
> T(G^\*,Y)= \\text{sim}(f(G^\*),f(G_{Y})) + \\log(N-1),
> $$
>
> where each batch contains one positive pair and $N-1$ negatives. The positive pair follows the joint distribution and contributes to the first term, while the negatives are drawn from the product of marginals and contribute to the second term. Applying Jensen's inequality to approximate the expectation over negative samples and substituting the critic yields:
>
> $$
> I(G^*;Y)
> \\ge
> \\mathbb{E}\\left[
> \\log
> \\frac{
> \\exp(\\text{sim}^{+})
> }{
> \\frac{1}{N-1}\\sum_{j=1}^{N-1}\\exp(\\text{sim}^{-}_j)
> }
> \\right].
> $$
>
> Adding the positive term into the denominator produces a valid but looser bound:
>
> $$
> I(G^\*;Y)
> \\ge
>  \\log(N-1)
> +
> \\mathbb{E}\\left[
> \\log
> \\frac{
> \\exp(\\text{sim}^{+})
> }{
> \\sum\_{j=0}^{N-1}\\exp(\\text{sim}\_j)
> }
> \\right]
> \=
> \\log(N-1)+\\mathbb{E}[\\ell_{\\text{InfoNCE}}].
> $$
>
> [1]Poole, Ben, et al. "On variational bounds of mutual information." International conference on machine learning. PMLR, 2019.

---

> > ### Comment · Reviewer_zcgv · 2025-11-26
> >
> > I appreciate the authors' response. The authors have addressed my concerns regarding W1 and W3. However, there are some still unresolved concerns about W2, W4 and W5. Thus, I will keep my score.

---

> > > ### Author Response · Authors · 2025-12-01
> > > **Response to Latest Comment W2**
> > >
> > > **I appreciate the authors' response. The authors have addressed my concerns regarding W1 and W3. However, there are some still unresolved concerns about W2, W4 and W5. Thus, I will keep my score.**
> > >
> > > We sincerely thank the reviewer for the follow-up response and for acknowledging the clarifications provided for W1 and W3. We appreciate the reviewer’s continued engagement with our work. We understand that concerns remain regarding W2, W4, and W5, and we will provide additional clarification and analysis below.
> > >
> > > **W2.The definition of Problem 1 (Post-hoc Instance-level GNN Explanatory Method) lacks a reference; is it widely acknowledged by other researchers, or is there any new novelty in this definition?**
> > >
> > > For W2, Problem 1 (Post-hoc Instance-level GNN Explanation) has been studied in prior work, as we summarized with a subset of representative works in the Response to W2 & Q1. However, our work focuses on the OOD issue inherent in such post-hoc instance-level explanation methods. To address this issue, TAME introduces **a structural mixup framework based on hard perturbations**, which generates **mixup graphs** that are closer to the original graph distribution compared to soft-mask-based methods, thereby enhancing the quality of the identified explanatory subgraphs. Furthermore, by integrating the structural mixup framework with a theoretically grounded, **task-agnostic GIB objective** derived from the Donsker–Varadhan (DV) representation, TAME realizes a highly generalizable explainer that is adaptable to diverse tasks. Therefore, following exsiting work[1][2], we also provide the definition of Post-hoc Instance-level GNN Explanation Method as Problem 1 in our paper.
> > >
> > > [1]Schlichtkrull, Michael Sejr, Nicola De Cao, and Ivan Titov. "Interpreting Graph Neural Networks for NLP With Differentiable Edge Masking." International Conference on Learning Representations.
> > >
> > > [2]Fang, Junfeng, et al. "On regularization for explaining graph neural networks: An information theory perspective." IEEE Transactions on Knowledge and Data Engineering (2024).

---

> > > ### Author Response · Authors · 2025-12-01
> > > **Response to Latest Comment W4**
> > >
> > > **W4."Regarding hyperparameters, we apply grid search to determine the loss weight βand set the top-r ratios according to the ground-truth explanations." However, the detailed processes are not provided. Are there any theoretical promises for selecting such hyperparameters?**
> > >
> > > For W4, in the experiments, we perform the grid search  to identify the optimal value of $\beta$ for the BCE loss, following the practice established in  existing baselines[1][2]. Additionally, the hyperparameter sensitivity study on $\beta$ in Appendix H.3 shows that **our method is robust across both graph classification and regression datasets**, consistently outperforming the best baseline method across a wide range of values. Therefore, we select a relatively performing parameter configuration from this range as our reported results.
> > >
> > > Regarding the explanatory subgraph size parameter $r$, we provide an additional discussion. Since near graphs and the target graph share similar labels and motifs, their explanatory subgraphs are of comparable size, enabling reliable structural mixup. For far graphs, where subgraph sizes may differ, we use **a slightly larger top-k ratio during pooling** to extract a superset containing the relevant explanatory subgraph. **Any redundancy is then refined by the MLP-generated mask.** As demonstrated in Table 1 and the case study in Appendix H.5.4, TAME **precisely identifies and exchanges the key motif regions for both near and far graphs**, thus yielding faithful explanations.
> > > **Table 1. AUC results on datasets with varying motif sizes.**
> > >
> > > | Method     | Ba-HouseGrid      | BaHouse-AndGrid    | BA-Motif-Counting | House-OrGrid-Volume |
> > > |------------|--------------------|---------------------|--------------------|----------------------|
> > > | **TAME**   | 0.965 ± 0.026      | 0.979 ± 0.004       | 0.946 ± 0.004      | 0.941 ± 0.012        |
> > > | Runner-up  | 0.848 ± 0.039      | 0.806 ± 0.069       | 0.946 ± 0.095      | 0.741 ± 0.107        |
> > >
> > > [1]Luo, Dongsheng, et al. "Parameterized explainer for graph neural network." Advances in neural information processing systems 33 (2020): 19620-19631.
> > >
> > > [2]Zhang, Jiaxing, Dongsheng Luo, and Hua Wei. "Mixupexplainer: Generalizing explanations for graph neural networks with data augmentation." Proceedings of the 29th ACM SIGKDD conference on knowledge discovery and data mining. 2023.

---

> > > ### Author Response · Authors · 2025-12-01
> > > **Response to Latest Comment W5**
> > >
> > > **W5."TAME introduces a theoretically grounded objective based on GIB." However, the theoretical contributions are weak, it lacks proof (such as the derivation of the lower bound of mutual information).**
> > >
> > > For W5, as established in Section 3.1 and Appendices F & G, the InfoNCE objective serves as a tractable **lower bound** for mutual information  $I(G^\*;Y)$, providing the theoretical foundation for our **task-agnostic GIB objective**. Specifically, we apply the Donsker-Varadhan (DV) representation to approximate the GIB objective $I(G^\*;Y)$, and rigorously justify that $\mathrm{DV}(G^\*;Y)$ is a valid lower bound of $I(G^\*;Y)$ in Property 1 and Appendix F. To render the bound tractable, we instantiate the critic $T$ in Eq.\~(3) using representations, under which InfoNCE serves as a further lower bound to $\mathrm{DV}(G^\*;Y)$, as shown in Property 2 and Appendix G. This enables the task-agnostic optimization of the GIB objective in Eq.\~(7). A simplified version is provided in Response to W5, while the complete derivation is detailed in Section 3.1 and Appendices F & G.
> > >
> > > In summary, TAME makes two complementary contributions: (i) **a theoretically grounded, task-agnostic GIB objective** derived from the DV representation and applicable to both discrete and continuous prediction tasks; and (ii) **a structural mixup mechanism that produces in-distribution, non-redundant mixup graphs** and significantly improves the optimization of the GIB objective. These components jointly distinguish TAME from prior GIB-based explainers and enable its strong performance across diverse explanation tasks.

---

### Official Review · Reviewer_1m9m · 2025-10-29

**Soundness:** 2
**Presentation:** 2
**Contribution:** 1
**Rating:** 2
**Confidence:** 4

**Summary:**

This paper proposes TAME, a unified task-agnostic framework for GNN explainability for both graph classification and regression tasks, and also addresses distribution shift problems of explanatory graphs.

**Strengths:**

S1. A task-agnostic explainer on graphs presents an interesting and efficient approach for generating graph explanations across diverse problem settings, including regression tasks.

S2. Additionally, authors attempt to provide robust explanations by addressing OOD problems of explanatory graphs.

**Weaknesses:**

W1. The proposed TAME is not the first to propose a task-agnostic explainer model. TAGE (NeurIPS 2022) [1] also introduced task-agnostic explainers based on conditioned contrastive learning, demonstrating performance on both graph classification and node classification tasks, with potential extensibility to regression tasks as well.

W2. The effectiveness of structural mixup in Section 3.2 in addressing distribution shift may be limited in certain cases. Due to the inductive biases learned by GNNs, nodes with similar attributes or global structural roles can have similar embeddings even when they are not neighbors. In such scenarios, the proposed structural mixup strategy may have restricted effectiveness.

W3. In Table 3, the explanation embeddings are shown to be much closer to the original graph distribution than to the ground truth, suggesting that the extracted subgraphs should differ from the ground truth. However, Figures 4 and 5 show that the explanations are nearly identical to the ground truth, which seems contradictory.

[1] Task-Agnostic Graph Explanations, Neurips 22 (https://arxiv.org/pdf/2202.08335)

**Questions:**

Q1. Please clarify the novelty of TAME compared to TAGE (NeurIPS 2022). What are the key technical differences and contributions that distinguish your approach? Additionally, include TAGE as a baseline in your experiments and provide a comparative performance analysis.

Q2. The out-of-distribution (OOD) experiments in Table 3 are limited to comparisons with ground truth only. Please demonstrate the effectiveness of your proposed method by comparing it not only with ground truth but also with other baseline methods, such as MixupExplainer and ProxyExplainer, that address distribution shift.

Please consider the weaknesses in conjunction with the questions.

---

> ### Author Response · Authors · 2025-11-23
> **Response to W1&Q1**
>
> **W1:The proposed TAME is not the first to propose a task-agnostic explainer model. TAGE (NeurIPS 2022)[1] also introduced task-agnostic explainers based on conditioned contrastive learning, demonstrating performance on both graph classification and node classification tasks, with potential extensibility to regression tasks as well. & Q1:Please clarify the novelty of TAME compared to TAGE (NeurIPS 2022). What are the key technical differences and contributions that distinguish your approach? Additionally, include TAGE as a baseline in your experiments and provide a comparative performance analysis.**
>
> Response to W1&Q1:
>
> We thank the reviewer for bringing this work to our attention. We acknowledge that TAGE is related to our study and provides a valuable contribution by proposing a task-agnostic explainer trained in an unsupervised manner. However, TAME differs from TAGE in several fundamental aspects.
>
> First, TAME derives a task-agnostic GIB objective through a rigorous theoretical formulation, ensuring task independence at the optimization level. In contrast, TAGE separates the explainer into embedding and downstream components and relies on gradient-based techniques for the downstream explainer, achieving task-agnostic behavior. Notably, GradExplainer was originally introduced as a baseline in GNNExplainer [1], and its explanation quality may be limited.
>
> Second, TAME effectively mitigates OOD issues in the GIB objective through its structural mixup strategy, whereas TAGE does not address OOD challenges in its InfoNCE objective .
>
> Third, the InfoNCE loss designs differ between TAME and TAGE: in TAME, positive pairs consist of the target graph’s mixup graph and its near graph, while negative pairs consist of the mixup graph and its far graph; in contrast, TAGE defines positive pairs as the target graph and its explanation graph, and negative pairs as the target graph and explanations of other graphs.
>
> Although both TAME and TAGE are described as task-agnostic, TAME introduces a theoretically grounded task-agnostic GIB objective and a novel structural mixup strategy to address OOD issues, which possess certain innovation and contribution. We appreciate the reviewer’s suggestion and have included a discussion of TAGE in the revised manuscript.
>
> **Table:** Edge-level explanation accuracy (**AUC-ROC**) on **graph classification** datasets.
> | Method       | BA-2motifs      | BA-HouseGrid    | SPMotif         | BA-HouseAndGrid | Alkane-Carbonyl | Fluorid-Carbonyl | Benzene         |
> | ------------ | --------------- | --------------- | --------------- | --------------- | --------------- | ---------------- | --------------- |
> | TAGExplainer | $0.676\pm0.148$ | $0.848\pm0.039$ | $0.531\pm0.026$ | $0.647\pm0.072$ | $0.842\pm0.059$ | $0.752\pm0.046$  | $0.760\pm0.067$ |
> | Our          | $0.971\pm0.021$ | $0.965\pm0.026$ | $0.748\pm0.123$ | $0.979\pm0.004$ | $0.944\pm0.020$ | $0.807\pm0.011$  | $0.861\pm0.004$ |
>
> **Table:** Edge-level explanation accuracy (**AUC-ROC**) on **graph regression** datasets.
> | Method        | BA-Motif-Volume     | House-Grid-Volume   | BA-Motif-Counting   | Triangles         | Crippen           | House-OrGrid-Volume |
> |---------------|---------------------|---------------------|---------------------|-------------------|-------------------|---------------------|
> | TAGExplainer  | $0.548\pm0.262$     | $0.856\pm0.105$     | $0.763\pm0.354$     | $0.610\pm0.163$   | $0.486\pm0.004$   | $0.698\pm0.167$     |
> | Our           | $0.997\pm0.001$     | $0.966\pm0.003$     | $0.946\pm0.004$     | $0.648\pm0.020$   | $0.565\pm0.038$   | $0.941\pm0.012$     |
>
> [1]Ying, Zhitao, et al. "Gnnexplainer: Generating explanations for graph neural networks." Advances in neural information processing systems 32 (2019).

---

> ### Author Response · Authors · 2025-11-23
> **Response to W2**
>
> **W2.The effectiveness of structural mixup in Section 3.2 in addressing distribution shift may be limited in certain cases. Due to the inductive biases learned by GNNs, nodes with similar attributes or global structural roles can have similar embeddings even when they are not neighbors. In such scenarios, the proposed structural mixup strategy may have restricted effectiveness.**
>
> Response to W2:
>
> We sincerely thank the reviewer for the thoughtful feedback. To our understanding, the concern is that the embedding-similarity-based sampling strategy may affect the explainer's performance.
>
> We would like to clarify that the embeddings used for sampling are obtained from the penultimate layer of the model being explained, which encode meaningful label-related information. Thus, sampling neighbors based on embedding similarity effectively approximates sampling based on label similarity. By pulling the target graph closer to label-similar graphs and pushing it away from label-dissimilar graphs, the contrastive objective can be correctly optimized to learn faithful explanations.
>
> Furthermore, TAME is designed to explain the behavior of the target model. Since the embeddings are generated by the model being explained, the similarity-based sampling reflects the model's own notion of near and far neighbors. Consequently, the explainer is trained in alignment with the model's decision, enabling it to accurately capture and interpret the model's behavior.
>
> If your concern differs from our understanding above, please let us know and we would greatly appreciate the opportunity to discuss it with you further.

---

> ### Author Response · Authors · 2025-11-23
> **Response to W3&Q2**
>
> **W3. In Table 3, the explanation embeddings are shown to be much closer to the original graph distribution than to the ground truth, suggesting that the extracted subgraphs should differ from the ground truth. However, Figures 4 and 5 show that the explanations are nearly identical to the ground truth, which seems contradictory. & Q2:The out-of-distribution (OOD) experiments in Table 3 are limited to comparisons with ground truth only. Please demonstrate the effectiveness of your proposed method by comparing it not only with ground truth but also with other baseline methods, such as MixupExplainer and ProxyExplainer, that address distribution shift.**
>
> Response to W3&Q2:
>
> We thank the reviewer for raising this important concern and are pleased to make clearly clarify and explain.
> In Table 3, the ground-truth explanatory subgraph refers to a subgraph derived by removing non-explanatory parts from the original graph. When this subgraph is fed into a GNN pre-trained on the original graph, the distribution of explanatory subgraph differs from that of the original graphs, leading to a shift in graph embeddings and thus an out-of-distribution (OOD) problem.
>
> In contrast, the mixup graph is constructed by mixing the explanatory subgraph with the non-explanatory part to approximate the distribution of the original graphs, thereby solving the OOD issue during prediction. Therefore, in Table 3, the embeddings of the mixup graphs are closer to those of the original graphs than the embeddings of the explanatory subgraphs.
>
> In summary, Table 3 demonstrates that our proposed mixup method better approximates the original graph distribution and effectively addresses the OOD problem. In Fig. 4, we visualize the mixup graphs generated by different methods. The visualizations clearly show that our method produces mixup graphs that more closely resemble the original graph compared to existing approaches and Fig. 5 visualizes the explanatory subgraphs extracted by each method. Together, the results in Table 3 and Figs. 4, 5 are consistent, our method better approximates the original graph distribution, thereby solving the OOD problem that arises during optimization and improving the quality of the identified explanatory subgraphs.
>
> Additionally, in the OOD experiments, we include MixupExplainer and ProxyExplainer as baselines for classification tasks and RegExplainer for regression tasks. By evaluating the similarity between the generated mixup graphs and the original graphs, the results show that TAME consistently produces mixup graphs that remain closest to the original graph distribution. Consequently, TAME avoids inducing OOD-related prediction deviations during training, effectively mitigating distribution shift and thereby enabling the learning of more accurate explanations.
>
> | Method                                               | Ba-HouseGrid | Ba-HouseAndGrid | Benzene   | Fluorid-Carbonyl |
> | ---------------------------------------------------- | ------------ | --------------- | --------- | ---------------- |
> | $\text{Cos}(h_G, h_{G^*}) \uparrow$                  | 0.688        | 0.613           | 0.821     | 0.918            |
> | $\text{Cos}(h_G, h_{G_{\text{mixup}}})$              | 0.781        | 0.966           | 0.907     | 0.895            |
> | $\text{Cos}(h_G, h_{G_{\text{proxy}}})$              | 0.802        | 0.970           | 0.565     | 0.723            |
> | $\text{Cos}(h_G, h_{G_{\text{our}}^{\text{(mix)}}})$ | **0.868**    | **0.971**       | **0.941** | **0.937**        |
> | $\text{Euc}(h_G, h_{G^*}) \downarrow$                | 1.044        | 1.436           | 0.953     | 0.684            |
> | $\text{Euc}(h_G, h_{G_{\text{mixup}}})$              | 0.973        | 0.477           | 0.771     | 0.782            |
> | $\text{Euc}(h_G, h_{G_{\text{proxy}}})$              | 0.706        | 0.4424          | 1.471     | 1.272            |
> | $\text{Euc}(h_G, h_{G_{\text{our}}^{(\text{mix})}})$ | **0.661**    | **0.425**           | **0.578**   | **0.604**            |
>
> | Method                                               | BA-Motif-Volume | BA-Motif-Counting | Crippen | Triangles |
> | ---------------------------------------------------- | --------------- | ----------------- | ------- | --------- |
> | $\text{Cos}(h_G, h_{G^*}) \uparrow$           | 0.696           | 0.663             | 0.869   | 0.897     |
> | $\text{Cos}(h_G, h_{G_{\text{reg}}})$ | 0.876           | 0.950             | 0.847   | 0.953     |
> | $\text{Cos}(h_G, h_{G_{\text{our}}^{\text{(mix)}}})$         | **0.953**           | **0.969**             | **0.936**   | **0.985**     |
> |$\text{Euc}(h_G, h_{G^*}) \uparrow$           | 1.138           | 1.196             | 0.653   | 0.389     |
> | $\text{Euc}(h_G, h_{G_{\text{reg}}})$ | 0.775           | 0.485             | 0.794   | 0.247     |
> | $\text{Euc}(h_G, h_{G_{\text{our}}^{(\text{mix})}})$         | **0.458**           | **0.400**             | **0.431**   | **0.143**     |

---

> > ### Comment · Reviewer_1m9m · 2025-11-24
> > **Response to Authors**
> >
> > I appreciate the authors’ efforts in conducting thorough experiments and demonstrating strong performance compared to the baselines. Apart from the strong performance, most of the claimed novelty appears incremental.
> >
> > TAME claims that it derives its objective from a theoretically well-defined GIB foundation, but it largely leverages prior literature. Since GSAT proposed a GIB-based objective for interpretable graph learning, it has already been extensively discussed in post-hoc settings in MixupExplainer and ProxyExplainer. Furthermore, I have not yet found any new insight or theoretical contribution regarding GIB.
> >
> > The original claim of being a task-agnostic explainer is also not novel. The authors mention negative sampling as a differentiating factor, but there is insufficient discussion on why the TAGE approach shows weak performance.
> >
> > The main difference from TAGE is the structural mix-up. As discussed in the rebuttal for W2 of Reviewer N8oL, the assumption from the authors is quite limited for generalized settings since semantic similarity does not guarantee similar graph sizes.
> >
> > Based on the experimental results in the tables, the method clearly shows strong performance, and aside from the novelty issue, most of my earlier concerns have been addressed. Therefore, I will update my score from 2 to 4.

---

> > > ### Author Response · Authors · 2025-11-28
> > > **Response to Latest Comment W1**
> > >
> > > **I appreciate the authors’ efforts in conducting thorough experiments and demonstrating strong performance compared to the baselines. Apart from the strong performance, most of the claimed novelty appears incremental.**
> > >
> > > We sincerely thank the reviewer for acknowledging the strengths of our work and for the improved score. We truly appreciate the reviewer’s balanced and constructive feedback, which has significantly helped us enhance the clarity and completeness of the manuscript.
> > >
> > > **W1.TAME claims that it derives its objective from a theoretically well-defined GIB foundation, but it largely leverages prior literature. Since GSAT proposed a GIB-based objective for interpretable graph learning, it has already been extensively discussed in post-hoc settings in MixupExplainer and ProxyExplainer. Furthermore, I have not yet found any new insight or theoretical contribution regarding GIB.**
> > >
> > > Response to Latest Comment W1：
> > >
> > > We thank the reviewer for their continued engagement. We are pleased to have the opportunity to further clarify the distinctions between the GIB formulation proposed in TAME and prior GIB-based methods, and to emphasize that our contributions encompass both a **task-agnostic GIB objective and task-agnostic structural mixup method**, forming a unified and highly generalizable explainer adaptable to diverse tasks.
> > >
> > > The GIB proposed by GSAT aims to find an explanatory subgraph by optimizing $\arg\min_{G^\*} I(G, G^\*) - \alpha I(Y, G^\*)$. However, because the model $f$ to be explained is trained on the original graph distribution but makes predictions on the OOD explanatory subgraph $G^\*$, the estimation becomes unreliable. MixupExplainer addresses the OOD issue by constructing a mixed graph $G^{(mix)} = G^\* + G^\Delta$, while ProxyExplainer introduces a variational graph auto-encoder to generate the proxy graph that approximates the original graph distribution. And they adhere to using the traditional cross-entropy loss $\text{CE}(Y, f(G^{(mix)}))$ to approximate $I(Y, G^\*)$ for an optimizable objective. RegExplainer, based on the continuous label assumption, proves that an InfoNCE-style loss is a lower bound for the $I(Y^\*, Y)$. Thus, existing methods' estimation of $I(Y, G^\*)$ **remains task-specific**. In contrast, TAME directly instantiates the Donsker-Varadhan representation with a critic function at the representation level to estimate the $I(Y, G^\*)$ in a **task-agnostic manner**, thus resulting in a unified GIB objective across diverse tasks. Additionally, built upon our proposed GIB objective, we introduce a **task-agnostic structural mixup framework**. The structural mixup framework, which employs hard perturbations, generates mixup graphs that are closer to the original graph distribution than soft-mask-based methods, resulting in more effective mitigation of distribution shift and consistently enhance the quality of the identified explanatory subgraphs across multiple tasks. This is substantiated by our OOD experiments in W3 & Q2 and AUC-ROC results in our paper experiment.
> > >
> > > In summary, TAME makes two complementary contributions: (i) a **theoretically grounded, task-agnostic GIB objective** derived from the DV representation and applicable to both discrete and continuous prediction tasks; and (ii) a structural mixup method that produces **in-distribution, non-redundant mixup graphs** and significantly improves the optimization of the GIB objective.

---

> > > ### Author Response · Authors · 2025-11-28
> > > **Response to Latest Comment W2**
> > >
> > > **W2.The original claim of being a task-agnostic explainer is also not novel. The authors mention negative sampling as a differentiating factor, but there is insufficient discussion on why the TAGE approach shows weak performance.**
> > >
> > > Response to Latest Comment W2:
> > >
> > > We sincerely thank the reviewer for their thoughtful follow-up and agree that a deeper discussion of the fundamental differences between TAME and TAGE is warranted. While TAGE contributes task-agnostic explanations via decoupling the explainer into embedding and downstream components, it fundamentally differs from TAME. The detailed differences are as follows.
> > >
> > > First, TAME derives a **task-agnostic GIB objective** through a rigorous theoretical formulation. In contrast, TAGE decouples the explainer into embedding and downstream components and relies on gradient-based techniques from the downstream explainer to achieve multi-task explanations. Notably, such gradient-based explainers (Grad) were originally introduced as a baseline in GNNExplainer [1] and are also included as a baseline in our experiments; however, its explanation quality may be limited. Second, the explainer of TAGE incurs an **OOD issue**, as discussed in W3 & Q2, TAGE generates explanation embeddings by feeding them into a GNN pretrained on the original graphs, and uses these embeddings in the InfoNCE loss. This introduces the **OOD issue**.
> > >
> > > Our structural mixup method not only effectively **mitigates this OOD issue** but also outperforms existing soft-mask-based methods[2][3], as demonstrated by our OOD experiments (W3 & Q2) and AUC-ROC results in our paper experiment. The mixup graphs generated by TAME exhibit higher cosine similarity and lower Euclidean distance to the original graph distribution. Third, TAME performs InfoNCE via representation-similarity-based sampling and structural mixup, positive pairs are formed by a target graph’s mixup graph and its near graph, while negative pairs consist of the mixup graph and its far graph. In contrast, TAGE defines positives as the target graph and its explanation subgraph, and negatives as the target graph paired with explanation subgraphs from other graphs.
> > >
> > > In summary, TAME combines a theoretically grounded task-agnostic GIB objective with a novel structural mixup strategy to mitigate OOD issues, yielding improved explanation quality over baselines across multiple tasks.
> > >
> > > [1]Ying, Zhitao, et al. "Gnnexplainer: Generating explanations for graph neural networks." Advances in neural information processing systems 32 (2019).
> > >
> > > [2]Zhang, Jiaxing, Dongsheng Luo, and Hua Wei. "Mixupexplainer: Generalizing explanations for graph neural networks with data augmentation." Proceedings of the 29th ACM SIGKDD conference on knowledge discovery and data mining. 2023.
> > >
> > > [3]Chen, Zhuomin, et al. "Generating in-distribution proxy graphs for explaining graph neural networks." arXiv preprint arXiv:2402.02036 (2024).

---

> > > ### Author Response · Authors · 2025-11-28
> > > **Response to Latest Comment W3**
> > >
> > > **W3.The main difference from TAGE is the structural mix-up. As discussed in the rebuttal for W2 of Reviewer N8oL, the assumption from the authors is quite limited for generalized settings since semantic similarity does not guarantee similar graph sizes.**
> > >
> > > Response to Latest Comment W3:
> > >
> > > We thank the reviewer for pointing out the limitation that structural mixup requires explanatory subgraphs of matching size. We provide further clarification regarding the feasibility of this design. In TAME, the target graph is mixed with both a near graph and a far graph sampled via embedding-similarity–based retrieval. Near graphs typically share similar labels and motif types with the target graph, and therefore tend to produce explanatory subgraphs of comparable size, allowing structural mixup to operate reliably.
> > >
> > > For far graphs, the explanatory subgraph sizes may differ. In practice, we apply a slightly larger top-k ratio during pooling so that the extracted subgraph forms a superset that reliably includes the relevant explanatory structure. Any redundant components are further refined by the MLP-generated mask. Table 1 reports the AUC results on datasets with varying motif sizes, showing that TAME consistently achieves strong performance on both classification and regression benchmarks.
> > >
> > > **Table 1. AUC results on datasets with varying motif sizes.**
> > >
> > > | Method     | Ba-HouseGrid      | BaHouse-AndGrid    | BA-Motif-Counting | House-OrGrid-Volume |
> > > |------------|--------------------|---------------------|--------------------|----------------------|
> > > | **TAME**   | 0.965 ± 0.026      | 0.979 ± 0.004       | 0.946 ± 0.004      | 0.941 ± 0.012        |
> > > | Runner-up  | 0.848 ± 0.039      | 0.806 ± 0.069       | 0.946 ± 0.095      | 0.741 ± 0.107        |
> > >
> > >
> > > We further conduct a case study in Appendix H.5.4 on the Ba-HouseGrid dataset, visualizing the structural mixup process when the target and neighbor graphs have differing explanation sizes. The visualizations demonstrate that TAME accurately identifies and exchanges the key motif regions for both near and far graphs, ultimately yielding faithful explanations.
> > >
> > > We would greatly appreciate any further suggestions or insights from the reviewer.

---

### Official Review · Reviewer_dyXP · 2025-10-31

**Soundness:** 3
**Presentation:** 4
**Contribution:** 3
**Rating:** 4
**Confidence:** 5

**Summary:**

This paper proposes a task-agnostic framework for explaining Graph Neural Networks (GNNs) named TAME,  which unifies explanation across both classification and regression tasks. TAME reformulates the Graph Information Bottleneck (GIB) framework through InfoNCE. And it introduces a structural mixup method based on graph pooling with a joint sampling strategy to generate in-distribution mixup graphs. Experiments show that TAME significantly outperforms prior explainers in accuracy and robustness.

**Strengths:**

1. This paper provides a rigorous theoretical foundation for deriving the proposed task-agnostic loss function.

2. The paper is easy to follow, and the figures are well-designed and visually appealing.

3. The experimental results in this paper support the method proposed in this paper.

**Weaknesses:**

1. The contribution of the paper appears limited. TAME mainly improves upon the structural sampling step of the previous structure mixup method like MixupExplainer. Can the author clarify in more detail how TAME is substantially different from MixupExplainer. In addition, please explain how the InfoNCE-based loss in TAME differs from the loss function used in RegExplainer.
2. As a post-hoc explanation method, the paper should consistently use the term “explainability” rather than “interpretability”, which is more appropriate in this context [1].
3. The Sampling Neighbors procedure is not clearly described. In particular, it is unclear how the negative neighbors $\left\{{y_j^-} \right\}_{j=1}^{N-1} are selected during the sampling process.
4. In the Introduction, the paper states that “soft-mask-based explanatory methods struggle to effectively drive edge weights toward a binary form.” Please clarify how TAME overcomes this limitation, given that it still employs a soft mask generated by an MLP.
5. The paper claims that TAME is task-agnostic. Could the authors discuss whether the proposed framework can be extended to other tasks such as node-level classification or edge-level link prediction?
6. The paper does not specify which explanation network architectures were used for MixupExplainer and RegExplainer when serving as baselines. Were their original explainer modules reimplemented according to their respective papers, or were they unified under the same explainer structure for fairness? Clarifying this is crucial for assessing the validity of the comparisons.

Reference: [1] Yuan, H., Yu, H., Gui, S., & Ji, S. (2022). Explainability in graph neural networks: A taxonomic survey. IEEE transactions on pattern analysis and machine intelligence, 45(5), 5782-5799.

**Questions:**

See weaknesses.

---

> ### Author Response · Authors · 2025-11-23
> **Response to W1**
>
> **W1:The contribution of the paper appears limited. TAME mainly improves upon the structural sampling step of the previous structure mixup method like MixupExplainer. Can the author clarify in more detail how TAME is substantially different from MixupExplainer. In addition, please explain how the InfoNCE-based loss in TAME differs from the loss function used in RegExplainer.**
>
> Response to W1:
>
> We thank the reviewer for raising this important concern. First, TAME fundamentally differs from MixupExplainer in its mixup approach. MixupExplainer is a soft-mask-based mixup method that constructs mixup graphs by fully merging the structures of the target graph and its neighbors, and then learns a soft mask over edges to identify the explanation and environment subgraphs. However, since the edge mask is optimized in an unsupervised manner, it is difficult to obtain a clean binarization, resulting in mixup graphs that retain a substantial amount of irrelevant and redundant information.
>
> In contrast, TAME employs a graph-pooling-based hard-perturbation mixup, where graph pooling identifies and crops out the explanation and environment subgraphs, which are then directly fused to generate mixup graphs without redundancy. A subsequent MLP is used to learn a soft mask to maintain consistency with prior explanation outputs. Experimental results on both regression and classification tasks show that, compared with soft-mask–based mixup methods, the mixup graphs generated by TAME better align with the original graph distribution and mitigate distribution shift more effectively.
>
> | Method                                               | Ba-HouseGrid | Ba-HouseAndGrid | Benzene | Fluorid-Carbonyl |
> | -------------- | ------------ | --------------- | ------- | ---------------- |
> | $\text{Cos}(h_G, h_{G^*}) \uparrow$           | 0.688        | 0.613           | 0.821   | 0.918            |
> | $\text{Cos}(h_G, h_{G_{\text{mixup}}})$ | 0.781        | 0.966           | 0.907   | 0.895            |
> | $\text{Cos}(h_G, h_{G_{\text{proxy}}})$ | 0.802        | 0.970           | 0.565  | 0.723           |
> | $\text{Cos}(h_G, h_{G_{\text{our}}^{\text{(mix)}}})$         | **0.868**        | **0.971**           | **0.941**   | **0.937**            |
> | $\text{Euc}(h_G, h_{G^*}) \uparrow$             | 1.044        | 1.436           | 0.953   | 0.684            |
> | $\text{Euc}(h_G, h_{G_{\text{mixup}}})$ | 0.973       | 0.477          | 0.771  | 0.782           |
> | $\text{Euc}(h_G, h_{G_{\text{proxy}}})$ | 0.706       | 0.4424          | 1.471  | 1.272           |
> | $\text{Euc}(h_G, h_{G_{\text{our}}^{(\text{mix})}})$           | **0.661**        | **0.425**           | **0.578**   | **0.604**            |
>
> | Method                                               | BA-Motif-Volume | BA-Motif-Counting | Crippen | Triangles |
> | ---------------------------------------------------- | --------------- | ----------------- | ------- | --------- |
> | $\text{Cos}(h_G, h_{G^*}) \uparrow$           | 0.696           | 0.663             | 0.869   | 0.897     |
> | $\text{Cos}(h_G, h_{G_{\text{reg}}})$ | 0.876           | 0.950             | 0.847   | 0.953     |
> | $\text{Cos}(h_G, h_{G_{\text{our}}^{\text{(mix)}}})$         | **0.953**           | **0.969**             | **0.936**   | **0.985**     |
> |$\text{Euc}(h_G, h_{G^*}) \uparrow$           | 1.138           | 1.196             | 0.653   | 0.389     |
> | $\text{Euc}(h_G, h_{G_{\text{reg}}})$ | 0.775           | 0.485             | 0.794   | 0.247     |
> | $\text{Euc}(h_G, h_{G_{\text{our}}^{(\text{mix})}})$         | **0.458**           | **0.400**             | **0.431**   | **0.143**     |
>
> Second, there is a main difference between TAME and RegExplainer in deriving the InfoNCE-based loss to approximate $I(G^\*, Y)$.
>
> RegExplainer is designed for regression, and its lower bound for $I(Y^\*,Y)$ relies on the assumption $\mathrm{sim}(Y^\*,Y)\propto p(Y|Y^\*)/p(Y)$, which is only well-defined for continuous outputs and is not parameterized for discrete labels. Its proof further assumes a deterministic mapping $Y^\*=h(G^\*)$ and explanation groups $G(y^\*)$, constructions tailored to continuous regression targets and not satisfied by categorical or softmax predictions.
>
> One of TAME's main contributions is a theoretically grounded, task-agnostic GIB objective. Its derivation uses the Donsker–Varadhan (DV) variational representation [1] to approximate $I(G^\*; Y)$, and the InfoNCE-based loss lower bound is derived by instantiating the critic function T in a label-agnostic manner, enabling a task-agnostic GIB objective. Moreover, TAME introduces a novel structural mixup strategy to generate in-distribution mixup graphs for computing the InfoNCE-based loss to optimize the GIB objective, whereas RegExplainer computes the InfoNCE-based loss via a soft-mask-based mixup approach.
>
> [1]Poole, Ben, et al. "On variational bounds of mutual information." International conference on machine learning. PMLR, 2019.

---

> ### Author Response · Authors · 2025-11-23
> **Response to W2,W3,W4,W6**
>
> **W2:As a post-hoc explanation method, the paper should consistently use the term “explainability” rather than “interpretability”, which is more appropriate in this context [1].**
>
> Response to W2:
>
> We thank the reviewer for the valuable suggestion. We have incorporated the corresponding updates in the revised version.
>
> **W3:The Sampling Neighbors procedure is not clearly described. In particular, it is unclear how the negative neighbors $\left{{y_j^-} \right}_{j=1}^{N-1} are selected during the sampling process.**
>
> Response to W3:
>
> We thank the reviewer for this question. The sampling process in TAME proceeds as follows. We first randomly sample N candidate graphs as neighbors from the candidate set, which is typically the entire dataset or the set of graphs to be explained. We then compute the similarity between each neighbor and the target graph, where the most similar neighbor is treated as the positive sample and the remaining ones as negative samples. In practice, we set N=2, using the near graph as the positive sample and the far graph as the negative sample.
>
> We have included the pseudocode for the neighbor-sampling procedure in Appendix D in the revised manuscript.
>
> **Algorithm: Similarity-based Graph Sampling**
>
> **Input:** Graph dataset $\\mathcal{G}$, pretrained GNN encoder $f$, target graph index $n$, sample size $N$
> **Output:** Positive graph $G^+$, negative graphs $\\{G^-_i\\}\_{i=1}^{N-1}$
>
> 1. For each $i = 1,\dots,|\\mathcal{G}|$:
>    Compute graph embedding
>    $$h_i \\leftarrow f(X_i, E_i)$$
> 2. For each pair $(i,j)$:
>    $$
>    \\text{Sim}[i,j] =
>    \\begin{cases}
>    1.0, & i = j \\\\
>    \\text{CosineSim}(h_i, h_j), & i \\neq j
>    \\end{cases}
>    $$
> 3. Sample a set
>    $$\\mathcal{S} \\leftarrow \\text{RandomSample}(\\mathcal{G}, N)$$
> 4. Select the positive graph
>    $$G^+ \\leftarrow \\arg\\max\\limits_{G_i \\in \\mathcal{S}} \\text{Sim}[n, i]$$
> 5. Define the remaining graphs as negatives
>    $$\\{G^-_i\\} _{i=1}^{N-1} \\leftarrow \\mathcal{S} \\setminus \\{G^+\\}$$
> **Return:** $G^+$ and $\\{G^-_i\\}\_{i=1}^{N-1}$
>
> **W4:In the Introduction, the paper states that “soft-mask-based explanatory methods struggle to effectively drive edge weights toward a binary form.” Please clarify how TAME overcomes this limitation, given that it still employs a soft mask generated by an MLP.**
>
> Response to W4:
>
> We thank the reviewer for pointing this out. Although both TAME and soft-mask–based explanation methods ultimately produce a soft mask, the role of the soft mask differs fundamentally.
>
> Soft-mask methods rely on the mask to separate and extract the explanation and environment subgraphs. As a result, when the edge weights cannot be cleanly binarized, the explanation and environment subgraphs become indistinguishable, redundant structures are retained, and the ability to address OOD issues is substantially weakened.
>
> In TAME, we instead use graph pooling to extract the explanation and environment subgraphs through a hard cropping process, effectively avoiding structural redundancy during mixup. The soft mask produced by the MLP in TAME is used to maintain consistency with prior explanation formats and to provide a smooth resampling effect, whereas the actual subgraph extraction is carried out by pooling rather than by the soft mask.
>
> **W6:The paper does not specify which explanation network architectures were used for MixupExplainer and RegExplainer when serving as baselines. Were their original explainer modules reimplemented according to their respective papers, or were they unified under the same explainer structure for fairness? Clarifying this is crucial for assessing the validity of the comparisons.**
>
> Response to W6:
>
> We thank the reviewer for raising this important concern. Both MixupExplainer and RegExplainer use a three-layer GCN as the model to be explained and a two-layer MLP as the explanation network architecture. When using them as baselines, we strictly adhered to the original implementations of them without any architectural modifications. Specifically, for classification tasks, we adopted the same experimental settings as MixupExplainer, and for regression tasks, we followed those of RegExplainer, ensuring fair and comparable results across all datasets.

---

> ### Author Response · Authors · 2025-11-23
> **Response to W5**
>
> **W5:The paper claims that TAME is task-agnostic. Could the authors discuss whether the proposed framework can be extended to other tasks such as node-level classification or edge-level link prediction?**
>
> Response to W5:
>
> We thank the reviewer for this valuable point. TAME introduces a task-agnostic GIB objective that supports explanation learning through representation-level contrastive training, and is therefore applicable to a broad range of tasks, including node classification and link prediction. For node-classification explanations, we extract a 3-hop ego subgraph centered on the target node as the target graph. For link-prediction explanations, we follow SEAL [1] to construct a local subgraph around the target edge. TAME then applies the same perturbation-based procedure as in graph-level tasks, identifying critical substructures within the target graph as explanations.
>
> For node-classification tasks, we follow the standard setup used in prior work [2], employing a three-layer GCN as the base model to be explained. We conduct experiments on the BA-Shapes, BA-Community, and Tree-Grids datasets, and compare TAME against MixupExplainer [3] and PGExplainer [4]. The results show that TAME consistently outperforms these baselines, generating in-distribution mixup graphs and producing faithful explanations.
>
> For link-prediction explanation, we use SEAL as the base model to be explained and implement edge-level variants of both TAME and PGExplainer. Experiments are conducted on BA-Shapes, BA-Community, and Tree-Grids. Because ground-truth explanations are unavailable for these datasets, we adopt FID+ and FID– [5] as evaluation metrics. The results show that TAME_link consistently outperforms PGExplainer_link, suggesting that TAME’s structural mixup mechanism effectively mitigates distribution shift and yields more precise explanations.
> Across node classification, graph classification, graph regression, and link prediction, the explanation results show that
>
> TAME consistently achieves the strongest performance. These findings suggest that the proposed structural mixup mechanism effectively alleviates distribution shift and facilitates better optimization of the GIB objective.
>
> | Method          | BA-Shapes        | BA-Community     | Tree-Grid       |
> |-----------------|----------------|----------------|----------------|
> | PGExplainer     | 0.902 ± 0.027  | 0.653 ± 0.111  | 0.205 ± 0.113  |
> | MixupExplainer  | 0.801 ± 0.145  | 0.654 ± 0.113  | 0.179 ± 0.054  |
> | TAME            | **0.991 ± 0.001**  | **0.906 ± 0.154**  | **0.678 ± 0.039**  |
>
> | Method             | BA-Shapes ($Fid_{\alpha_1,+} \uparrow$)     | BA-Shapes ($Fid_{\alpha_2,-} \downarrow$)     | BA-Community ($Fid_{\alpha_1,+} \uparrow$) | BA-Community ($Fid_{\alpha_2,-} \downarrow$) | Tree-Grid ($Fid_{\alpha_1,+} \uparrow$)    | Tree-Grid ($Fid_{\alpha_2,-} \downarrow$)    |
> |--------------------|-----------------------|------------------------|----------------------|-----------------------|----------------------|-----------------------|
> | PGExplainer_link   | 0.201 ± 0.197         | 0.285 ± 0.177          | 0.132 ± 0.135        | 0.229 ± 0.101         | 0.112 ± 0.048        | 0.178 ± 0.053         |
> | TAME_link      | **0.225 ± 0.139**     | **0.044 ± 0.077**      | **0.221 ± 0.101**    | **0.016 ± 0.038**     | **0.166 ± 0.057**    | **0.021 ± 0.028**     |
>
> [1]Zhang, Muhan, and Yixin Chen. "Link prediction based on graph neural networks." Advances in neural information processing systems 31 (2018).
>
> [2]Ying, Zhitao, et al. "Gnnexplainer: Generating explanations for graph neural networks." Advances in neural information processing systems 32 (2019).
>
> [3]Zhang, Jiaxing, Dongsheng Luo, and Hua Wei. "Mixupexplainer: Generalizing explanations for graph neural networks with data augmentation." Proceedings of the 29th ACM SIGKDD conference on knowledge discovery and data mining. 2023.
>
> [4]Luo, Dongsheng, et al. "Parameterized explainer for graph neural network." Advances in neural information processing systems 33 (2020): 19620-19631.
>
> [5]Zheng, Xu, et al. "Towards robust fidelity for evaluating explainability of graph neural networks." arXiv preprint arXiv:2310.01820 (2023).

---

### Author Response · Authors · 2025-12-02
**Summary for the Area Chair (cont)**

**Strengths Highlighted by Reviewers**

**1.Meaningful and Challenging Problem Setting**

Reviewers recognized that our work is motivated by an important and practical challenge in GNN explainability: existing explainers are largely **task-specific** and often rely on **soft-mask-based mixup mechanisms** that struggle to maintain structural consistency or produce reliable in-distribution mixup graphs. In contrast, TAME provides a theoretically grounded, **task-agnostic framework** enabled by a **structural mixup strategy**, offering a principled step toward more general-purpose GNN explanations [$\color{red}{\text{Reviewer dyXP, 1m9m, N8oL}}$].

**2.Strong Theoretical Foundation**

Reviewers noted that our derivation of the task-agnostic objective is supported by a rigorous theoretical foundation [$\color{red}{\text{Reviewer dyXP}}$]. In particular, the **reformulation of the GIB objective using InfoNCE** was highlighted as theoretically well-motivated and principled.

**3.Innovative Strategy for Addressing OOD Challenges**

Multiple reviewers emphasized the novel structural mixup strategy based on graph pooling, which performs hard structural perturbations to generate **in-distribution mixup graphs** [$\color{red}{\text{Reviewer dyXP, zcgv, N8oL}}$]. This was viewed as a meaningful and innovative design not present in prior work.

**4.Strong and Comprehensive Empirical Evidence**

Reviewers repeatedly emphasized the strong empirical performance of TAME, demonstrating clear improvements over prior explainers—both in **explanation quality and OOD mitigation**—across synthetic and real-world datasets [$\color{red}{\text{Reviewer dyXP, 1m9m, zcgv, N8oL}}$].

**5.Clear Writing and High-Quality Presentation**

The clarity and presentation quality of the paper—**easy to follow narrative, clear logic, and visually appealing figures**—were explicitly praised [$\color{red}{\text{Reviewer dyXP}}$].

---

We have deeply valued the opportunity to engage with the reviewers, whose insights have led to a marked improvement in the paper’s quality. We are confident that our active efforts to address the identified gaps have successfully resolved the reviewers’ concerns, and we believe the revised paper now presents a clearer and more solid contribution through its **task-agnostic framework** and **principled approach to distribution-shift issues** in GNN explanation. Given the consensus on the paper’s strengths and its rigorous theoretical grounding, we hope the revised manuscript is viewed as a valuable addition to the conference program.

---

### Author Response · Authors · 2025-12-02
**Summary for the Area Chair**

Dear Area Chair,

Thank you for taking the time to carefully consider our rebuttal. We are grateful for the reviewers’ constructive feedback, which has been invaluable in improving the clarity and quality of our manuscript. Throughout the discussion phase, we engaged fully with all reviewers and addressed their concerns in detail, which led Reviewer 1m9m to raise their initial rating. Although the discussion period was unexpectedly suspended, we continued to provide thorough clarifications to ensure that all remaining concerns were resolved to the best of our ability. To further support your assessment, we respectfully summarize below the key revisions and clarifications, as well as the main strengths highlighted by the reviewers.


**Key Revisions & Clarifications**

**1.Comprehensive Expansion of Experiments and Evaluation Metrics**

In response to reviewer concerns about **baseline coverage, task generalization, and evaluation reliability**, we substantially expanded our experimental evaluation across three dimensions. First, we integrated **TAGExplainer**  into the AUC-based quantitative evaluation[Lines 408-412; $\color{red}{\text{Reviewer 1m9m - Q1}}$] and added **MixupExplainer, ProxyExplainer, and RegExplainer** as baselines for the OOD analysis [Lines 378-395, 415-431; $\color{red}{\text{Reviewer 1m9m - Q2}}$]. Second, we extended our evaluation to include **Node Classification and Link Prediction** [Lines 1244-1279 in Appendix H.1; $\color{red}{\text{Reviewer dyXP - W5, zcgv - W3}}$]. Third, to ensure rigorous assessment, we implemented **SimOAR and Robust Fidelity metrics** [Lines 1281-1287, 1296-1311 in Appendix H.2; $\color{red}{\text{Reviewer N8oL - W3}}$]. Results across these settings confirm that our method **effectively mitigates distribution shifts and generates faithful explanations**.

**2.Visualization Analysis of Structural Mixup Strategy**

We added visualization analyses to evaluate the impact of **Graph Invariance** (performing mixup under shuffled node index orderings) and **Pooling Size** (handling varying explanatory subgraph sizes) on the effectiveness and stability of the proposed structural mixup strategy. [Lines 1566-1670 in Appendix H.5; $\color{red}{\text{Reviewer N8oL - W2, 1m9m - Follow-up Comment}}$].
These supplementary results show that the structural mixup mechanism consistently produces **in-distribution mixup graphs** and yields **accurate, reliable explanatory subgraphs**.

**3.Additional Discussions and Clarifications**

Specifically, we clarified the **theoretical novelty of our task-agnostic framework compared to prior methods** [$\color{red}{\text{Reviewers dyXP - W1, N8oL - W1}}$] and **rigorous derivations for the mutual information lower bound** [$\color{red}{\text{Reviewer zcgv - W5}}$]. In addition, we added **detailed pseudocode (Algorithm 2)** for the sampling strategy [Lines 886-897, 927-930 in Appendix D; $\color{red}{\text{Reviewer dyXP - W3}}$], specified **baseline architectures** [Lines 962-966 in Appendix E.1; $\color{red}{\text{Reviewer dyXP - W6}}$] and elucidated **methodological designs regarding MLP layer** and **sampling strategy** [$\color{red}{\text{Reviewer dyXP - W4, 1m9m - W2}}$]. Finally, we **standardized terminology** [Lines 016, 038, 088; $\color{red}{\text{Reviewer dyXP - W2, N8oL - W4}}$] and further provided detailed discussions on **computational scalability**, **hyperparameter selection, problem definitions** [$\color{red}{\text{Reviewer zcgv - W1, W2, W4, Q1, Q2}}$] and **analysis of OOD results** [$\color{red}{\text{Reviewer 1m9m - W3}}$].

---

### Meta-Review · Area_Chair_SbfY · 2026-01-06

**Summary:**

While reviewers appreciated the strong empirical performance of the proposed method on both synthetic and real-world benchmarks, they consistently raised concerns regarding the paper’s limited novelty and overall contribution, noting overlap in design choices with prior work. Additionally, reviewers identified conceptual limitations in the proposed mixup strategy. Although the rebuttal clarified several technical aspects and expanded the empirical evaluation, which led to modest score increases, the core concerns outlined above appear to remain insufficiently addressed. Therefore, I recommend rejection.

**Reviewer Concerns:**

Reviewers raised several concerns, including limited novelty, unclear applicability to other tasks (e.g., link prediction), lack of comparison with relevant baselines (e.g., TAGE), inappropriate use of the term “task-agnostic framework”, insufficient evaluation of the proposed structural mixup strategy in OOD settings, and conceptual issues with the mixup design. Also, additional clarifications were requested regarding the problem formulation, computational overhead, model selection, baseline implementation details, and negative neighbor sampling, among others.

During the rebuttal, the authors provided detailed responses that clarified several technical aspects, such as model selection, negative neighbor sampling, and the use of softmax in the proposed method. They also demonstrated applicability to node/link prediction tasks, included additional baseline comparisons (in Table 3), and expanded the evaluation of the structural mixup strategy relative to soft-mask-based approaches in OOD scenarios (using SimOAR and Fidelity metrics).

Nonetheless, I believe the following important concerns remain insufficiently addressed.

**Novelty and limited contribution**

Reviewers ``dyXP``, ``1m9m``, and ``N8oL`` expressed concerns regarding the novelty and overall contribution of the work. In particular, the use of a contrastive InfoNCE-based loss has already been explored in RegExplainer. Similarly, mixup-based ideas have been previously investigated by both RegExplainer and MixupExplainer, with the main distinction of TAME being the replacement of soft-mask-based mixup with a structural mixup strategy based on pooling. Moreover, reviewer ``1m9m`` noted that TAGE (NeurIPS 2022) had already introduced task-agnostic explainers via conditioned contrastive learning.

In response, the authors argue that their derivation relies on different assumptions and applies more broadly across settings involving continuous and discrete variables in a task-agnostic manner. However, given that the final objective function is identical to that of RegExplainer, reviewers remain unconvinced that the method constitutes a sufficiently novel or impactful contribution. Additionally, reviewer ``1m9m`` questioned both the novelty of the task-agnostic claim and whether the proposed differences in negative sampling and structural mixup adequately explain the reported empirical gains.

**Conceptual issues with the mixup strategy**

Given the similaritiy between the proposed objective and prior work, the structural mixup mechanism appears to be the main contribution. However, reviewers ``1m9m`` and ``N8oL`` highlighted limitations of this strategy, particularly: (i) the reliance on explanation subgraphs of equal size during mixup, and (ii) the lack of permutation invariance (i.e., dependence on arbitrary node orderings). To address (i), the authors argued that “near graphs” may have explanation subgraphs of similar size and provided empirical evidence to support their design choice. For (ii), the authors acknowledged the limitation and provided additional experiments, claiming that permutation sensitivity does not significantly affect performance, though without offering a compelling theoretical or intuitive justification. Overall, I found the responses somewhat handwavy and lacking a more principled or formal analysis of the proposed mixup strategy and its limitations.

**Reviewer Scores:**

Reviewer ``dyXP``. Most of the reviewer’s concerns were adequately addressed during the rebuttal, with the exception of the novelty issue. As a result, I believe this reviewer could have increased their score from 4 to 6.

Reviewer ``1m9m`` engaged during the discussion phase and updated their assessment from 2 to 4. As discussed above, concerns regarding limited novelty and conceptual shortcomings of the proposed mixup strategy prevented stronger support for acceptance. In my view, the additional author responses would unlikely change this assessment.

Reviewer ``zcgv``. This reviewer explicitly stated during the rebuttal that “there are still unresolved concerns regarding W2 (problem definition), W4 (model selection), and W5 (weak theoretical contribution),” and consequently maintained the original score of 6. While I believe the authors made reasonable efforts to address all raised concerns, it seems unlikely that this reviewer would become more supportive than their current stance.

Reviewer ``N8oL``. Echoing concerns raised by reviewer ``1m9m``, this reviewer participated in the discussion and noted that the authors’ initial responses were insufficient to resolve issues related to novelty and conceptual problems of the mixup strategy (e.g., lack of invariance). I anticipate that the authors’ subsequent replies, which mainly focused on additional empirical evidence, would not be sufficient to shift the reviewer toward supporting acceptance.

---

### Decision · Program_Chairs · 2026-01-26

Reject